# Single cell deciphering of progression trajectories of the tumor ecosystem in head and neck cancer

Z. L. Liu[1,6], X. Y. Meng[1,6], R. J. Bao[2,6], M. Y. Shen[2], J. J. Sun[3], W. D. Chen[4], F. Liu[5] & Y. He [ID][1] ✉

Head and neck squamous cell carcinoma is the sixth most common cancer worldwide and has high heterogeneity and unsatisfactory outcomes. To better characterize the tumor progression trajectory, we perform single-cell RNA sequencing of normal tissue, precancerous tissue, early-stage, advanced-stage cancer tissue, lymph node, and recurrent tumors tissue samples. We identify the transcriptional development trajectory of malignant epithelial cells and a tumorigenic epithelial subcluster regulated by *TFDP1*. Furthermore, we find that the infiltration of *POSTN*⁺ fibroblasts and *SPP1*⁺ macrophages gradually increases with tumor progression; their interaction or interaction with malignant cells also gradually increase to shape the desmoplastic microenvironment and reprogram malignant cells to promote tumor progression. Additionally, we demonstrate that during lymph node metastasis, exhausted CD8⁺ T cells with high *CXCL13* expression strongly interact with tumor cells to acquire more aggressive phenotypes of extranodal expansion. Finally, we delineate the distinct features of malignant epithelial cells in primary and recurrent tumors, providing a theoretical foundation for the precise selection of targeted therapy for tumors at different stages. In summary, the current study offers a comprehensive landscape and deep insight into epithelial and microenvironmental reprogramming throughout initiation, progression, lymph node metastasis and recurrence of head and neck squamous cell carcinoma.

Therapeutic strategies for head and neck squamous cell carcinoma (HNSCC) have undergone a paradigm shift from chemotherapy, radiotherapy, and targeted therapy to immunotherapy, through the targeting of tumor cells and immune checkpoints (PD-1, PD-L1, and CTLA4). Nevertheless, all of these treatments have limited effectiveness and unsatisfactory outcomes, which may be ascribed to intratumor, intertumor and intercellular heterogeneity, reflecting complex cancer–immune–stromal communication in the tumor microenvironment (TME). Specifically, therapeutically targeted pathways are not restricted to cancer cells but also involve other cellular

[1]Department of Oral Maxillofacial & Head and Neck Oncology, Shanghai Ninth People's Hospital, Shanghai Jiao Tong University School of Medicine; College of Stomatology, Shanghai Jiao Tong University; National Center for Stomatology; National Clinical Research Center for Oral Diseases; Shanghai Key Laboratory of Stomatology Shanghai, Shanghai 200011, China. [2]Shanghai Institute of Immunology, State Key Laboratory of Oncogenes and Related Genes, Department of Immunology and Microbiology, Shanghai Jiao Tong University School of Medicine, Shanghai 200025, China. [3]Department of Oral Pathology, Shanghai Ninth People's Hospital, Shanghai Jiao Tong University School of Medicine; College of Stomatology, Shanghai Jiao Tong University; National Center for Stomatology; National Clinical Research Center for Oral Diseases; Shanghai Key Laboratory of Stomatology Shanghai, Shanghai 200011, China. [4]Novel Bioinformatics Co., Ltd, Shanghai, China. [5]Department of Oncology, Shanghai Ninth People's Hospital, Shanghai Jiao Tong University School of Medicine, Shanghai 200011, China. [6]These authors contributed equally: Z. L. Liu, X. Y. Meng, R. J. Bao. ✉e-mail: william5218@126.com

components in the TME; this topic has puzzled cancer biologists and clinicians for some time[1,2]. High transcriptomic heterogeneity and diverse landscapes of the TME within or between tumors are intimately associated with poor prognosis[3]. Given the potential impact of the TME on patient care, a better understanding of the ecosystem diversity in HNSCC may lead to a compelling approach for promoting therapeutic efficacy.

Recent advances in single-cell transcriptomics have provide an avenue for dissecting the constitutive and functional heterogeneity of HNSCC at an individual-cell resolution[4]. Based on single-cell RNA sequencing (scRNA-seq), investigators have identified the partial epithelial-mesenchymal transition (p-EMT) program in HNSCC, observed the localization of p-EMT cells to the leading edge near cancer-associated fibroblasts (CAFs), determined the transcriptional profiles of nonimmune and immune populations within tumors of HPV⁻ and HPV⁺ HNSCC, and investigated the immune infiltrate in HNSCC and site-matched inflamed tissues in efforts to elucidate the transformation from inflammation to carcinoma[1,5–7]. These efforts have provided insight into the distinct landscapes of the TME between normal tissue and malignant tissue, laid the foundation for a renewed understanding of cancer biology, and elucidated candidate targets for cancer therapy.

Nonetheless, the dynamic profiling of HNSCC initiation and progression has not been fully elucidated. Choi *et al*. explored the ecosystem of HNSCC evolution and focused on tumor cells, CAFs, and Treg cells. They reported that fibroblast-derived COL1A1 interacts with CD44 in malignant cells and that *CXCL8*-expressing CAFs and *LAIR2* expression in Treg cells are associated with HNSCC progression[8]. They mainly concentrated on the precancerous status and the development from the primary tumor to lymph node metastasis and has contributed to the in-depth understanding of HNSCC microenvironment. However, since HNSCC progresses stepwise from normal tissue (NT) to precancerous lesions (pre-Ca), followed by early cancer (E), advanced cancer (A), recurrent cancer (R), and ultimately lymph node (LN) metastasis and distant spread (*e.g.*, the lung), there are several key scientific issues have not been settled with regard to the HNSCC progression: Firstly, how HNSCC initiates from precancerous lesions and progresses to advanced disease has not been fully characterized. Secondly, how tumor cells, CAFs, macrophages, and T cells, the most important cell types in the TME of HNSCC, undergo reprogramming and aberrant cross-talking thus contributing to HNSCC progression have not been elucidated. Thirdly, extranodal extension (ENE) is the most serious situation during lymph node metastasis and indicates the most advanced stage for locoregional metastasis and signifies a particularly poor prognosis for HNSCC patients. Overall, knowledge of the microenvironment features of lymph nodes with ENE and the underlying mechanism of ENE are scarce[9]. Fourthly, the differences in the ecosystem between primary and recurrent tumors and whether this diversity plays a determining role in targeted therapeutic selection are unclear.

In this work, we perform scRNA-seq profiling of normal tissues, precancerous tissues, early-stage cancer, advanced-stage cancer, recurrent cancer, and metastatic LN (ENE⁺/ENE⁻) tissues to dissect the dynamic transition of the TME in HNSCC and attempt to address the abovementioned limitations. We subsequently explore the developmental trajectory of HNSCC and assess dynamic alterations in the infiltrative proportion and the biological function of malignant cells, immune cells and stromal cells, contributing to a comprehensive understanding of the HNSCC ecosystem during tumor initiation, progression, lymph node metastasis, and recurrence.

## Results
### A single-cell expression atlas of HNSCC ecosystems during tumor progression
To comprehensively explore tumor ecosystem heterogeneity across various stages of HNSCC, including initiation, progression, lymph node metastasis, and recurrence, we performed scRNA-seq (10X Genomics) to profile malignant and nonmalignant cells from 26 fresh specimens from 13 patients (Fig. 1A), including 3 adjacent normal tissues (NT), 3 precancerous lesions (Pre), 3 early-stage tumors (E), 6 advanced-stage tumors (A), 3 intracapsular metastatic (ENE⁻) lymph nodes (LN-in), 2 extracapsular metastatic (ENE⁺) lymph nodes (LN-out), 2 normal lymph nodes (LN-normal), and 4 recurrent tumors (R), which span the cascade from normal epithelial to local advanced and metastatic cancer and provide a relatively comprehensive collection of tissues mirroring the tumor progression process (Supplementary Fig. 1a). Among them, samples from patients with stage NT-Pre-E disease were paired; advanced-stage tumors and metastatic lymph nodes were also paired. All diagnoses were made after careful pathological investigation, and the detailed demographic and clinical information of the patients can be found in Supplementary Data 1.

After strict quality control and filtration, a total of 120, 952 single cells with a median of 1, 642 expressed genes were retained for subsequent analysis. We integrated all cells from various stages of HNSCC using Harmony[10] to remove the batch effect, performed graph-based clustering and used marker-based annotation to define each cluster. All cells were classified into 9 major cell clusters, according to the uniform manifold approximation and projection (UMAP) tool (Fig. 1B, Supplementary Fig. 1b), which included epithelial cells (n = 11, 722) identified by *EPCAM* and *CDH1*; T cells (n = 42, 609) expressing the T-cell receptor (TCR) signaling mediators *CD3D*, *CD3E*, and *CD3G*; myeloid cells (n = 20, 306) marked as *CD14* and *FCGR3A*; plasma cells (n = 5, 146) defined by *SDC1* and *MZB1*; B cells (n = 10, 177) annotated by *CD19* and *MS4A1*; fibroblast cells (n = 15, 291) positive for *COL1A1* and *COL3A1*; endothelial cells (n = 10, 005) positive for *PECAM1* and *CDH5*; pericytes/SMCs (n = 5, 310) marked as *MCAM* and *RGS5*; and Schwann cells (n = 386) characterized as expressing *GPM6B* and *S100B* (Fig. 1C). The cell composition and infiltration fraction of these 9 main cell types in tissues across diverse pathological stages revealed notable heterogeneity, as evidenced by the different distributions, numbers, and percentages of the individual cell clusters (Fig. 1D, E).

In summary, we herein provide an overview of the HNSCC microenvironment during tumor initiation, progression, lymph node metastasis, and recurrence. The results suggest that HNSCC ecosystems are highly heterogeneous, which was further investigated in our subsequent analyses.

### A specific malignant cell cluster determining the invasive phenotype
Given the central role of malignant cells in tumor progression and immunosuppression, we investigated the transcriptomic diversity of these pivotal cell types. As shown in Fig. 2A, the expression patterns of epithelial cells exhibited substantial heterogeneity. Subsequently, 7, 054 malignant epithelial cells of epithelial origin, and 4, 336 nonmalignant cells were identified according to the Copy Number Karyotyping of Aneuploid Tumors (CopyKAT) algorithm (Fig. 2B, Supplementary Fig. 2a)[11]. The epithelial cells from Pre, E, and A stage samples contained a greater proportion of cycling cells than did those from normal tissues (Supplementary Fig. 2b). Interestingly, a proportion of epithelial cells in the NT sample were identified as aneuploid. Therefore, we re-analyzed epithelial cells in NT, pre-, and E-stage samples of representative patient P13. The results showed that, generally, the predicted copy numbers of epithelial cells in the NT samples were smaller than those in the pre- and E-stage samples (Supplementary Fig. 2c). Among the aneuploid epithelial cells in the NT samples, genes related to oncogenesis processes such as 'cell growth', 'epithelial cell proliferation', and 'Wnt signaling pathways' were upregulated compared to those in their diploid counterparts (Supplementary Fig. 2d, e). These results suggested that aneuploidy epithelial cells in NT samples may be in a transitional status from bona fide normal cells to precancerous cells. Then, we examined the tumor progression

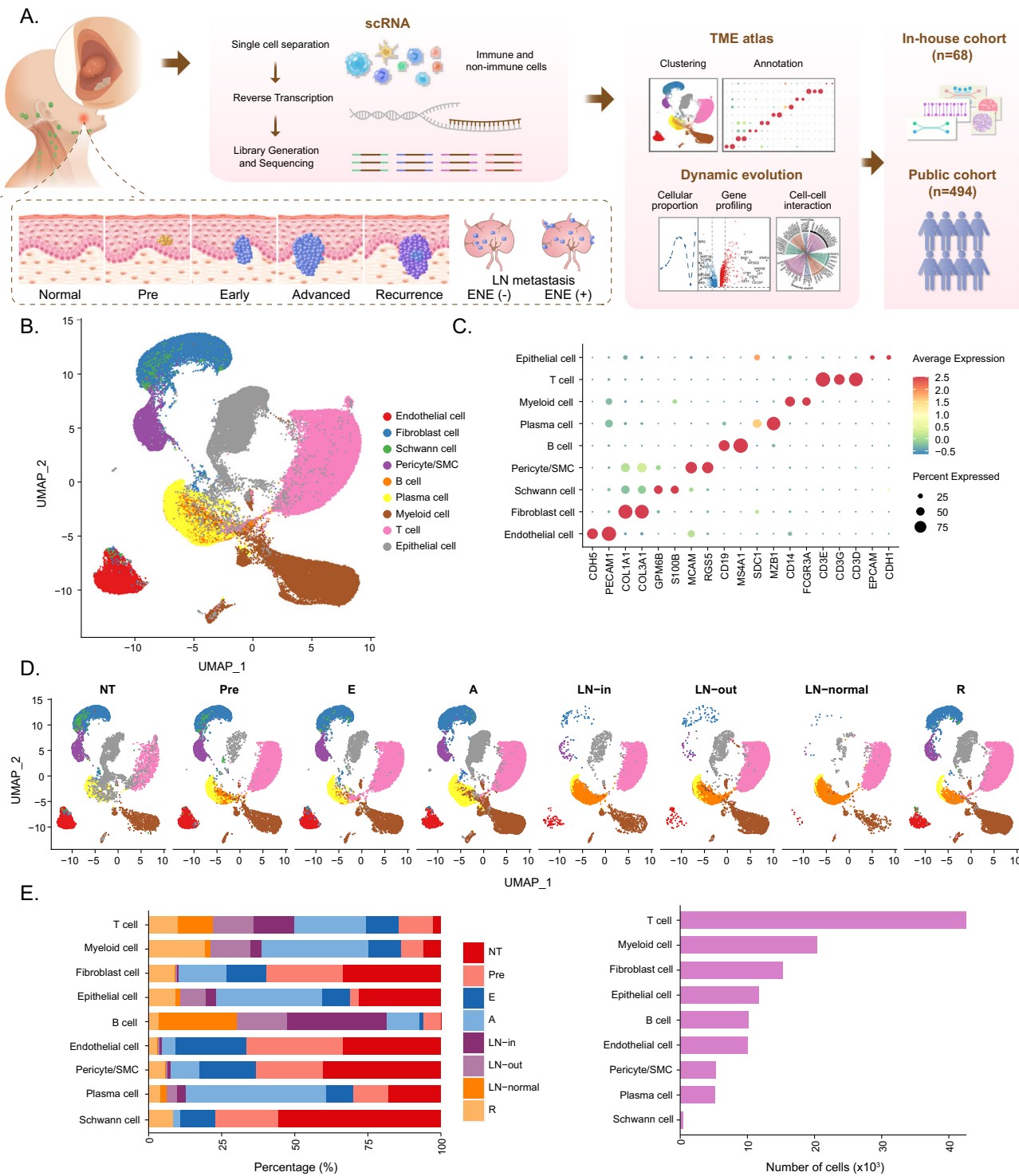

**Fig. 1 | Single-cell expression profiling of human HNSCC. A** A schematic graph shows the study design. **B** Uniform manifold approximation and projection (UMAP) visualization of 120952 cells from 26 samples, showing 9 clusters in different colors. **C** Dot plots show average expression of known markers in indicated cell clusters. The dot size represents percent of cells expressing the genes in each cluster. The expression intensity of markers is shown. **D** UMAP visualization of 17527 cells from 3 NT samples, 16726 cells from 3 Pre samples, 14437 cells from 3 E samples, 29186 cells from 6 A samples, 10920 cells from 3 LN-in samples, 11824 cells from 2 LN-out samples, 8564 samples from 2 LN-normal samples, and 11768 cells from 4 R samples. **E** Proportion of 9 major cell types showed in bar plots in different sample types (left panel) and total cell number of each cell type (right panel) are shown. Source data are provided as a Source Data Fig. 1B–E.

process. Epithelial cells were found to express cytokines in a stage-dependent manner; for instance, *CXCL14*, *IL-18*, and *TYMP* were consistently upregulated across the Pre, E, A, and R stages. However, protumor cytokines, such as *TNFRSF12A*, *PLAU*, and *SDC1*, were found mainly in A/R stage tumors and metastatic lymph nodes. Moreover, *EGFR*, *SAA1* and *SAA2* were specifically expressed in ENE⁺ lymph nodes (Supplementary Fig. 2f). These results reflect the self-renewal and phenotypic transition of epithelial cells during the multistep development of HNSCC, which may be pivotal contributors to tumor progression[12–14].

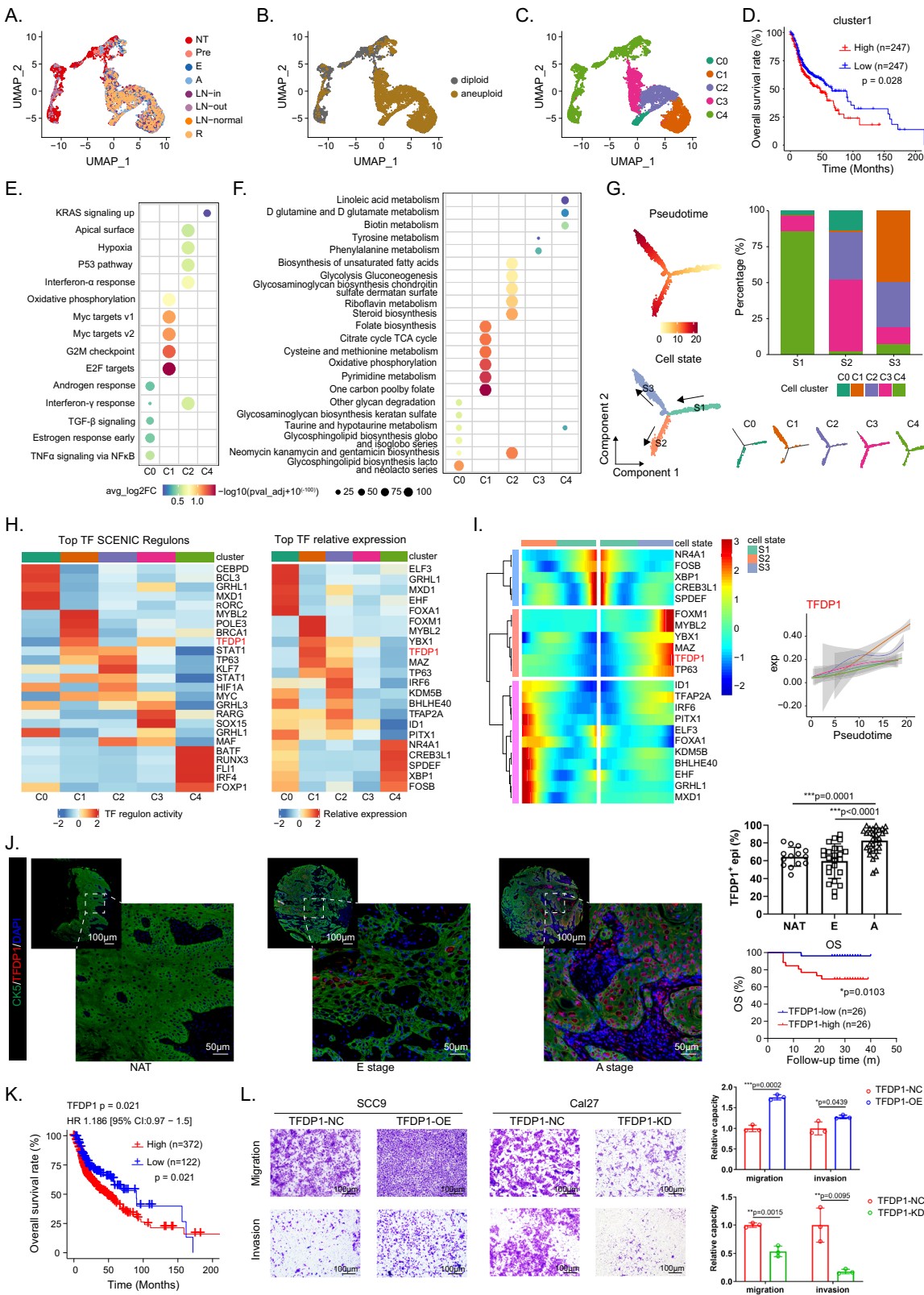

To probe transcriptional heterogeneity, epithelial cells were then classified into 5 subclusters termed C0 to C4 (Fig. 2C, Supplementary Fig. 2g). Based on the copy number variation (CNV) profile, cluster 4 was uniquely present in the NT samples as the major cell subtype; clusters 1 and 2 mainly appeared in tumor tissues. We constructed gene signatures of different clusters using CibersortX algorithm. The abundance of cluster 1 significantly associated with unfavorable overall

survival (OS) in the bulk RNA-seq data of The Cancer Genome Atlas (TCGA)-HNSCC cohort (Fig. 2D, Supplementary Fig. 2h–j), indicating that cluster 1 acquired an aggressive phenotype, and promoted HNSCC progression. To further determine the biological characteristics of the different epithelial cells, we calculated the enrichment scores of the hallmark gene sets from the Molecular Signatures Database (MsigDB)[15] and Kyoto Encyclopedia of Genes and Genomes (KEGG) pathways and

**Fig. 2 | Identification and transcriptional characterization of a malignant epithelial cluster.** UMAP projection of 7054 epithelial cells from 26 samples colored by groups (**A**), diploid/aneuploid status (**B**), and clusters (**C**). **D** The Kaplan-Meier curves showed patients with higher infiltration of cluster1 epithelial cells are associated with worse overall survival (OS) in TCGA-HNSCC cohort ($n = 494$, 247 samples for each group). Dot plot of top 5 hallmarks (**E**) and 6 metabolic pathways (**F**) for differentially expressed genes (DEGs) in each epithelial cluster. **G** Potential trajectory of epithelial cells inferred by Monocle2. The trajectory was divided into three states indicated as S1, S2, and S3. **H** Heatmap shows normalized activity of top 5 transcription factors (TF) regulons predicted by the SCENIC algorithm (left) and the relative expression (z-score) of top 5 TF genes (right) in epithelial cell clusters. **I** Heatmap shows the dynamic changes in TF and regulons expression along the pseudotime. The expression of *TFDP1* is shown in the right panel. **J** Representative images of

multiplex immunohistochemistry (mIHC) staining of *TFDP1*$^+$ epithelial cells in HNSCC tumor and nonmalignant samples. Scale bar, 100 µm and 50 µm. The quantitative results are shown on the right. Upper line: *TFDP1*$^+$ epithelial cell ratio in different stages ($n = 69$). Lower line: The Kaplan-Meier OS curves of validation cohort patients stratified by *TFDP1*$^+$ expression level ($n = 52$, 26 samples for each group). **K** The Kaplan-Meier curves of samples with high ($n = 372$) and low ($n = 122$) *TFDP1* expression level in TCGA-HNSCC cohort ($n = 494$). **L** Images of Transwell assays for migration and invasion in different cell lines with *TFDP1* overexpression or knockdown. Scale bar = 100 µm. The quantitative analysis is shown on the right. $n = 3$ biologically independent experiments. Data represent mean ± SD. *P* values were calculated by two-side Student's *t*-test in **J** and **L**, by one-way ANOVA test in **E** and **F**, and by two-sided log-rank test in **D**, **J** and **K**. Source data are provided as a Source Data Fig. 2A–L.

analyzed the differences among the clusters of epithelial cells. Clusters 1 and 2 were composed mainly of cells from stages A, E, and R but exhibited significant differences in function. Cluster 1 exhibited significant enrichment of *MYC* target expression, *G2M* checkpoint, and *E2F* target expression, indicating increased cell proliferation and division rate (Fig. 2E). Then we contextualized our tumor subpopulations with malignant cell signatures in Puram's work[5] and found cluster 1 had high expression of the cell cycle signature, indicating that it had a more malignant phenotype than the other clusters (Supplementary Fig. 2k). This discovery was also confirmed by Gene Ontology (GO) analysis, which revealed that nuclear division was obviously enriched in cluster 1 and that keratinocyte differentiation and cornification were significant in cluster 2 (Supplementary Fig. 2l, m). To further uncover metabolic diversity, we observed that cluster 1 was enriched in the citrate cycle, oxidative phosphorylation, folate biosynthesis, and pyrimidine metabolism. However, cluster 2 exhibited high expression of the glycolysis pathway, demonstrating metabolic heterogeneity in different cancer cell subtypes (Fig. 2F).

To better understand the trajectory of malignant epithelial cluster development, we applied the monocle2 and RNA velocity algorithms in a two-dimensional UMAP representation to visualize the status transition of epithelial cells along a dynamic biological timeline. The pseudotime trajectory produced a prominent linear (right to left) trajectory with 2 branches (up-State 3 and bottom-State 2), indicating 2 developmental fates of malignant cells both starting from State 1, with the lowest pseudotime value (Fig. 2G). Among them, cluster 1 was localized almost entirely in State 3, skewing toward the ends of the trajectory; this was verified by RNA velocity analysis, which showed that cluster 1 originated from cluster 2 (Supplementary Fig. 2n, o). We also performed the trajectory analysis of epithelial cells obtained from the same patient (P2, P10, and P13) with NT, pre, and E samples and found consistent developmental trajectories of these three patients with respect to total malignant epithelial cells (Supplementary Fig. 2p–r). The developmental trajectory of cluster 1 is orchestrated by a comprehensive network of transcription factors (TFs) that regulate each other and their effectors by interacting with their cofactors and downstream genes. Therefore, we evaluated the 5 most highly expressed TFs and 5 most strongly expressed TFs in the TF regulatory network via the SCENIC algorithm[16] (Fig. 2H). The most prominent finding was that *TFDP1* was highly activated in cluster 1 and was the top-ranked TF regulon, which was consistent with the trend revealed by the heatmap and tumor samples in our validation cohort (Fig. 2I, J, Supplementary Fig. 2s). *TFDP1* participates in cell proliferation and DNA damage repair and has been considered to contribute to tumor progression in breast cancer and hepatocellular carcinoma[17]. Prognostic analysis of the TCGA-HNSCC cohort and in-house validation cohort both demonstrated that the abundance of *TFDP1* was associated with worse outcomes, suggesting that *TFDP1* promotes HNSCC development (Fig. 2J, K, Supplementary Fig. 2t, u). We confirmed this with an

in vitro transwell assay. When tumor cells were induced to overexpress *TFDP1*, their migration and invasion capacity were significantly increased, and vice versa (Fig. 2L).

In summary, we identified a group of malignant epithelial cells that may be regulated by *TFDP1* during HNSCC progression and promote cancer development.

## Delineation of the dynamic microenvironment landscape of HNSCC

To systematically contextualize both immune and stromal compartment shifts in the cascade from normal tissue to local advanced tumors, we attempted to comprehensively elucidate the dynamic transformation of the TME during HNSCC progression. Fibroblasts are the main stromal cells and play a critical role in TME remodeling. Hence, we further stratified the 15, 291 fibroblasts into 8 subsets according to representative gene signatures (Fig. 3A, B, Supplementary Fig. 3a). Common fibroblast markers such as *COL1A1* and *COLA12*, were found to be expressed across all 8 subpopulations, confirming their fibroblastic cell identity. In addition to feature low fibroblasts, other subpopulations of fibroblasts exhibited significant expression of the following marker genes: *RSPO1*$^+$ fibroblasts, which were characterized by *RSPO1* and *CRABP1*; *POSTN*$^+$ fibroblasts, which exhibited specific expression of *POSTN* and *LAMP5*; *DES*$^+$ myofibroblasts, which were positive for *DES* and *MYF5*; proliferating fibroblasts, which were characterized by *MKI67* and *TOP2A*; and *CCL19*$^+$ fibroblasts, which exhibited high expression of inflammatory genes (*e.g.*, *CCL2* and *CCL19*).

Among these subclusters, *RSPO1*$^+$ fibroblasts and *POSTN*$^+$ fibroblasts exhibited opposite proportions of change trends and were associated with opposite prognoses during stepwise progression of HNSCC (Fig. 3C, D, Supplementary Fig. 3c). The proportion of *RSPO1*$^+$ fibroblasts gradually decreased during HNSCC progression, whereas the proportion of *POSTN*$^+$ fibroblasts gradually increased from the NT to the A stage and a signature score in the top 50 was associated with a worse prognosis ($p = 0.03$).

Myeloid cells are a main determinant of the intratumoral immune landscape and contribute to tumorigenesis and therapeutic resistance[18]. A total of 20, 306 myeloid cells in our cohort were further clustered into 11 individual subpopulations according to the reported marker genes (Fig. 3E, F, Supplementary Fig. 3b). Four clusters of dendritic cells (DCs) were identified: *LAMP3*$^+$ DCs, pDCs, cDC1s, and cDC2s. Unsupervised clustering of macrophages with *CD68* expression revealed *FOLR2*$^+$ macrophages, *SPP1*$^+$ macrophages, *C1QC*$^+$ macrophages, and *CXCL10*$^+$ macrophages in the TME of HNSCC patients. The other myeloid cells were quantified as proliferating myeloid cells (characterized by *MKI67*), mast cells (characterized by *TPSB2* and *CMA1*), and monocytes (characterized by *CD14*, *FCN1* and *VCAN* but lacking *CD68*). Like those of fibroblasts, the proportions of *FOLR2*$^+$ macrophages and *SPP1*$^+$ macrophages exhibited opposite trends, as did the prognostic significance (Fig. 3G, H, Supplementary Fig. 3d). The frequency of *FOLR2*$^+$ macrophages decreased and that of *SPP1*$^+$

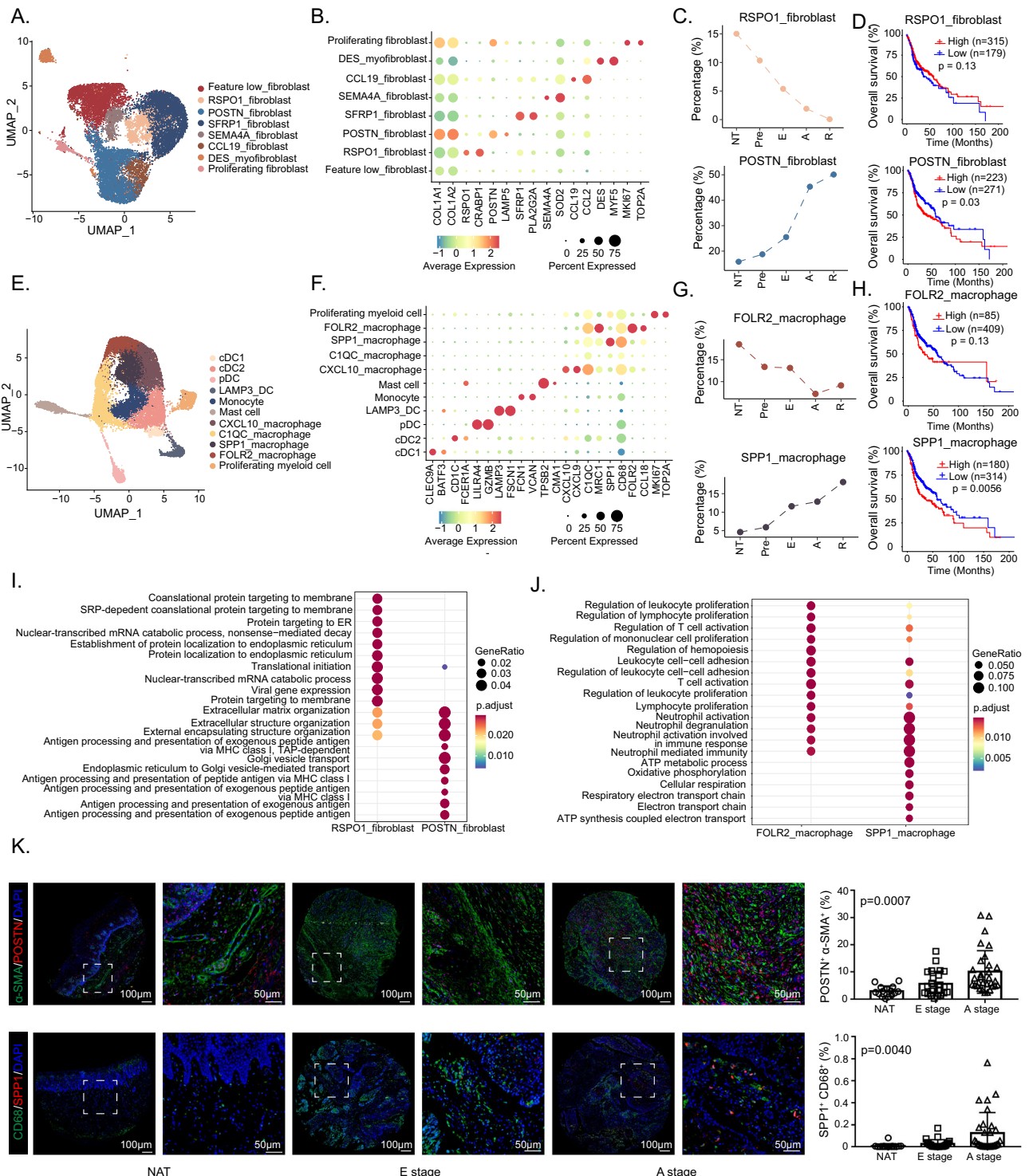

**Fig. 3 | Characterization of fibroblasts and myeloid cells during HNSCC progression.** UMAP plots show the composition of 15291 fibroblasts (**A**) and 20306 myeloid cells (**E**) from 26 samples. Dot plots show the expression of common marker genes as well as top 2 most variable genes across each fibroblast (**B**) and myeloid cell (**F**) subset. (**C**) The infiltration proportion of *RSPO1*[+] fibroblasts (above) and *POSTN*[+] fibroblasts (below). **D** The Kaplan-Meier overall survival curves of TCGA-HNSCC patients stratified by *RSPO1*[+] fibroblasts (above) and *POSTN*[+] fibroblasts (below) infiltration, *n* = 494. **G** The infiltration proportion of *FOLR2*[+] macrophages (above) and *SPP1*[+] macrophages (below). **H** The Kaplan-Meier overall survival curves of HNSCC patients stratified by *FOLR2*[+] macrophages (below) and *SPP1*[+] macrophages (below) infiltration, n = 494. **I** Bubble plot shows comparison of

DEG enrichment GO terms between *RSPO1*[+] fibroblasts and *POSTN*[+] fibroblasts. (**J**) Bubble plot shows comparison of DEG enrichment GO terms between *FOLR2*[+] macrophages and *SPP1*[+] macrophages. **K** Representative images of mIHC staining of *POSTN*[+] fibroblasts (POSTN[+] α-SMA[+] double positive) and *SPP1*[+] macrophages (SPP1[+] CD68[+] double positive) in HNSCC tumor and nonmalignant samples. Scale bar = 50 μm. The quantitative results are shown on the right. *n* = 70 for *POSTN*[+] fibroblasts, *n* = 68 for *SPP1*[+] macrophages. Data represent mean ± SD. *P* values were calculated by two-sided Wilcoxon signed-rank test in **I** and **J**, by one-way ANOVA test in K, and by two-sided log-rank test in D and H. Source data are provided as a Source Data Fig. 3A–K.

macrophages increased across multiple aggressive tumors. Moreover, the top 50 signature score of *SPP1*+ macrophage signature was significantly associated with worse overall survival (*p* = 0.0056).

To further explore the biological functional differences between these two pairs, we conducted GO analysis and discovered that genes highly expressed in *RSPO1*+ fibroblasts are related to the endoplasmic reticulum, translation initiation, and membrane protein and that pathways apparently enriched in *POSTN*+ fibroblasts are related to extracellular matrix (ECM) organization, and antigen processing and presentation, suggesting that *POSTN*+ fibroblasts shape a more aggressive phenotype of HNSCC through ECM remodeling and immune regulation (Fig. 3I). We contextualized fibroblast subpopulations according to currently known subtypes[5,19,20] and found *POSTN*+ fibroblasts tended to be myCAFs, dCAFs and CAF1 cells related to desmoplastic component production (Supplementary Fig. 3e–g).Differentially expressed genes (DEGs) and GO analysis between *FOLR2*+ and *SPP1*+ macrophages revealed that genes upregulated in the former participate in immune activation, such as leukocyte proliferation, T-cell activation, and lymphocyte proliferation, and that the genes highly expressed in the latter are related to neutrophil activation, ATP metabolic process, oxidative phosphorylation and respiratory electron transport chain (Fig. 3J). Additionally, *SPP1*+ macrophages showed a higher M2 score than *FOLR2*+ macrophages[21] (Supplementary Fig. 3h). These results indicate that *FOLR2*+ macrophages may exhibit the biological function of T-cell recruitment as well as activation but that *SPP1*+ macrophages represent specific metabolic features and may affect the TME in a way different from that of *FOLR2*+ macrophages. Moreover, in a validation cohort of 68 HNSCC samples, we confirmed the prevalence of *POSTN*+ fibroblasts and *SPP1*+ macrophages in the tumor stroma. The percentage of *POSTN*+ fibroblasts with *SPP1*+ macrophages increased during HNSCC progression (*p* < 0.001 for *POSTN*+ fibroblasts, *p* = 0.004 for *SPP1*+ macrophages), which is consistent with the in silico results (Fig. 3K).

Other cell types in the TME include CD4+ T cells, CD8+ T cells, B cells, which are immune components and endothelial cells, which are stromal components. Regarding the immune compartment, we identified the following: 7 subpopulations of CD4+ T cells - CD4-central memory T cells (Tcm), CD4-effector memory T cells (Tem), Th17, T follicular helper cells (Tfh), Tfh/Th1, ISG-CD4 T, and Treg cells (Supplementary Fig. 3i); 7 clusters of CD8+ T cells - CD8-resident memory T cells (Trm), CD8-Tcm, CD8-Tem, CD8-recently activated effector memory T cells (Temra), CD8-Tex, ISG-CD8 T, and proliferating T cells (Tprof) (Supplementary Fig. 3j); and 3 subsets of B cells - naïve B cells, memory B cells, and germinal center B cells (GCB) (Supplementary Fig. 3k). We found an overall trend of elevated Treg infiltration with decreasing Th17, CD4_Tem, and CD8_Tem proportions during HNSCC progression, which indicated the transition from an immune-activated TME to an immune-exhausted TME. Interestingly, the cell proportion of CD8_Tex decreased in A stage compared to E stage, consistent with the findings of a recent report of "partial immune recovery" in advanced tumors (Supplementary Fig. 3i, j)[22]. With respect to the stromal component, we subclassified endothelial cells into 6 clusters: immature ECs, arterial ECs, capillary ECs, venous ECs, tip ECs, and lymphatic ECs (Supplementary Fig. 3l). Immature ECs have recently been identified in NSCLC and are associated with angiogenesis, and an increasing proportion of these cells may be linked to angiogenic cell proliferation and reprogramming during HNSCC development[23].

Overall, we established a dynamic microenvironment landscape during HNSCC progression. Among nonmalignant cell types, *RSPO1*+ fibroblasts and *FOLR2*+ macrophages were identified as potential tumor-suppressing populations and *POSTN*+ fibroblasts and *SPP1*+ macrophages were identified as tumor-promoting populations.

## The cellular network of *POSTN*+ fibroblasts, *SPP1*+ macrophages and malignant cells during HNSCC progression

To further investigate the underlying mechanism by which dynamic changes in cellular composition regulate tumor development, we focused on interactions between tumor cells and gradually increasing or decreasing levels of fibroblasts (*POSTN*+ fibroblasts, *RSPO1*+ fibroblasts) and macrophages (*SPP1*+ macrophages, *FOLR2*+ macrophages). First, the CellChat algorithm[24] was used to calculate the interaction weights among *POSTN*+ fibroblasts, *SPP1*+ macrophages, *RSPO1*+ fibroblasts, *FOLR2*+ macrophages, and tumor cells (Fig. 4A). We found that the interaction of *POSTN*+ fibroblasts with *SPP1*+ macrophages, *POSTN*+ fibroblasts with tumor cells, and *SPP1*+ macrophages with tumor cells gradually increased from NT to Pre, E and A stages, which was also observed in separate analyses in P2, P10, and P13 (Supplementary Fig. 4a). However, no trend toward a gradual increase in the interaction density was observed between *POSTN*+ fibroblasts and *FOLR2*+ macrophages or between *SPP1*+ macrophages and *RSPO1*+ fibroblasts, indicating the specificity of the cellular interactions among *POSTN*+ fibroblasts, *SPP1*+ macrophages and tumor cells during the multistep progression of HNSCC. We also validated the strongest interaction between *POSTN*+ fibroblasts and *SPP1*+ macrophages in single A-stage samples from the in-house cohort and public datasets (GSE188737[25], GSE182227[26], and GSE234933[27]) (Supplementary Fig. 4b, c). Furthermore, the percentage of closely related of *POSTN*-positive and *SPP1*-positive cells increased significantly from the E stage to the A stage, supporting the gradual increase in the interaction between *POSTN*+ fibroblasts and *SPP1*+ macrophages (Fig. 4B). Based on these results, we further focused on *POSTN*+ fibroblasts and *SPP1*+ macrophages and investigated how they interact with each other and how their cellular communication regulates tumor cells.

Because the infiltration and interaction of *POSTN*+ fibroblasts and *SPP1*+ macrophages were greatest in the A stage, we then analyzed the ligand–receptor (LR) interaction between these two cell types and tumor cells. The A stage-specific ligands from *POSTN*+ fibroblasts to *SPP1*+ macrophages included *EGF, FGF, HGF, GDNF, PDGF* and *NRG*; the ligands from *SPP1*+ macrophages to *POSTN*+ fibroblasts comprised *CD23, FGF, NGF, FLT3, EPHB* and complement (Fig. 4C). Considering that the interaction weights between these two tumor-promoting cells increase during HNSCC progression, we speculated that such cell–cell communication would be conducive to tumor development. Therefore, we constructed an interaction signature by selecting overlapping ligands and found that patients with high *POSTN*+ fibroblasts and high *SPP1*+ macrophages expression had significantly poorer OS, consistent with our hypothesis (Fig. 4D, Supplementary Fig. 4d).

To explore the mediators and downstream targets of the *POSTN*+ fibroblasts and *SPP1*+ macrophages interaction, we performed Nichenet analysis[28] and found *MDK* and *FGF* to be specifically expressed in A-stage *POSTN*+ fibroblasts with high ligand activity. In addition, the ligand encoded by *MDK* bound to the receptor encoded by *NUPR1* and the ligand *FGF* bound to the receptor *SPP1* on *SPP1*+ macrophages, resulting in expression of target genes involved in cell metabolic processes such as ATP metabolic, glycolytic, and NAD metabolic processes, thus promoting fibroblast proliferation. Regarding the interaction of *SPP1*+ macrophages with *POSTN*+ fibroblasts, ligands encoded by *FN1, ILRN*, and *MMP9* presented the highest ligand activity in the A stage. *FN1* and *ILRN1* bind to *MMP1* and *MMP13*, respectively, and *MMP9* bind to *ACTA2* on *POSTN*+ fibroblasts, triggering downstream pathways, including those related to ECM organization, cell-substrate adhesion and focal adhesion (Fig. 4E–H, Supplementary Fig. 4e–g); these processes may contribute to the formation of a desmoplastic microenvironment, as reported in a previous study[29,30]. Moreover, the ligand-receptor interaction between *POSTN*+ fibroblasts and *SPP1*+ macrophages and downstream regulatory effects were well validated in other HNSCC datasets (Supplementary Fig. 4h–k). Taken together,

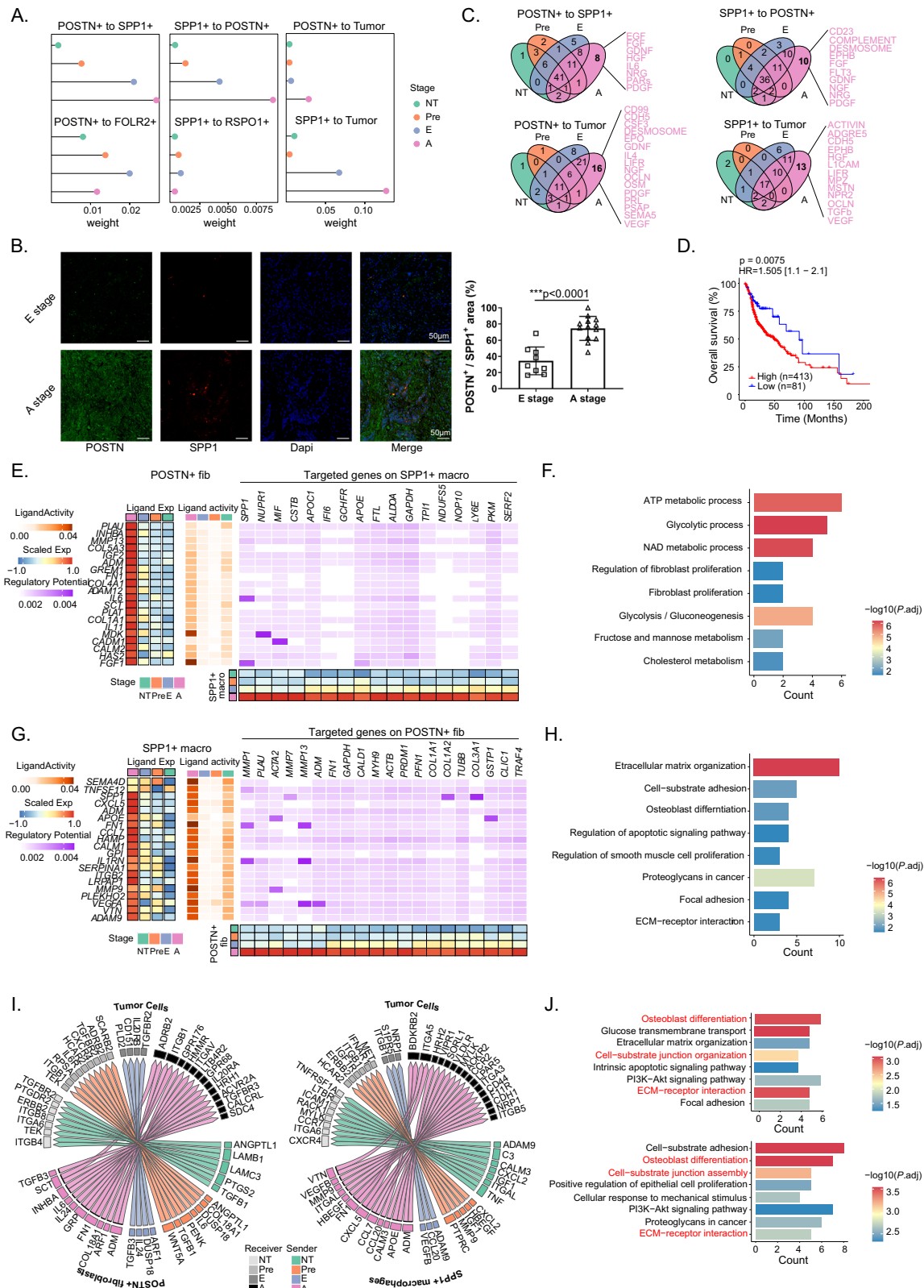

these results suggest that in the A stage, *POSTN*[+] fibroblasts regulate the metabolic characteristics of *SPP1*[+] macrophages and that *SPP1*[+] macrophages regulate the ECM remodeling function of *POSTN*[+] fibroblasts, potentially promoting desmoplastic structure formation and abnormal metabolism in the TME. We consolidate this result in the TCGA-HNSCC cohort, finding that among the immune cell types, CD8[+] T cell infiltration was significantly less abundant in *POSTN*[+] fibroblast

high group, which was further validated by mIHC assay in the validation cohort (Supplementary Fig. 4l, m).

In addition to the interaction between *POSTN*[+] fibroblasts and *SPP1*[+] macrophages, we were interested in how they influence tumor cells because we found that their interaction weight with tumor cells increased from the NT to A stage (Fig. 4A). *HMGB2* in *POSTN*[+] fibroblasts is predicted to bind to *PLD2, LRP5, MYLK,* and *CD44* on tumor

**Fig. 4 | Characterization of cell-cell interactions of *POSTN*⁺ fibroblasts, *SPP1*⁺ macrophages and tumor cells during HNSCC progression. A** Comparison of interaction strength between different cells in NT, pre, E, and A stage. **B** Immunofluerence results show proportion of *POSTN*⁺ fibroblasts colocalized with *SPP1*⁺ macrophages in A stage samples (*n* = 9 for E stage, *n* = 13 for A stage). Scale bar = 50 μm. **C** Venn diagrams show A stage-specific ligands between different cells. **D** The Kaplan-Meier curve shows patients with higher interaction signature exhibit poorer OS in TCGA-HNSCC cohort (*n* = 494, 413 samples for high- and 81 samples for low-group). **E, G** Heatmaps of Nichenet analysis show regulatory patterns between *POSTN*⁺ fibroblasts to *SPP1*⁺ macrophages (E) and *SPP1*⁺ macrophages to

*POSTN*⁺ fibroblasts (**G**). Representative GO and KEGG pathways enrichment of the predicted target genes expressed in *SPP1*⁺ macrophages (**F**) and *POSTN*⁺ fibroblasts (**H**). **I** Circus plots show ligand and receptor pairs from *POSTN*⁺ fibroblasts (left) and *SPP1*⁺ macrophages (right) to tumor cells. **J** Representative GO and KEGG pathways enrichment of the target genes expressed in tumor cells. Ligands are from *POSTN*⁺ fibroblasts (above) and *SPP1*⁺ macrophages (below), respectively. Data represent mean ± SD. *P* values were calculated by two-side Student's *t*-test in **B**, by two-sided Wilcoxon signed-rank test in **F**, **H**, and **J**, and by two-sided log-rank test in **D**. Source data are provided as a Source Data Fig. 4A-J.

cells, regulating downstream genes, including *BIRC5, CCNA2, CCNB1, CCNB2, CDC20*, and *MKI67*, which are related to the cell cycle. INHBA on *SPP1*⁺ macrophages is predicted to interact with *ACVR1* on tumor cells, targeting downstream genes, including *SERPINE1* and *SMAD3*, which positively regulate cyclin-dependent protein kinase activity (Supplementary Fig. 4n–s). In the A stage, *POSTN*⁺ fibroblasts and *SPP1*⁺ macrophages represented specific ligand–receptor pairs (Fig. 4I). For example, *TGFB3-SDC4* and *INHBA-TGFBR3* were found between *POSTN*⁺ fibroblasts and tumor cells and *CXCL5-BDKRB2* and *CCL7-ACKR2* were found between *SPP1*⁺ macrophages and tumor cells. The overlap of target gene enriched pathways included cell-substrate junction assembly, ECM-receptor interaction, and osteoblast differentiation, suggesting the combined oncogenic effects of *POSTN*⁺ fibroblasts and *SPP1*⁺ macrophages on tumor cells (Fig. 4J). Interestingly, we observed a positive correlation between the percentage of cluster1 malignant epithelial cells and *POSTN* expression in *POSTN*⁺ fibroblasts, which supports the tumor-promoting effects of *POSTN*⁺ fibroblasts (Supplementary Fig. 4t).

## Reprogramming of *CXCL13*⁺ CD8⁺ Tex cells to tumor cells contributes to extranodal extension of lymph node

Extranodal extension (ENE) is associated with the poorest outcomes in HNSCC patients[9]. To determine the underlying mechanism of ENE, we analyzed the intracapsular (ENE⁻) (LN-in) and extracapsular (ENE⁺) (LN-out) metastatic lymph node samples from our cohort by comparing DEGs between malignant epithelial cells in these two groups and found that genes related to tumor invasion and metastasis (*e.g., MT2A, SAA1, STATH*, and *CST1*) were highly expressed in the LN-out group (Fig. 5A)[9,31]. Immune response-related pathways (response to interferon-γ, antigen processing and presentation of peptide antigen, response to type I interferon) were enriched in the LN-out group compared to the LN-in group (Fig. 5B), suggesting that malignant cells may reshape the immune microenvironment, facilitating their metastasis out of the lymph node capsule. Therefore, we computed the interaction events of malignant cells with surrounding immune cells as well as stromal cells in LN-out, LN-in, and LN-normal specimens (Fig. 5C). Consistent with our speculation, malignant cells displayed higher levels of inferred interplay with CD4⁺ T, CD8⁺ T and myeloid cells in the LN-out group than in the LN-in or LN-normal group.

As an important constituent of CD4⁺ T cells, Tregs have been reported to promote tumor progression by shaping the immunosuppressive microenvironment[32], and their infiltration frequency was substantially increased in the LN-out group, which was validated in lymph node sections (Supplementary Fig. 5a–c). Moreover, circle plots illustrated that the likelihood of interaction between tumor cells and Tregs was significantly greater in LN-out samples (Supplementary Fig. 5d, e). With respect to CD8⁺ T cells, the immunosuppressive CD8 Tex cells also exhibited a prominently proportional increase in the LN-out group compared to the LN-in group (Fig. 5D-E). The DEGs of CD8 Tex cells in the LN-out group were functionally enriched in the interferon-γ-mediated signaling pathway, response to type I interferon, and other immune-related pathways (Supplementary Fig. 5f). Further CellChat analysis revealed that the interaction intensity of

tumor cells with CD8 Tex cells and that of CD8 Tex cells with tumor cells was much greater in the LN-out group than in the LN-in group (Fig. 5F). Inspired by these results, we investigated elucidate the functional significance of the ligand–receptor interaction as well as downstream signaling between tumor cells and CD8 Tex cells during extracapsular lymph node metastasis. Significant differences in the ligand–receptor expression of immune checkpoints between LN-in and LN-out malignant cells and CD8 Tex cells were observed (Supplementary Fig. 5g, h). The expression of ligands, including *TNFRSF14, CD274*, and *LGALS9*, on tumor cells increased in the LN-out group. In addition, the expression of receptors, including *TNFSF14, CD40, LTBR*, and *VTCN1*, was elevated in LN-out tumor cells. In other words, the ligand–receptor pairs *CD40LG-CD40, TNFSF14-TNFRSF14, TNFSF14-LTBR, BTLA-TNFRSF14, TNFRSF14-BTLA, CD274-PDCD1*, and *LGALS9-HAVCR2* were upregulated during the ENE process, shaping the immunosuppressive microenvironment for metastasis. Subsequent pathway analysis indicated that, influenced by tumor cells, the receptors of CD8 Tex cells in LN-out samples are involved in the response to interferon-gamma, antigen processing and presentation, and regulation of myeloid cell differentiation, suggesting that the cell–cell interaction of tumor cells with CD8 Tex cells favors the ENE by modulating the immunosuppressive efficacy of CD8 Tex cells and even the immune microenvironment (Supplementary Fig. 5i). In turn, CD8 Tex cells in the LN-out highly expressed *TGF-β1, IFN-γ*, and *ITGB1*, which target *FN1, EGFR, CTNNB1, COL1A1*, and *CFLAR* in tumor cells, activating downstream pathways including *ERK1* and *ERK2* cascade (Fig. 5G). We validated this in lymph node metastasis and found that the pERK⁺ malignant epithelial cell ratio was greater in ENE⁺ samples. (Supplementary Fig. 5j). This study uncovers the underlying mechanism of CD8 Tex cell-mediated tumor cell reprogramming during the ENE process.

Recent studies have identified a CD8 Tex cell subtype, *CXCL13*⁺ Tex cells[32], which represent the terminal exhaustion status of the CD8⁺ T differentiation trajectory. Interestingly, when comparing the DEGs of CD8 Tex cells between the LN-in and LN-out groups, we found that *CXCL13* was highly expressed in LN-out CD8 Tex cells, along with another immune checkpoint inhibitor, *LAG3*, suggesting a more dysfunctional phenotype of CD8⁺ T cells in lymph nodes with ENE (Fig. 5H, I). A feature plot also validated the presence of this subtype among CD8 Tex cells (Supplementary Fig. 5k). Then, we performed multiplex immunohistochemistry (mIHC) analysis for ENE⁺ and ENE⁻ metastatic lymph nodes. In the ENE⁺ lymph nodes, the infiltrating fraction of *CXCL13*⁺ Tex (CXCL13⁺ PD1⁺ CD8⁺ cells) was significantly greater than that in the ENE⁻ lymph nodes. Furthermore, the number of *CXCL13*⁺ T cells was positively correlated with pERK⁺ tumor cell ratio (Supplementary Fig. 5j), revealing the role of *CXCL13*⁺ Tex cells in activating ERK signaling and shaping the prometastatic niche (Fig. 5J).

In summary, we dissected the microenvironment between intracapsular and extracapsular metastatic lymph nodes and revealed the regulatory role of Tex cells among malignant cells during ENE. Moreover, the high level of infiltration of *CXCL13*⁺ Tex cells in extracapsular metastatic lymph nodes may constitute a potential therapeutic target in HNSCC.

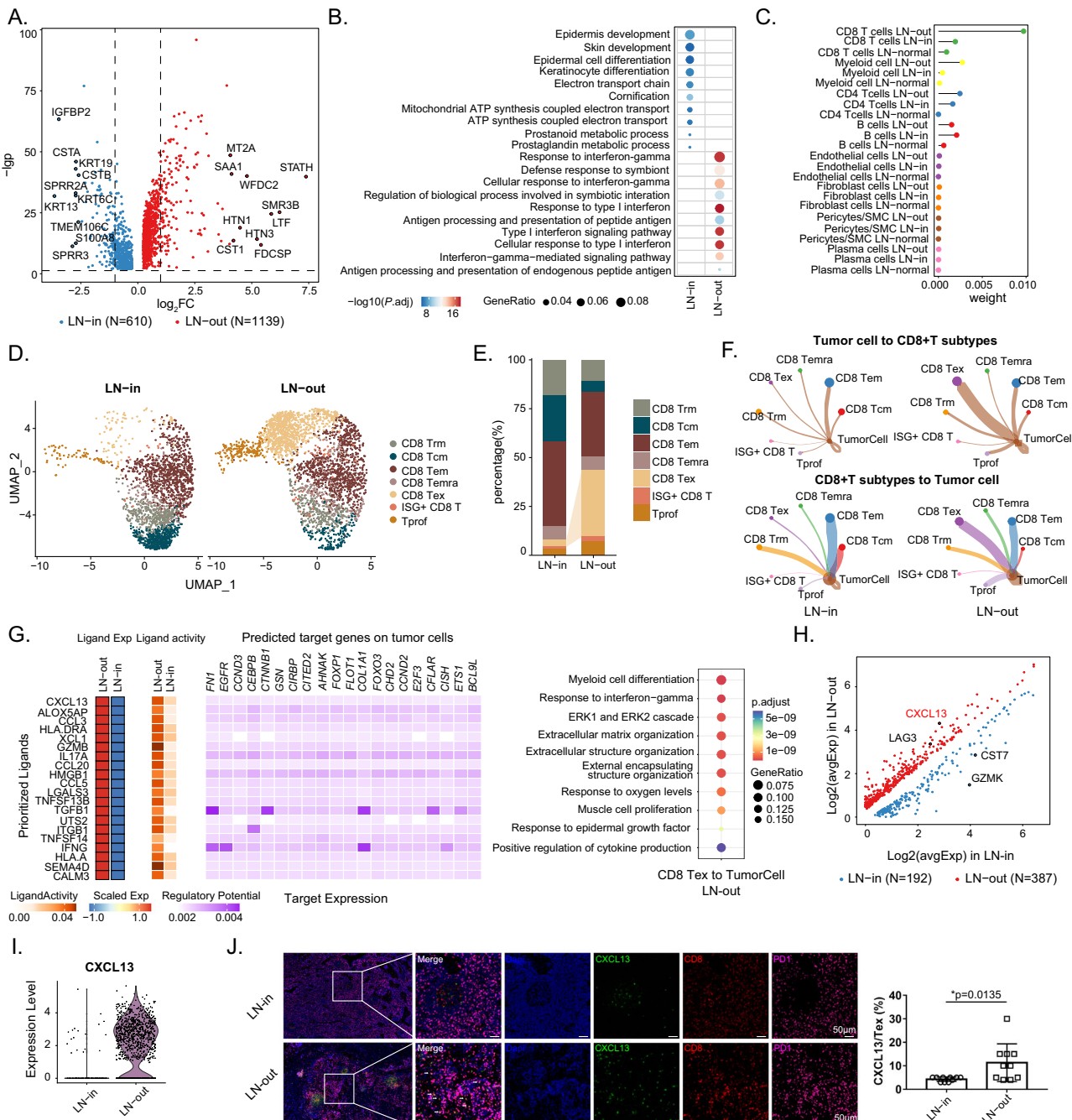

**Fig. 5 | Tumor cell reprogramming by exhausted CD8⁺ T during extracapsular metastasis. A** Volcano plot shows DEGs in malignant epithelial cells in LN-out (1139 genes up-regulated) and LN-in (610 genes up-regulated) samples. **B** Bubble chart shows the enrichment of specific pathways, based on the HALLMARK gene set of upregulated genes, in LN-out and LN-in malignant epithelial cells. **C** Comparison of interaction strength between different cells in LN-out, LN-in, and LN-normal samples. **D** UMAP plot shows 2079 CD8⁺ T cells from 3 LN-in samples and 2878 CD8⁺ T cells from 2 LN-out samples. **E** Bar plot shows the proportion differences of CD8⁺ T cell subclusters between LN-out and LN-in samples. **F** Circle plots show cell-cell interaction strength differences of tumor cells and CD8⁺ T cell subclusters in LN-out and LN-in samples. **G** Heatmap of Nichenet analysis shows regulatory patterns of CD8 Tex cells to tumor cells. Representative GO and KEGG pathways enrichment of the predicted target genes expressed in tumor cells are exhibited in the right. Volcano plot (**H**) and violin plot (**I**) show DEGs in CD8 Tex in LN-out (387 genes up-regulated) and LN-in (192 genes up-regulated) samples. **J** Representative mIHC staining of lymph node samples. Dapi (blue), CXCL13 (green), CD8 (red), PD1 (purple), in individual and merged channels are shown. Scale bar = 50 μm. Proportion of *CXCL13⁺* Tex is compared between LN-in and LN-out samples and is shown in the right (*n* = 10 for each group). Data represent mean ± SD. *P* values were calculated by two-side Student's *t*-test in **J**, and by two-sided Wilcoxon signed-rank test in **A**, **B**, **G**, and **H**. Source data are provided as a Source Data Fig. 5A-J.

## Distinct phenotypes of malignant epithelial cells favor HNSCC recurrence

Finally, we probed the tumor niche atlas in recurrent (R)-stage and primary (A)-stage tumors. To determine differences in tumor cells between R and A stage patients, we calculated the CNV score by the inferCNV algorithm and found that malignant cells exhibited a significantly greater CNV level in R stage than in A stage patients, suggesting genetic evolution during the recurrence process (Fig. 6A, Supplementary Fig. 6a)[33]. Subsequently, DEG analysis was conducted between A and R stage malignant cells. We found that genes with copy

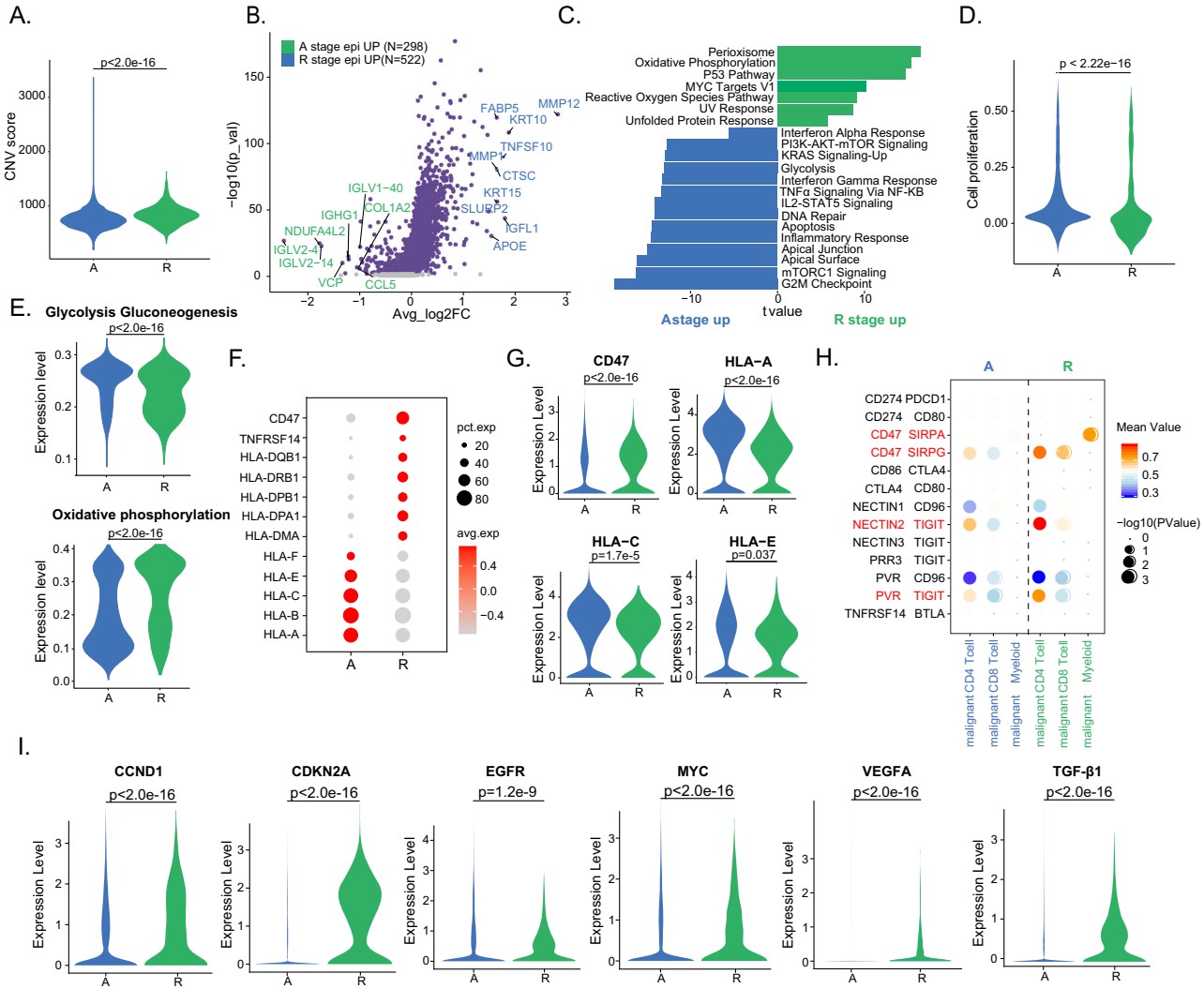

**Fig. 6 | Characteristics of malignant epithelial cells in recurrent tumors. A** Violin plot shows CNV scores in malignant epithelial cells in **A** (n = 3952) and R (n = 875) samples. **B** Volcano plot shows DEGs between A (298 genes up-regulated) and R stage (552 genes up-regulated) malignant cells. The names of the most significant genes are labeled in the plots. **C** Bar chart shows the enrichment of specific pathways, based on the HALLMARK gene set of upregulated genes, in A or R stage malignant cells. **D** Violin plot shows cell proliferation score in malignant epithelial cells in A and R samples. **E** Violin plots show the glycolysis gluconeogenesis and oxidative phosphorylation scores of malignant cells in A and R stage. Bubble plots (**F**) and violin plots (**G**) show the expression of selected genes related to immune surveillance in malignant cells in A and R stage. **H** Bubble plots show the interaction between malignant cells and myeloid cells, CD4⁺, and CD8⁺ T cells, based on selected ligand and receptor pairs. These scores are normalized expression level, and the sizes of the bubbles indicate the significance of the interactions, calculated by CellPhoneDB. **I** Violin plots show the expression of selected genes in malignant cells in A and R stage. Data represent mean ± SD. *P* values were calculated by two-sided Wilcoxon signed-rank test in **A**, **B**, **D**, **E**, **G**, and **I**, and was calculated based on the interaction and the normalized cell matrix achieved by Seurat Normalization in **H**. Source data are provided as a Source Data Fig. 6A–I.

number gains in the R stage intersected with those upregulated in R stage based on the scRNA-seq results (Supplementary Fig. 6b). These results revealed that the top-ranked genes upregulated in malignant cells in the R stage were *MMP12*, *MMP1*, and *CTSC*, which are associated with tumor invasion and metastasis (Fig. 6B)[34,35]. The genes upregulated in A-stage malignant cells were enriched in immune-response-related pathways (*e.g.*, interferon alpha response, interferon gamma response, *IL2-STAT5* signaling, and inflammatory response), whereas the genes involved in stress-response pathways (*e.g.*, reactive oxygen species pathway, UV response, and unfolded protein response) were upregulated in R-stage tumors (Fig. 6C). Interestingly, malignant cells in the R stage were observed to have a low cell proliferation score[36], suggesting that tumor cells in the R stage have lower proliferative function signals (Fig. 6D). The metabolic patterns of malignant cells were adapted to their proliferation mode and functional transition. Unlike cancer cells in primary tumors, recurrent malignant cells exhibit

a decrease in glycolysis metabolism but an increase in oxidative phosphorylation (Fig. 6E, Supplementary Fig. 6c). We also observed similar trends in public scRNA-seq[27] and bulk RNA-seq datasets[37] (Supplementary Fig. 6d, e). These results suggest that malignant epithelial cells in recurrent tumors proliferate relatively poorly and have greater metabolic activity than malignant epithelial cells in primary tumors.

Next, we examined the expression of immune checkpoint ligands and receptors (LRs). Among the genes related to immune surveillance, differences in antigen presentation-related gene expression were detected. We observed increased *CD47, HLA-DQB1, HLA-DRB1, HLA-DPA1*, and *HLA-DM1* expression but decreased enrichment of *HLA-A, HLA-B, HLA-C, HLA-E*, and *HLA-F* in R stage malignant cells (Fig. 6F, G), suggesting that malignant cells exert different functions regarding immune activation and suppression between primary and recurrent tumors. In primary lesions, tumor cells express MHC I molecules and

activate the antitumor immunity of CD8[+] T cells. However, in recurrent lesions, tumor cells lose MHC I molecule expression, which may promote immune evasion in HNSCC patients during relapse[38]. Furthermore, many LR pairs, such as *CD274-PDCD1* and *CD86-CTLA4*, were expressed at low levels in both A- and R-stage malignant cells. In contrast, *CD47-SIRPG* was significantly upregulated in malignant cells from the R stage with CD4[+] and CD8[+] T cells, which was validated in the public scRNA-seq dataset (Fig. 6H, Supplementary Fig. 6f). Therefore, we postulate that recurrent malignant cells may inhibit the antitumor immunity of T cells and augment the protumor immune response of myeloid cells through these interaction molecular pairs, rather than through classic inhibitory signals.

To date, several targeted genes and corresponding drugs have been proposed as clinical therapies for advanced HNSCC patients, but these treatments are limited by discrepancies in outcomes between primary and recurrent tumors. This inspired us to rethink therapeutic selection regarding which targeted therapy is more suitable for primary tumors and the kind of targeted drugs that may benefit R-stage patients, as these targets become amplified at different levels in patients with primary and recurrent tumors. Accordingly, we compared the expression of target genes (common in HNSCC) among these two groups and found that the expression levels of *CDKN2A*, *EGFR*, *VEGFA*, and *TGF-β1* were greater in the R-stage samples, suggesting that these patients may benefit less from CDK4/6 inhibitor intervention but may achieve better clinical outcomes from the inhibition of *EGFR, VEGFR* and *TGF-β1* (Fig. 6I, Supplementary Fig. 6g). This information provides a theoretical foundation for the precise selection of targeted therapy for primary and recurrent tumors. In conclusion, our analyses together highlight the different characteristics of malignant epithelial cells in primary and recurrent tumors.

## Discussion

With the application of the emerging technology scRNA-seq to delineate intratumoral heterogeneity, the complex cellular ecosystem of HNSCC has been comprehensively analyzed with regarding immune and nonimmune compartments. Key findings of such studies include identification of the *p*-EMT program in malignant cells in vivo and its association with metastasis, discovery of germinal center B cells and divergent myeloid states in HPV[+] TILs, and knowledge of the dominant contributing role of tumor-associated macrophages in the expression of *PD-L1* and other immune checkpoint ligands[1,5,6]. However, the TME profiles involved in HNSCC progression, especially from normal tissue to premalignant, early malignant, advanced malignant, recurrent lesions, and even lymph node metastasis, have not been totally elucidated. Choi *et. al* delineated the heterogeneity of tumor cells, CAFs, and Treg cells during the progression of HNSCC from normal tissues to leukoplakia, primary cancer, and lymph node metastasis. They have identified CAF-derived COL1A1 interacts with CD44 in malignant cells, revealing the promotive role of *CXCL8*-expressing CAFs and *LAIR2* expression level in Treg cells during HNSCC progression[8]. In our current study, through the integrated analyses of scRNA-seq data of our in-house cohort, scRNA-seq and bulk RNA-seq data of public accessible cohorts, and IF, IHC, and mIHC data, we elucidated the comprehensive landscape of the TME in stepwise HNSCC progression from normal tissues to precancerous, early malignant, advanced malignant, recurrent lesions, and lymph node metastasis at a single-cell resolution. We mainly focused on the tumor cells, CAFs, macrophages, and CD8[+] T cells and obtained several original discoveries that were firstly demonstrated in HNSCC ecosystem: Firstly, fibroblasts from our samples were stratified in 8 clusters and 2 critical subpopulations (*POSTN*[+] and *RSPO1*[+] fibroblasts) showed distinct changes of both infiltrative proportions and biological functions during HNSCC stepwise progression. Secondly, we classified macrophages into 4 subtypes and delineated the different proportional and functions changes between *SPP1*[+] and *FOLR2*[+] macrophages during the development of

HNSCC. The third and the most important discovery was the reprogramming role of cellular interaction between *SPP1*[+] macrophages and *POSTN*[+] fibroblasts and tumor cells, thus reshaping the desmoplastic TME. Fourthly, we also proposed that *CXCL13*[+] Tex contributed to ENE through reshaping the malignant cells. Fifthly, we elucidated the TME heterogeneity of primary and recurrent tumors, uncovered the underlying mechanism of malignant cell-mediated tumor relapse, which may contribute to further development of precise therapeutics for patients with primary and recurrent tumors. This is an investigation to describe in detail a single-cell atlas of immune/nonimmune compartments to determine the possible progression trajectories and transition fates of HNSCC cells at whole clinical stages, including tumor initiation, progression, recurrence and metastasis.

Our key finding is the enhanced infiltration of *POSTN*[+] fibroblasts and *SPP1*[+] macrophages and the occurrence cell-cell interactions between these two cell types in multiple processes during HNSCC development. Among all cell types in the tumor stroma, fibroblasts, especially CAFs, are the predominant mesenchyme-derived stromal component in the TME and dynamically evolve along the tumor. Accumulating evidence indicates that CAFs are highly heterogeneous and composed of a dynamic collection of subclusters with distinct phenotypes and biological functions that participate in tumor progression. Hence, CAFs may be potential therapeutic targets for cancer treatment[39–41]. The matricellular protein periostin, which is encoded by *POSTN*, and uniquely expressed in CAFs but rarely detected in normal tissues, facilitates tumor cell adhesion and migration, contributes to the formation of cancer stem cells and premetastatic niches, and supports TME remodeling and tumorigenesis[42–44]. However, fluctuating changes in the infiltration ratio and functional phenotype of *POSTN*[+] fibroblasts during HNSCC progression have not been determined. We identified that the abundance of *POSTN*[+] fibroblasts increase during the transition from normal tissues to premalignant lesions and from early to advanced and recurrent tumors. High levels of *POSTN*[+] fibroblasts are positively associated with poor clinical outcome in HNSCC patients in the TCGA database, in accordance with findings for gastric cancer[45]. In contrast, the infiltration level and predictive role of *RSPO1*[+] fibroblasts in predicting patient prognosis showed a trend opposite to that of *POSTN*[+] fibroblasts. This discovery suggested that *POSTN*[+] and *RSPO1*[+] fibroblasts perform distinct biological functions, as evidenced by the greater enrichment of pathways related to ECM organization as well as antigen processing and presentation in *POSTN*[+] fibroblasts than in *RSPO1*[+] fibroblasts.

Macrophages are critical regulators of the TME and are involved in multiple aspects of tumor immunity[18]. In our study, four macrophage subsets were identified in primary tumor and metastatic specimens: *SPP1*[+] macrophages, *C1QC*[+] macrophages, *CXCL10*[+] macrophages and *FOLR2*[+] macrophages. Among these cells, *FOLR2*[+] macrophages, which are tissue-resident macrophages, were preferentially enriched in normal tissues and exhibited decreased infiltration during HNSCC carcinogenesis and progression. These tissue-resident *FOLR2*[+] macrophages were previously reported to exhibit fetal-liver macrophage features and participate in onco-fetal reprogramming of the ecosystem in hepatic cellular cancer[46]. In contrast, *SPP1*[+] macrophages, which exhibit higher M2 signatures and express *SPP1*, resemble a cluster of angiogenesis- and ECM reorganization-associated macrophages, as indicated by functional phenotype analysis of angiogenic and phagocytic signatures[21,47,48]. Interestingly, the *SPP1*[+] macrophages identified in our samples also showed preferential expression of gene sets involved in neutrophil activation and degranulation. A higher expression of *SPP1* correlates intimately with neutrophil extracellular trap (NET) formation, and *SPP1* influences NET-induced malignant capacity, indicating that the underlying mechanism of the regulatory effect of *SPP1* on neutrophils is a promising research direction[49]. Like those in *POSTN*[+] fibroblasts, the number of *SPP1*[+] macrophages in tumor tissues

were dramatically increased, and higher infiltration correlated with a shorter overall survival in TCGA HNSCC patients.

According to our single-cell data, the interaction counts between CAFs and macrophages were greater than those between CAFs and other cell subpopulations, highlighting the potential role of these cells in regulating and remodeling the TME. The macrophage-fibroblast network shapes an immunosuppressed TME lacking antitumor CD8[+] T cells. Immune surveillance escape is the most important hallmark of cancer, and CAFs can facilitate this process not only by forming a physical barrier but also by influencing the immune TME, such as tumor-promoting macrophages[48]. Tumor-specific *FAP*[+] fibroblasts and *SPP1*[+] macrophages are positively associated with published data from colorectal cancer cohorts, and this interaction promotes the formation of immune-excluded desmoplastic structures and the exclusion of T-cell infiltration, limiting the therapeutic benefit of immune therapy[29]. According to pancancer single-cell analysis by Luo *et al.*, the crosstalk between CAFs and proximal *SPP1*[+] macrophages participate in the endothelial-to-mesenchymal transition and contributes to survival stratifications[50]. However, they did not distinguish which subpopulation of CAFs strongly interacted with *SPP1*[+] macrophages, nor did they take tumor stage into consideration. Several studies have investigated *POSTN*[+] fibroblasts and *SPP1*[+] macrophages in HNSCC; however, these studies reported only the correlation of individual gene expression and survival and lack deep insight into their interactions and effects on the TME[51,52]. Notably, this crosstalk was investigated from multiple aspects in the current study, including single-cell transcriptomics, immuno-fluorescent labeling of clinical specimens, and bioinformatics analysis of published datasets. An interesting discovery is that specific reciprocal communication occurs between *POSTN*[+] fibroblasts and *SPP1*[+] macrophages, with an upward trend in the interaction strength during tumor progression. This finding was also verified in the HNSCC cohort from TCGA and other scRNA-seq datasets, in which *SPP1*[+] macrophages were the most relevant cells that interact with *POSTN*[+] fibroblasts; the interaction intensity was positively associated with poor overall survival[29,53]. Spatial proximity between macrophages and CAFs enables paracrine interactions through the ligand–receptor pattern in CRC, and this intercellular communication influences the transcriptional phenotype of both cell types[48]. ECM remodeling is indispensable for the formation of desmoplastic regions, and *SPP1*[+] macrophages facilitate ECM organization and cell–substrate adhesion in *POSTN*[+] fibroblasts by secreting cytokines encoded by *MMP1*, *MMP13*, *COL1A1* and *COL3A1*; these findings suggest that the desmoplastic microenvironment of HNSCC is governed by *SPP1*[+] macrophages and *POSTN*[+] fibroblasts. Conversely, *POSTN*[+] fibroblasts influence *SPP1*[+] macrophages via enhanced ATP/NAD metabolism and glycolytic processes. Macrophage metabolism plays a determining role in their functional phenotype, and alterations in lipid metabolism are associated with poor prognosis in CRC[54,55]. Taken together, these findings suggest that the metabolic remodeling role of *POSTN*[+] fibroblasts with respect to *SPP1*[+] macrophages may contribute to HNSCC development. Furthermore, the interaction between *POSTN*[+] fibroblasts and *SPP1*[+] macrophages contribute to ECM remodeling and coordinates to form a desmoplastic microenvironment by enhancing tumor cell ECM-receptor interactions, and cell-substrate junction organization. Our study highlights the potential value of identifying and establishing therapeutic strategies targeting *POSTN*[+] fibroblasts, *SPP1*[+] macrophages, or the molecules involved in their crosstalk to inhibit HNSCC progression.

In general, the spread of cancer cells from a primary tumor to a locoregional lymph node is an important indication of HNSCC progression and a predictor of survival in patients with epithelial carcinoma[56]. The presence of extranodal extension (ENE[+]) in tumor-draining lymph nodes contributes to stage classification according to American Joint Committee on Cancer (AJCC). For example, HNSCC patients who are ENE[+] may be classified as N3b, the most advanced stage for locoregional metastasis, regardless of the status of the primary tumor, and this could be an indication for adjuvant chemotherapy or radiotherapy. Hence, the ENE is one of the most pivotal parameters for assessing tumor progression. However, the mechanisms by which some tumor cells detach from the primary lesion to colonize distant sites have not been fully elucidated. In the present investigation, we focused on the underlying mechanism of ENE by dissecting the microenvironment of normal, intracapsular, and extracapsular metastatic lymph node tissues. The most prominent finding was the obviously greater fraction of exhausted CD8[+] T cells in ENE[+] lymph nodes than in normal and ENE[-] lymph nodes. An identical trend was found for cell–cell communication between exhausted CD8[+] T cells and tumor cells, reflecting the presence of a metastatic niche driving effector T cells toward exhaustion and an immunosuppressive status. For example, tumor-derived *IL-8* upregulates *PD-1* expression in CD8[+] T cells, promoting lymph node metastasis. NETs contain the immunosuppressive ligand *PD-L1*, which is responsible for T-cell exhaustion and dysfunction. Breast cancer cells transfer *TGF-β* type II receptors through extracellular vesicles to induce CD8[+] T-cell exhaustion via the *TGF-β* signaling pathway[57–60]. However, the mechanism by which CD8 Tex cells influence malignant cells and promote LN metastasis has not been elucidated. Based on our data, CD8 Tex cells in LN-out upregulate of *TGF-β1, IFN-γ*, and *ITG-β1*, which subsequently target *FN1, EGFR, CTNNB1*, and *COL1A1* in tumor cells, activating the *ERK1* and *ERK2* cascade pathways and facilitating tumor progression. This study has interpreted the underlying mechanism of CD8 Tex cell-mediated tumor cell reprogramming during lymph node metastasis, especially for the ENE process.

*CXCL13* was found to be preferentially enriched in dysfunctional CD8[+] T cells within the melanoma ecosystem, and together with *TIGIT*, *PDCD1* and *LAG3*, it has been used to define the dysfunctional state of T cells[61]. A subcluster of terminal Tex cells from ovarian cancer express *FOXP3*, a dominant transcription factor in Treg cells, and these *CD8*[+] *FOXP3*[+] T cells shared TCRs with *CXCL13*[+] cells. In addition, the RNA velocity of these terminal Tex cells points to that of *CXCL13*[+] cells, suggesting a more terminal exhaustion state of *CXCL13*[+] CD8 Tex cells[62]. In our study, *CXCL13*[+] CD8 Tex cells were highly infiltrated in LN-out samples, as verified by mIHC analysis of LN-in and LN-out samples. This special subset of CD8 Tex cells may be a pivotal contributor to lymph node metastasis, especially for ENE patients.

The main limitation of our study is the combined trajectory plots and analysis of tumor cells. Although an increasing number of studies have combined tumor cells from different patients to study different states and potential trajectory characteristics of tumor cells[6,63–67], these tumor cells came from patients with distinct genetic entities, which have not been revealed in the current study. It may be more reasonable and acceptable to perform longitudinal analysis of tumor cells individually and add genetic sequencing data into the context. In our future study, we would enlarge the sample number and include multi-omics data to generate a more comprehensive landscape.

In conclusion, the current study provides insight into the progression of HNSCC stepwise progression in a niche atlas. We explored the dynamic alterations in the infiltration proportions and biological functions of malignant cells, immune cells and stromal cells, contributing to a comprehensive understanding of the HNSCC ecosystem during tumor initiation, development, recurrence and lymph node metastasis. Our study describes strategies for molecular intervention involving the cellular interaction of stromal cells (*POSTN*[+] fibroblasts) with immune components (*SPP1*[+] macrophages), and suggests treatment decision-making for primary and recurrent tumors, which might ultimately improve survival in HNSCC patients.

## Methods
### Clinical cohort information
The detailed demographic and clinical information of the patients in the scRNA-seq cohort are displayed in Supplementary Data 1. The

information of the patients in validation Cohort1 and Cohort 2 is displayed in Supplementary Data 2 and Supplementary Data 3. Gender analysis was not performed because we focused on the differences between cancer stages instead of genders.

## Clinical sample collection

Adjacent normal mucosa, precancerous lesions, tumor tissues, and lymph nodes were collected from HNSCC patients with informed written consent and under the approval of the local medical ethics committee of Shanghai Ninth People's Hospital Affiliated with Shanghai Jiao Tong University. Fresh tissues were stored in RPMI 1640 containing 10% FBS on ice for transport.

## Tissue dissociation

Fat tissues and visible blood vessels were removed before tissue processing. Fresh oral mucosa, HNSCC tissues, and lymph nodes were washed with ice-cold PBS and cut into small pieces. For the oral mucosa and lymph nodes, the tissues were placed in 10 mL of EDTA-containing buffer (5 mM EDTA, 15 mM HEPES, 1 mM DTT, and 10% FBS-supplemented PBS) and shaken for 1 h at 37 °C. The tissues were incubated with 10 mL of DTT (65 mM)-containing PBS (supplemented with 10% FBS) for 15 min at 37 °C with shaking. EDTA and DTT were then removed by washing with PBS twice. The small tissue pieces were minced and digested with 0.38 mg/mL collagenase VIII and 0.1 mg/mL DNase I in complete RPMI 1640 medium (containing 10% FBS, 100 U/mL penicillin, and 100 mg/mL streptomycin) for 1 h at 37 °C. After digestion, the samples were shaken vigorously for 5 min, and 21-gauge syringes were used to dissociate the cells mechanically. The cells were filtered through a 100-μm filter, pelleted and washed twice with PBS. The freshly prepared cell suspensions were subjected to scRNA-seq and flow cytometry staining.

## Single-cell RNA sequencing

The scRNA-seq libraries were generated using a 10X Genomics Chromium Controller Instrument and a Chromium Single Cell 5′ library & gel bead kit. Briefly, cells were concentrated to approximately 1000 cells/μL and loaded into each channel to generate single-cell gel bead-in-emulsions (GEMs). After the RT step, the GEMs were broken, and barcoded cDNA was purified and amplified. The amplified barcoded cDNA was fragmented, A-tailed, ligated with adaptors and index PCR amplified. The final libraries were quantified using the Qubit High Sensitivity DNA Assay (Thermo Fisher Scientific), and the size distribution of the libraries was determined using a High Sensitivity DNA chip with a Bioanalyzer 2200 (Agilent). All the libraries were sequenced using an Illumina sequencer (Illumina, San Diego, CA) with a 150-bp paired-end run.

## Data preprocessing

scRNA-seq data preprocessing was performed by NovelBio Co., Ltd. with NovelBrain Cloud Analysis Platform (www.novelbrain.com). We applied fastq[68] with default parameter filtering of the adaptor sequence and removed low-quality reads to obtain clean data. Then, feature-barcode matrices were obtained by aligning reads to the human genome (GRCh38 Ensemble: version 100) using CellRanger v3.1.0. We performed a downsample analysis of the samples sequenced according to the mapped barcoded reads per cell of each sample and finally achieved the aggregated matrix. Cells containing more than 200 expressed genes and a mitochondrial UMI percentage less than 20% passed cell quality filtering, and mitochondrial genes were removed from the expression table.

## Dimension reduction and clustering analysis

Dimension reduction and unsupervised clustering were performed according to the standard workflow in Seurat (v4.1.1)[69]. To integrate cells from different samples into a shared space for unsupervised clustering, we used the harmony algorithm in the R package (v0.1.0)[10] to perform batch effect correction. For clustering and visualization, we applied the *FindCluster* function in *Seurat* to obtain cell clusters at various resolutions and reduced the dimensionality of the data using UMAP implemented in the *RunUMAP* function with the following settings: reduction = 'harmony', dims = 1:20.

## Analysis of differentially expressed genes

We applied the *FindMarkers* function in Seurat to identify DEGs between two groups with the min.pct parameter set at 0.2, which considers only genes expressed in more than 20% of cells. The nonparametric Wilcoxon rank-sum test was used to obtain the $p$ value for comparisons, and the adjusted $p$ value based on Bonferroni correction was calculated. Genes with adjusted $p < 1 \times 10^{-5}$ and $\log_2[\text{fold change}] > 0.25$ were considered differentially expressed.

## Functional annotation analyses

Kyoto Encyclopedia of Genes and Genomes (KEGG) and Gene Ontology (GO) enrichment analyses were carried out for DEGs between two groups or target genes of cell-to-cell communication by the R package *clusterProfiler* (v4.0.5)[70]. We considered gene pathways with $p < 0.05$ to be significantly enriched.

## Trajectory analysis

To clarify the differentiation trajectory among malignant subtypes, *Monocle* (v2.20.0)[71] was used to illustrate the differentiation of malignant cells. First, we loaded the normalized count matrices and metadata information to create a new CellDataSet object. As the count matrices had been normalized, we used the following setting: expressionFamily = uninormal. During the construction of the single-cell trajectories, we first used the *VariableFeatures* function in Seurat (v4.1.1) to filter a list of gene IDs to be used for defining progression. Then, dimensional reduction was performed using the DDRTree method. Finally, we ordered cells in the state of S1 as the root. The results were visualized using the function *plot_cell_trajectory*, with color_by = "DefineTypes", "Pseudotime" or "State ". The function *plot_genes_branched_heatmap* was used to create a heatmap to demonstrate the bifurcation of gene expression along S2 and S3.

The R package velocyto. R (v1.0.8)[72] was used to calculate RNA velocity values for each gene in tumor cells from malignant subtypes. The resulting RNA velocity vector was subsequently embedded into the UMAP space.

## Analysis of malignant tumor differentiation-related TFs

To identify TFs that may play a role in malignant tumor development, the *DifferentialGeneTest* function in monocle was applied for malignant tumors. TFs with $p$ values < 0.05 and $q$ values < 0.05 were defined as subtypes of malignant cell differentiation-related genes. At the same time, we excluded TFs with expression whose expression decreased along with pseudotime by filtering out those whose expression was greater at the starting state than at the end state.

## Transcription factor regulon analysis

The R package SCENIC (v1.1.3)[16] was used to infer the activated regulons of each subtype from malignant tumor cells. The input files consisted of the expression matrix and phenotype information. Then, the co-expression network was calculated by *GRNBoost2*, and the regulons were identified by *RcisTarget*. Next, the regulon activity for each cell was scored by *AUCell*. A differentially expressed regulon was identified by the Wilcoxon rank-sum test in the *FindAllMarkers* function in the R package Seurat with the following parameters: min.pct = 0.2, logfc.threshold = 0.25, and only.pos = T. Scaled expression of regulon activity was used to generate a heatmap.

## Cell–cell communication analysis

The R package CellChat (v1.5.0)[24] was employed to analyze cell-to-cell communication between tumor cells and other cell types. First, a CellChat object was created by grouping defined clusters. The ligand–receptor interaction database we used for analysis was "CellChatDB.human", without additional supplementation. Preprocessing steps were all conducted with default parameters. The functions *computeCommunProb* and *computeCommunProbPathway* were applied to infer the network of each ligand–receptor pair and each signaling pathway separately. A hierarchy plot, circle plot and heatmap were used as different visualization forms.

The R package NicheNet (v1.1.0)[28] was used to infer mechanisms of interaction in *POSTN*+ fibroblasts, *SPP1*+ macrophages, and malignant cells. For ligand and receptor interactions, clustered cells with gene expression over 10% were considered. The top 100 ligands and top 1,000 targets of differentially expressed genes of "sender cells" and "receiver cells" were extracted for paired ligand–receptor activity analysis. When *POSTN*+ fibroblasts or *SPP1*+ macrophage as receiver, other subtypes of fibroblasts or macrophages were considered reference cells. The function *ligand_activity_target_heatmap* in Nichenet_output was used to display the regulatory activity of ligands.

Mechanisms of interaction among macrophages, fibroblasts, CD8+ Tex cells, and malignant cells at different stages were compared. We defined niches of interest for each stage, and each niche involved at least one sender cell population and one receiver cell population at the same stage. The next steps followed the pipeline of differential NicheNet analysis between conditions of interest with default parameters. Before visualization, we defined the most important ligand–receptor pairs per niche. The functions *make_ligand_receptor_lfc_plot* and *make_ligand_activity_target_exprs_plot* were used to plot ligand expression activity and target genes, and *make_circos_lr* was applied to plot ligand–receptor pair circles.

## Single-cell copy number analysis

Copy number instability was assessed with the R package infercnv (v1.8.1)[33], which is designed to infer copy number alterations from tumor single-cell RNA-seq data. This package compares expression intensities of genes across malignant cells in advanced and recurrent tumor tissues. Epithelial cells in normal tissues were used as a reference.

The R package copykat(v1.1.0)[11] was employed to predict the aneuploid/diploid cells for epithelial cells, function copykat with the following parameters: id.type = "S", ngene.chr=1, win.size=25, KS.cut=0.1, sam.name = "test", distance = "euclidean", n.cores=20,output.seg = "FLASE".

## Signature score calculation

The signature genes (Supplementary Data 4) of CAFs, M1, and M2 from previous study[5,19–21,36], these signature scores calculated by "AddModulScore" function with default parameters in Seurat. Score expression plots were generated with the "VlnPlot" function in Seurat package. The *POSTN*+ fibroblasts and *SPP1*+ macrophages interaction signature was obtained by overlapping genes from specific ligands from the *POSTN*+ fibroblasts (sender)-*SPP1*+ macrophages (receiver) interaction and *SPP1*+ macrophages (sender)-*POSTN*+ fibroblasts (receiver) interaction at advanced stages. The cell type signatures were obtained from the top 50 marker genes in the corresponding cell type. Then, the interaction signature score and cell type signature score calculated by R package gene set variation analysis (GSVA; v1.40.1)[73] in the TCGA-HNSCC cohort.

## Correlation analysis

To compute the pearson correlations of the percentage of Cluster 1 of tumor cells and *POSTN* expression in *POSTN*+ fibroblasts, we calculated average expression of *POSTN* according to sample in *POSTN*+ fibroblasts as described in a previous study[74]. Then the cor.test function

from the R stats (v4.1.0) package was applied to compute pearson correlation coefficients and p-values.

## Survival and multivariable Cox regression analysis

The function *maxstat.test* in the R package maxstat (v0.7.25)[75] was used to divide all samples into signature score-high and signature score-low groups based on the optimal cutoff point. The hazard ratio (HR) was calculated by the Cox proportional hazards model by R package survival (v3.3.1), and the 95% CI is reported. Kaplan–Meier comparative survival analyses for prognostic analysis were carried out, and the log-rank test was used to determine statistical significance. Multivariable Cox regression was performed by R package survival (v3.3.1) by considering the confounding factors, including HPV status[76,77], margin status, and stages.

## Cell lines

Cal27 and SCC9 cells were obtained from American Type Culture Collection (Manassas, VA, USA) and were cultured in DMEM (GIBCO, #11965) supplemented with 10% fetal bovine serum (FBS). The cells were authenticated based on the morphology under microscope and growth rate. Cell lines were tested negative for mycoplasma contamination. No misidentified lines were used.

## Generation of *TFDP1*-overexpressing and knockdown cells

The cells were transfected using Lipofectamine 3000 (ThermoFisher, L3000001). *TFDP1*-overexpressing plasmids were purchased from Shanghai Nomics Co., Ltd. A plasmid vector was used as negative control. Small interfering RNAs (siRNAs) specific for *TFDP1* were purchased from Shanghai Genepharma Co., Ltd. A scrambled nontargeting siRNA was used as negative control. The cells were cultured under basal conditions in vitro for 6 h and then washed with Opti-MEM. Then, the HNSCC cell lines were transfected with plasmids or siRNAs according to the manufacturer's protocol.

## Transwell assay

The migration and invasion abilities of HNSCC cells were determined by Transwell assays (8.0 mm pore size, Corning, USA). Cells ($1.0 \times 10^5$ for migration and $2.0 \times 10^5$ for invasion) were cultured in serum-free DMEM in the upper chambers. DMEM containing 10% FBS was added to the lower chambers. After the cells had cultured for 24 h, the cells that had migrated to the opposite side of the Transwell filter were fixed with 4% paraformaldehyde and stained with crystal violet staining solution (Beyotime, Shanghai, China). For the transwell invasion assay, the top chamber was coated with Matrigel (1:10 in DMEM dilution, Corning, USA), the other procedures were the same as those used for the transwell migration assay. Five fields were randomly selected under a 100× microscope for image acquisition.

## Immunohistochemical staining

Formalin-fixed, paraffin-embedded (FFPE) lymph node tissues were separately sliced into 4-μm sections and mounted on glass slides. The slides were baked at 65 °C overnight. After deparaffinization and hydration, these slides were boiled in citrate buffer at 100 °C for 15 min. Subsequently, a 3% $H_2O_2$ solution was used to block endogenous peroxidase activity for 20 min. To prevent nonspecific antibody binding, the slides were then incubated with 5% normal goat serum for 1 h at room temperature. Then these slides were incubated at 4 °C overnight with an anti-foxp3 primary antibody (Abcam, ab210034, 1:500). After 3 washes with TBST, the slides were incubated with an HRP-conjugated goat anti-rabbit/mouse secondary antibody (GeneTech, GK500705) for 1 h at room temperature. The sections were stained with DAB and then counterstained with haematoxylin according to the manufacturer's instructions. The 3Dhistech Pannoramic Scan system was used for image acquisition.

## Immunofluorescence staining

Four-micron-thick sections were dewaxed in xylene, rehydrated in alcohol, and subjected to heat-induced antigen retrieval in 0.1 M 95–99 °C sodium citrate (pH 6.0) for 10 min. After washing, the sections were incubated with 5% donkey serum in PBS for 60 min at room temperature to block nonspecific binding. Subsequently, the slides were incubated overnight at 4 °C with the first primary antibody (anti-POSTN) and then at room temperature for 60 min in a dark chamber with the first fluorophore-conjugated secondary antibody diluted in PBS with 5% donkey serum. The slides were washed with PBS and then incubated with the second primary antibody (anti-SPP1), followed by staining with the second fluorophore-conjugated secondary antibody diluted in PBS with 5% donkey serum. The last two steps were conducted at room temperature for 60 min in a dark chamber. The 3Dhistech Pannoramic Scan system was used for image acquisition. To calculate the strength of the interaction between $POSTN^+$ fibroblasts and $SPP1^+$ macrophages, we randomly obtained 5 images of $SPP1^+$ macrophages from each slide and calculated the proportion of $POSTN^+$ fibroblasts that colocalized with the $SPP1^+$ cells.

## Multiplexed immunofluorescence staining

Multiplexed immunofluorescence staining of 4-μm formalin-fixed, paraffin-embedded sections was performed using the PANO 4-plex IHC kit (abs50012, Absin) according to the manufacturer's instructions. Different primary antibodies were sequentially applied, followed by horseradish peroxidase-conjugated secondary antibody incubation and tyramide signal amplification. The glass slides were microwave heat-treated following each round of TyramideSignal Amplification. Nuclei were stained with 4′–6′-diamidino- 2-phenylindole (DAPI, D9542, Sigma–Aldrich) after labeling human antigens. The following antibodies were used: anti-TFDP1 primary antibody (Proteintech, 11043-1-AP, 1:200), anti-CK5 primary antibody (Abcam, ab52635, 1:200), anti-POSTN primary antibody (Abcam, ab152099, 1:1000), anti-α-SMA antibody (Abcam, ab124964, 1:1000), anti-SPP1 antibody (Abcam, ab214050, 1:1000), anti-CD68 antibody (Abcam, ab955, 1:3000), anti-CXCL13 (Abcam, ab246518, 1:1000), anti-CD8 (CST, #70306, 1:400), anti-PD1 (Abcam, ab216352, 1:50), and p-ERK (Abcam, ab201015, 1:500). The 3Dhistech Pannoramic Scan system was used for image acquisition.

## Statistics and reproducibility

All the statistical analyses were performed using R (version 3.6.1). Student's $t$ test, Wilcoxon rank-sum test, Pearson's chi-square test, log-rank test, Pearson's correlation coefficient and Spearman's rank correlation coefficient were utilized in this study. No sample size calculation was performed. We followed the routine biological replicate requirement in experiment section, $n \geq 3$ for each group. For sequencing data, we excluded low-quality cells if abnormalities exist in (1) cell library sizes; (2) the numbers of expressed genes; (3) the proportion of mitochondrial gene counts. The details of cut-off line could be checked in Methods. Randomization is not relevant to our study.

## Reporting summary

Further information on research design is available in the Nature Portfolio Reporting Summary linked to this article.

## Data availability

The bulk RNA-seq publicly available data used in this study are available in the TCGA portal (http://gdac.broadinstitute.org/) and the Gene Expression Omnibus under accession code GSE173855[37]. The processed publicly available scRNA-seq data used in this study are available in the Gene Expression Omnibus under accession code GSE188737[25], GSE182227[26], and GSE234933[27]. The raw data of single-cell RNA-seq generated in this study were deposited in Genome Sequence Archive (GSA) with accession ID HRA004648. Since these data are related to human genetic resources, raw data can be obtained directly by requesting and following the GSA guidelines for academic use at https://ngdc.cncb.ac.cn/gsa-human/browse/HRA004648 after the user log in to the GSA database with the email address of the academic institution. The request will be responded to within two weeks. Once access is granted, users have six months to download the data. The guidance for making a data access request of GSA for humans can be downloaded from https://ngdc.cncb.ac.cn/gsa-human/document/GSA-Human_Request_Guide_for_Users_us.pdf. The signature gene lists from other studies are listed in Supplementary Data 4. The remaining data are available within the Article, Supplementary Information or Source Data file. Source data are provided with this paper.

## Code availability

Codes were implemented in R 4.1.0 and are deposited in https://github.com/hedyBao/HNSCC_scRNA/tree/main.

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

## Acknowledgements

The authors would thank Dr. Youqiong Ye from Department of Immunology and Microbiology, Shanghai Jiao Tong University School of Medicine for consultation in bioinformatics analysis. This work was supported by the National Natural Science Foundation of China (NSFC: 82173451, Y. H.), Project of Biobank (YBKB202105, Y. H.) from Shanghai Ninth People's Hospital, Shanghai Jiao Tong University School of Medicine, and Shanghai Municipal Health Commission (No. 2022LJ001, Y. H.).

## Author contributions

L.Z.L., C.W.D., and L.F. acquired tissues and data, B.R.J. and S.Y. performed data analyses, M.X.Y. and S.J.J. performed experiments, L.Z.L. and M.X.Y. developed the study design and interpreted data, H.Y. supervised the study. All the authors contributed to writing the manuscript.

## Competing interests

The authors declare no competing interests.

### Ethical statement

This study was reviewed and approved by the local medical ethics committee of Shanghai Ninth People's Hospital Affiliated with Shanghai Jiao Tong University. Written informed consent was obtained from each patient prior to sample collection.
