## [Peer Review File · Nature Communications]

Single-Cell Deciphering of Progression Trajectories of the Tumor Ecosystem in Head and Neck CancerReviewers' Comments:

Reviewer #1:

Remarks to the Author:

The manuscript by Liu et al. addressed possible evolutionary trajectories and transition fates of cells in HNSCC at different clinical stages, including tumor initiation, progression, recurrence and metastasis. Through integrated analyses of scRNA-seq data, authors provide a comprehensive landscape of the TME in stepwise HNSCC progression at single-cell resolution and report the critical subpopulation of malignant cells and its role in tumor progression. The manuscript is well-written, the study design is original with innovative potential and presented data are interesting and of high clinical relevance for the field of interest. However, though some presented data are supported by analysis of publicly accessible bulk RNA-seq datasets and immunostaining analysis, several final conclusions are not supported by experimental evidence and remain speculative as outlined in more detail below.

1. Worth noting that all patients (n=13) had tumors in the oral cavity and other anatomical subsites for HNSCC were not represented in this study. Obviously, samples from recurrent tumors were not from treatment-naïve patients as stated at page 4, which needs to be corrected and information on therapy should be added to Supplemental Table S1. In addition, authors should also include information on epidemiological risk factors (e.g. tobacco, alcohol, viruses) in Supplemental Table S1.
2. To demonstrate clinical relevance of their findings by scRNA-seq profiling, authors analyzed molecular and clinical data from TCGA-HNSC and explored differences in overall survival by KM plots (Fig. 2D, Fig. 3D, Fig. 4D, Supplemental Fig. 2F & J). However, statistically significant differences are modest and should be confirmed by multivariate Cox regression hazard models adjusted for all relevant prognostic risk factors. Did the authors observe differences for subgroups related to anatomical subsites? Finally, the number of cases at risk for most KM plots is n=500 except for Fig. 4D (n=501) and Supplemental Fig. 2J (n=546) – why?
3. MYBL2 and TFDP1 are activated in cancer cells of cluster 1 and top ranked among TF regulons (Fig. 2H-I). Furthermore, TFDP1 is associated with worth OS in TCGA-HNSC (Fig. 2J), but no data on clinical relevance is given for MYBL2. Authors should consider a survival analysis for subgroups of tumors from TCGA-HNSC in which both transcription factors are co-expressed, absent or only one is highly expressed. The manuscript would benefit from experimental evidence substantiating the conclusion that TFDP1 and MYBL2 regulation promote cancer development and classify an aggressive subgroup of malignant epithelial cells in HNSCC progression. As examples: Are cancer cells with co-expression of both transcription factors as assessed by co-IF staining on tumor sections or spatial transcriptomics enriched at specific regions such as the invasive front? Does gain of function of one or both transcription factors promote malignant transformation of normal mucosal cells in 2D and 3D cultures or their loss of function impair tumor-relevant traits in cancer cell lines?
4. Data presented in Fig. 3 suggest that POSTN+ fibroblasts shape an aggressive phenotype of HNSCC through ECM remodeling and immune regulation. However, does the analysis of TCGA-HNSC confirm differences in the immunophenotype, quantity or quality of immune cell subsets between HNSCC (in particular OSCC) with high versus low POSTN transcript levels (Fig. 3D)? Any association between POSTN transcript values and abundance of cluster 1 cancer cells for TCGA-HNSC?
5. In a cohort of 68 HNSCC samples, authors confirmed the prevalence of POSTN+ fibroblasts and SPP1+ macrophages in the tumor stroma and representative images as well as quantitative data are presented in Fig. 3K. Are these tumor sections derived from the n=13 patients, from which samples were used for scRNA-seq or from an independent cohort. If latter, please provide clinical information in a Supplemental Table and include relevant information in the section Materials and Methods. By multiplex IF staining with tissue sections of this cohort authors could investigate the potential association between the prevalence of POSTN+ fibroblasts and abundance of distinct immune cell subsets in the TME.
6. Data presented in Fig. 4 indicate that the mutual interaction between POSTN+ fibroblasts and SPP1+ macrophages promotes desmoplastic structure formation in the TME of more advanced HNSCC. Though Fig. 4B shows representative images of a co-IF staining for POSTN and SPP1, experimental evidence to substantiate the final conclusion that this combination promotes formation of immune-

excluded desmoplastic structures and exclusion of T-cell infiltration is missing. Authors should consider a multiplex co-IF staining including antibodies for relevant T cell subsets and desmoplastic structure or alternatively conduct a histopathological staining to visualize desmoplastic structure on serial tissue sections.

7. In the discussion at page 21, authors state that the interaction between POSTN+ fibroblasts and SPP1+ macrophages contributes to ECM remodeling and coordinates to form a desmoplastic microenvironment by enhancing tumor cell ECM-receptor interactions, cell-substrate junction organization and the PI3K–Akt signaling pathway. Authors should present data on differences in phosphorylation of AKT and downstream targets in cancer cells on tumor sections with varying amounts of POSTN+ fibroblasts and SPP1+ macrophages or even better their spatial distribution within larger tumor areas. Authors claim that their study highlights the potential value of identifying and establishing therapeutic strategies targeting POSTN+ fibroblasts, SPP1+ macrophages, or the molecules involved in their crosstalk to inhibit HNSCC progression. Therapeutic strategies targeting HNSCC progression are rather unlikely as most patients are diagnosed at advanced stages and fundamental goal of the therapy of primary tumors is a treatment with curative intent. Pharmacological prevention of progression might be relevant for premalignant lesions in the oral cavity, but this was not the focus of this study.

8. Differences in the infiltrative frequency of CD4+ Tregs between the LN-out and LN-in groups are modest (Supplementary Fig. 5A-B) and should be confirmed by quantitative assessment (IF staining or FACS) in samples from a larger HNSCC cohort.

9. Are differences in expression of representative ligand-receptor pairs (Supplemental Fig. S5 F, violin plots) between CD8 Tex cells and tumor cells among LN-in versus LN-out samples statistically significant – please provide p values? Do LN-out samples show a higher relative frequency of cancer cells with phospho-ERK1/2 and phospho-STAT1/3 staining (by either IF or IHC staining) as compared to LN-in samples and is the staining intensity positively associated with the abundance of CXCL13+ Tex cells? In Fig. 5J, authors show the proportion of CXCL13+ Tex in LN-in and LN-out samples (n=10 for each group), but again information on the source of samples is missing.

10. The last section of the study addressed distinct phenotypes of malignant epithelial cells favoring HNSCC recurrence. Authors probed the tumor niche atlas in recurrent stage and primary stage tumors from different patient, but no matched samples for primary and recurrent tumors were included limiting the enthusiasm on presented results. Fig. 6A shows differences in global CNV scores, but it is unclear whether hot spot regions of copy number gain or losses exist, which might explain some differences in gene expression (Fig. 6B). Are molecular differences shown in Fig. 6, such as cell cycle and immune checkpoint genes, oxidative phosphorylation or amplification levels of selected candidate genes also evident in bulk DNA- or RNA-seq data of published studies with primary and recurrent HNSCC from larger cohorts?

11. Under the present guidelines of transparency authors should upload the scRNA-seq data at a publicly available database and provide all relevant data matrices for figures as source data.

Reviewer #2:

Remarks to the Author:

In this manuscript by Liu et al, the team from a high volume centre specialising in oral cancer treatment profile a series of pre-cancerous and cancerous SCC from the oral cavity using scRNAseq and attempt to build a genetic/transcriptomic framework of progression from normal tissue/pre-cancer, early, advanced and recurrent cancers, but placing tumours collected from different patients into that specific continuum. they also perform a similar analyses using normal, early (no ECE) and late (ECE+) nodes. Their findings are either validated using TCGA data for outcome or with mIHC using tissue that is available in their high volume centre. The scRNAseq data generated good quality RNAseq data for about 120K cells derived from 13 patients (26 samples), and translates to about 4K cell per sample with about 1.6K gene per cell, which is a good dataset and QC measure, and will undoubtedly be an excellent resource for future analyses as well.

However, in reviewing the manuscript I have a number of concerns that need to be addressed some of which challenges the questionable assumptions and are therefore outright unacceptable (but correctable) and others are more about the lack of orthogonal validations, which I will address in turn.

1. One of my major concerns relate to Figure 2 and Figure 6 and the analyses therein. In all previous scRNAseq and bulk RNAseq datasets, it is clear that tumors are distinct in individual tumours and hence the fact that they have been analysed together and put into the same continuum/trajectory is extremely worrying, unorthodox and scientifically incorrect. These are distinct genetic entities and the assumption that they can be analysed together is highly erroneous. It would have made more sense to focus on the few patients that had multiple samples from the same patient (eg NT-E or NT-E-A, A/E and LN+ etc). It makes no sense to run the cluster analyses the way it has been done and then declare that bunch of cancer related genes (mostly comparing normal with tumor) is prognostic!!! That to me is an unacceptable notion. Instead I would have preferred a much tighter longitudinal analyses, and then identify similarities between patient sets!
2. Similarly, comparing 4 distinct recurrent patients (Figure 6) with a number of de novo cancer merely tells us the difference between these distinct individuals and says nothing about progression from early to late cancers, since the recurrences are NOT from the same patient (and I notice have an overrepresentation of female patients, which in itself could skew the data!). There are many statements made about the gene difference being attributed to invasion and metastasis, even though it is clear that these very patients were operated on BECAUSE they didn't have distant diseases, which totally blows the assumptions out of the water. At the very least, the authors should have attempted some bulk analysis on the original tumors that these recurrences originated from. These could have been mitigated if the authors had tried to run some orthogonal validation from existing single cell or even bulk datasets, but none of these were done.
3. The non tumor samples are a bit problematic in that there is no mention on how and where these were derived from and the fact that a significant proportion of these were aneuploid (or had CNVs). These need to be re-analysed: are the patient specific? do these influence the clusters which are malignant but have NT cells? perhaps it even gives you an opportunity to have 2 different NT-genetically normal epithelium and those with CNVs which are truly pre-cancerous? and these can be analysis separately in a spectrum.
4. it makes more sense to analyse the data in this fashion for the fibroblasts and macrophages as in figures 3 and 4, and these represent some intriguing findings that certainly need further validation. Even in these, it doesn't make sense to add the LN samples representing progression across the malignancy scale for fibroblasts and macrophage subpopulations. Furthermore the latter needs more statistical tests to be believed and orthogonal validation. The authors should pull out published data from the original Puram or Kurten datasets they refer to, or can find more resources in the following publications: Quah et al (PMID: 36973261), Puram and colleagues (PMID: 37012457), and Zhang et al (PMID: 36928331), some or all of which could lend strong support to their conclusions, especially with regards to the cell chat and nechemet analyses of interactomes. I would also suggest that they authors contextualise their fibroblast and macrophage subpopulations to currently known subtypes (eg iCAF, myCAF, ApCAF, M1 or M2 macrophages etc). The dataset of 68 patients used to validate their findings has also not been well described- who are these patients, how were they treated? are they a consecutive or highly selected dataset?
5. all the cellular networks data remain circumstantial without validations, and the authors are referred to the many excellent single cell papers described above that they can access and re-analyze to their specific condition and question. I also suggest that when analysing the interactome with tumor cells, these should be individual patient specific analyses rather than trying to combine malignant cells (see point 1). There was some attempt at looking at proximity between POSTN fibroblasts and SPP1 macrophages, but I have no idea where this dataset comes from, what was done and how the analysis was conducted. Validation experiment such as these are the cornerstone to accept the circumstantial discovery data provided by expensing scRNAseq, and I do advise the authors to pay attention to these and expand on them, including re-analyses of existing bulk and scRNAseq data as mentioned before.
6. The analysis of the ENE data has many of the same issues described before, but if these are matched to ENE negative nodes from the same patient, it supports the data even better and these

need to be described...there is NO MENTION at all at how many or who the ENE+ and ENE- patients are, and again no attempt at orthogonal validation. The data concerning high CXCL13 Tex is interesting, but I would also like to know if these are present in advanced or recurrent tumors as these can also represent exhaustion over time.

7. sweeping statements on therapeutics such as targeting interactions or CXCL13 are pointless, inaccurate and even their own data suggest is wrong- these will only target a specific subpopulation in your own dataset, so why target them?!

All in all, this is a landscape paper that would be a valuable resource but has a number of erroneous assumptions and almost zero attempt at validation, which make these findings circumstantial at best.

Reviewer #3:

Remarks to the Author:

Liu et al perform scRNA-seq on 26 tumor specimens to delineate expression heterogeneity within the tumor progression model, including analysis of adjacent normal, precancerous, early stage cancer, advanced stage cancer, ENE- lymph node, ENE+ lymph node, and recurrent cancer samples.

A fundamental problem with the present work is that it generally fails to put findings in the context of previously published work. Without this contextualization, the novelty of the present work is unclear. In particular:

1. The authors should relate malignant and stromal cell subpopulations they uncover to previously described subpopulations in Puram et al, Cell, 2017 acknowledging of course that differences in scRNA-seq technique may create differences in subpopulations defined.
2. The authors should better relate malignant and stromal cell pseudotime subpopulation analysis to recently published similar data from Choi et al, Nature Communications, 2023. While the authors do reference Choi et al in the introduction, there needs to be a more specific discussion of why the present study aims to provide novel information and how the results relate to those previously published.
3. The authors identify POSTN+ fibroblasts and SPP1+ macrophages as key stromal cell types involved in tumor progression, and a primary claim related to poor outcomes is from analysis of TCGA RNA-seq data. These findings should be contextualized with prior research. For example, Luo et al., Nature Communications, 2022 describe an interaction between CAFs and SPP1+ macrophages using scRNA-seq data. Wenhua et al., Cancer Medicine, 2023 report that overall POSTN expression in the TCGA cohort is not associated with survival. In addition, Bie & Zhang, J Immunol Res, 2021 find a similar association between SPP1 and survival in TCGA.

Other comments:

1. The authors should provide histologic evidence to validate the identity of each sample analyzed – this is particularly important for pre-cancer samples to validate this designation histopathologically.
2. It is unclear how the authors designate early (E) vs. advanced (A) tumors. From Table S1, it appears that a T2N1 sample (P13) was listed as early, while a T3N1 sample (P8) was listed as advanced. Based on AJCC staging of oral cancers, both of these patients would be classified as overall stage III, making it unclear what differentiates the two. The analysis should be redone with a more consistent classification.
3. How was fibroblast and myeloid cell clustering conducted? Can the authors validate fibroblast and myeloid clusters in the context of published work? For example, Galbo et al., Clinical Cancer Research, 2021 identify 6 pan-cancer CAF subtypes – can the authors validate their own clustering in the context of these subtypes?
4. The immunofluorescence images provided all need to be brighter and clearer in order to enable independent interpretation of their utility in validating the findings of the sequencing analysis.
5. Please clarify what CD8-Tex cells are and how these are identified.
6. The authors provide a variety of ligand-receptor pairs across CD8 Tex and tumor cells suggesting

bidirectional signaling that is upregulated during extranodal extension. However, they do not provide any validation of these findings, making it unclear whether these are just an artifact of multiple hypothesis testing that results from analysis of large scale sequencing data or true biologically meaningful results. In the absence of additional validation, the authors should tone down their claims, particularly related to tumor cell reprogramming.

7. The authors provide an analysis of differentially expressed genes in recurrent vs. advanced stage cancers and relate their analyses to potential therapeutic targets. However, given the small number of samples, these results should be validated using previously published data or functional experiments.

Response Letter

Reviewer1

The manuscript by Liu et al. addressed possible evolutionary trajectories and transition fates of cells in HNSCC at different clinical stages, including tumor initiation, progression, recurrence and metastasis. Through integrated analyses of scRNA-seq data, authors provide a comprehensive landscape of the TME in stepwise HNSCC progression at single-cell resolution and report the critical subpopulation of malignant cells and its role in tumor progression. The manuscript is well-written, the study design is original with innovative potential and presented data are interesting and of high clinical relevance for the field of interest. However, though some presented data are supported by analysis of publicly accessible bulk RNA-seq datasets and immunostaining analysis, several final conclusions are not supported by experimental evidence and remain speculative as outlined in more detail below.

Response: We are very pleased that the reviewer considers our study is interesting. We greatly appreciated constructive suggestions from the reviewer. We have addressed all comments with point-to-point responses as follows.

1. Worth noting that all patients (n=13) had tumors in the oral cavity and other anatomical subsites for HNSCC were not represented in this study. Obviously, samples from recurrent tumors were not from treatment-naïve patients as stated at page 4, which needs to be corrected and information on therapy should be added to Supplemental Table S1. In addition, authors should also include information on epidemiological risk factors (e.g. tobacco, alcohol, viruses) in Supplemental Table S1.

Response: Thank the reviewer for pointing out the limitation about sample collection. Actually, oral squamous cell carcinoma (OSCC) is epidemiologically prominent in HNSCC [1], which could be a good starting point for us to study HNSCC.

As for the clinical information, we are sorry to make the reviewer confused and we have updated clinical information of our cohort in **Supplementary Table 1**. We have included tobacco use, alcohol use, perineural invasion (PNI), lymphovascular invasion (LVI), extranodal extension (ENE), and previous treatment in the updated table. In the meantime, related description has been corrected in the manuscript, page 4 (highlighted).

2. To demonstrate clinical relevance of their findings by scRNA-seq profiling, authors analyzed molecular and clinical data from TCGA-HNSC and explored differences in overall survival by KM plots (Fig. 2D, Fig. 3D, Fig. 4D, Supplemental Fig. 2F & J). However, statistically significant differences are modest and should be confirmed by multivariate Cox regression hazard models adjusted for all relevant prognostic risk

factors. Did the authors observe differences for subgroups related to anatomical subsites? Finally, the number of cases at risk for most KM plots is n=500 except for Fig. 4D (n=501) and Supplemental Fig. 2J (n=546) - why?

Response: Thank the reviewer's suggestion. To assess whether the factors mentioned by the reviewer are independent predictors in the corresponding plots above, we performed a multiple Cox regression analysis that newly included age, gender, and stage as variables. The results showed cluster 1 of epithelial cells, *POSTN*⁺ fibroblasts, *SPPI*⁺ macrophages, *TFDPI*, the interaction signature score of *POSTN*⁺ fibroblasts and *SPPI*⁺ macrophages are independent predictors (**Figure for Reviewer [Fig.R]1a-e**). The **Supplementary Fig. 2h** shows the association of other epithelial cells and OS, which indicates no significant related to OS. To compare the liability of all epithelial cell clusters as prognostic factors, we performed multivariate Cox regression analysis considering all clusters as variables, and we found the cluster 1 of epithelial cells was the only predictor that was significantly associated with poorer prognosis (**Fig. R1f**). We added these results in the revised manuscript, and added **Fig. R1a** as **Supplementary Fig. 2i**, **Fig. R1b-c** as **Supplementary Fig. 3c-d**, **Fig. R1d** as **Supplementary Fig. 2t**, **Fig. R1e** as **Supplementary Fig. 4d**, and **Fig. R1f** as **Supplementary Fig. 2j**.

Fig. R1. C1 of malignant epithelial cells, *POSTN*⁺ fibroblasts, *SPPI*⁺ macrophages, *TFDPI*, and the interaction signature score of *POSTN*⁺ fibroblasts and *SPPI*⁺ macrophages are independent predictors. Multivariate Cox regression model analysis, which included the factors of patient age, gender, TNM status, and the deconvoluted score of C1 of malignant epithelial cells (a), the top 50 signature score of *POSTN*⁺ fibroblasts (b), the top 50 signature score of *SPPI*⁺ macrophages (c), *TFDPI* (d) and the interaction signature score of *POSTN*⁺ fibroblasts and *SPPI*⁺ macrophages (e) in the TCGA-HNSCC data. (f) The hazard ratios for OS between the high- and low-proportion groups for each malignant cell cluster in the deconvoluted TCGA-HNSCC data.

As for the anatomic subsites concern, however, the anatomical subsites in TCGA refer to oral cavity, oropharynx and hypopharynx. When it comes to oral cavity, there was no information for specific subsites, such as tongue, buccal, mouth floor, et.al. Therefore, we didn't include this confounding factor.

As for the sample number issue, we checked the sample information to confirm how many cases can be involved in survival analysis. The total number of TCGA-HNSCC cohort is 546, including 44 adjacent normal tissues, 500 primary tumors, and 2 metastatic tumors. While 5 out of 500 primary samples were paraffin-embedded tissues, and one case without overall survival (OS) information, so the above survival analysis was regenerated using 494 samples of and we have replaced related figures with new ones (Fig. 2D and K, Fig. 3D and H, Fig. 4D, and Supplementary Fig. 2h).

3. MYBL2 and TFDP1 are activated in cancer cells of cluster 1 and top ranked among TF regulons (Fig. 2H-I). Furthermore, TFDP1 is associated with worth OS in TCGA-HNSC (Fig. 2J), but no data on clinical relevance is given for MYBL2. Authors should consider a survival analysis for subgroups of tumors from TCGA-HNSC in which both transcription factors are co-expressed, absent or only one is highly expressed. The manuscript would benefit from experimental evidence substantiating the conclusion that TFDP1 and MYBL2 regulation promote cancer development and classify an aggressive subgroup of malignant epithelial cells in HNSCC progression. As examples: Are cancer cells with co-expression of both transcription factors as assessed by co-IF staining on tumor sections or spatial transcriptomics enriched at specific regions such as the invasive front? Does gain of function of one or both transcription factors promote malignant transformation of normal mucosal cells in 2D and 3D cultures or their loss of function impair tumor-relevant traits in cancer cell lines?

Response: As suggested by the reviewer, we performed *MYBL2* regulon and survival analysis and we found that patients with higher expression of *MYBL2* regulon had significantly worse OS (Fig. R2a). To further assess regulatory roles of *TFDP1* and *MYBL2*, we performed a multiple Cox regression analysis that newly included *TFDP1*, *MYBL2*, age, gender, and stage as variables. The results showed that *TFDP1* is an independent predictor (Fig. R2b). Combined with experimental validation results, we ultimately decided to emphasize only the regulatory role of *TFDP1* and we also modified related description in the revised manuscript.

Fig. R2. *TFDP1* plays a more important role in promoting tumor progression than *MYBL2*. (a) Kaplan-Meier curves show overall survival in *MYBL2*-high (red) and -low (blue) in TCGA-HNSCC cohort. $P < 0.05$ in the two-sided log-rank test was considered

statistically significant. (b) Multivariate Cox regression model analysis, which included the factors of *TFDP1*, *MYBL2*, age, gender, stages, and patient outcomes in TCGA-HNSCC cohort.

For experimental validation, we also focused on *TFDP1*. First, we evaluated expression level of *TFDP1* in epithelial cells in different stages. Consistent with the analysis results, though there was no significant difference between NAT and E group, we indeed found that *TFDP1*⁺ epithelial ratio was higher in the A stage than that in the E stage, which indicated that *TFDP1* was associated with HNSCC progression (**Fig. R3a-b**). Furthermore, outcome analysis of our validation cohort showed that patients with higher *TFDP1* expression met shorter survival (**Fig. R3c**). According to current studies, *TFDP1* is associated with aggressive phenotypes of cells [2,3]. Therefore, we utilized the 2D cell culture model with *TFDP1* overexpression or knockdown to evaluate the relevance between *TFDP1* and invasion phenotype. By overexpressing *TFDP1* in SCC9 cell lines and knocking down *TFDP1* in Cal27 cell lines (both are HNSCC cell lines), we observed that HNSCC cells with higher *TFDP1* expression exhibited higher migration and invasion capability (**Fig. R4**). We have added these results in the revised manuscript, and added **Fig. R3** as **Fig. 2J**, and **Fig. R4** as **Fig. 2L**.

Fig. R3. Validation of TFDP1 function in HNSCC. (a) Representative images of multiplex immunohistochemistry (mIHC) staining of *TFDP1*⁺ epithelial cells (*TFDP1*⁺ *CK5*⁺ double positive) in HNSCC tumor and nonmalignant samples. Scale bar, 100 μm and 50 μm as indicated. (b) The quantitative results showed *TFDP1*⁺ epithelial cell ratio in different stages. (c) The Kaplan-Meier overall survival curves of validation cohort patients stratified by *TFDP1* expression level. *P* values were calculated by Student's *t*-test in (b). Two-sided log-rank test was used in (c). **p*<0.05, ****p*<0.001.

Fig. R4. Validation of TFDP1 function in HNSCC *in vitro*. (a) Images of Transwell assays for migration and invasion in different cell lines with *TFDP1* overexpression or knockdown. Scale bar = 100 μ m. (b) The quantitative analysis of migration and invasion capability. *P* values were calculated by Student's *t*-test. **p*<0.05, ***p*<0.01, ****p*<0.001

4. Data presented in Fig. 3 suggest that POSTN⁺ fibroblasts shape an aggressive phenotype of HNSCC through ECM remodeling and immune regulation. However, does the analysis of TCGA-HNSC confirm differences in the immunophenotype, quantity or quality of immune cell subsets between HNSCC (in particular OSCC) with high versus low POSTN transcript levels (Fig. 3D) Any association between POSTN transcript values and abundance of cluster 1 cancer cells for TCGA-HNSC?

Response: Since CD8⁺ T cells are the most important immune cells in the TME and the classification for immunophenotype, we compared the CD8⁺ T cell infiltration between *POSTN* expression low and high group which stratified by the median expression in TCGA-HNSCC cohort. We found *POSTN*^{high} group showed significantly lower CD8⁺ T cell infiltration compared to *POSTN*^{low} group (**Fig. R5a**), suggesting *POSTN*⁺ fibroblasts were associated with a desert immune microenvironment. In addition, we performed Pearson correlation analysis to reveal the mean expression of *POSTN* in *POSTN*⁺ fibroblasts significantly correlated with abundance of cluster 1 cancer cells in our cohort (**Fig. R5b**). We have added these results in the revised manuscript, and added **Fig. R5a** as **Supplementary Fig. 4i**, and **Fig. R5b** as **Supplementary Fig. 4t**.

Fig. R5. The *POSTN*^{high} group is associated with low-CD8⁺ T infiltrated microenvironment and the abundance of cluster 1 cancer cells. (a) Differences in the infiltrated CD8⁺ T cells between *POSTN*^{high} and *POSTN*^{low} group in TCGA-HNSCC cohorts. The Wilcoxon test was used to determine the statistical significance of the difference, and $P < 0.05$ was considered statistically significant. (b) The correlation between the mean expression of *POSTN* in *POSTN*⁺ fibroblasts and abundance of cluster 1 cancer cells in our cohort.

5. In a cohort of 68 HNSCC samples, authors confirmed the prevalence of *POSTN*⁺ fibroblasts and *SPP1*⁺ macrophages in the tumor stroma and representative images as well as quantitative data are presented in Fig. 3K. Are these tumor sections derived from the n=13 patients, from which samples were used for scRNA-seq or from an independent cohort. If latter, please provide clinical information in a Supplemental Table and include relevant information in the section Materials and Methods. By multiplex IF staining with tissue sections of this cohort authors could investigate the potential association between the prevalence of *POSTN*⁺ fibroblasts and abundance of distinct immune cell subsets in the TME.

Response: Thank the reviewer for these questions. The 68 HNSCC samples are collected from our independent validation cohort, whose clinical information has been included in **Supplementary Table 2**. We have also updated Materials and Methods section (highlighted).

As for the validation of the association between *POSTN*⁺ fibroblasts and immune infiltration, we found CD8⁺ T cell infiltration was significantly different between fibroblasts with *POSTN* low expression and high expression group, so we chose this immune cell type for further experiments (**Fig. R5a**).

Again, we used our validation cohort samples and by mIHC assay, we found that when *POSTN*⁺ fibroblasts widely distributed in the TME, forming desmoplastic barrier, CD8⁺ T cell infiltration was significantly decreased whereas *SPP1*⁺ macrophages infiltration increased (**Fig. R6**). We have added these results in the revised manuscript, and added

Fig. R6 as Supplementary Fig. 4m.

Fig. R6. The association between *POSTN*⁺ fibroblasts, *SPP1*⁺ macrophages and *CD8*⁺ T cells. (a) Representative mIHC staining of tumor sections. Dapi (blue), α -SMA (green), *POSTN* (red), *CD68* (orange), *SPP1* (grey), and *CD8* (purple), in individual and merged channels are shown. Scale bar = 50 μ m. (b) Correlation between *POSTN*⁺ fibroblasts ratio and *CD8*⁺ T cell ratio (n= 40). (c) Correlation between *POSTN*⁺ fibroblasts ratio and *SPP1*⁺ macrophages ratio (n= 40).

6. Data presented in Fig. 4 indicate that the mutual interaction between *POSTN*⁺ fibroblasts and *SPP1*⁺ macrophages promotes desmoplastic structure formation in the TME of more advanced HNSCC. Though Fig. 4B shows representative images of a co-IF staining for *POSTN* and *SPP1*, experimental evidence to substantiate the final conclusion that this combination promotes formation of immune-excluded desmoplastic structures and exclusion of T-cell infiltration is missing. Authors should consider a multiplex co-IF staining including antibodies for relevant T cell subsets and desmoplastic structure or alternatively conduct a histopathological staining to visualize desmoplastic structure on serial tissue sections.

Response: According to the reviewer's suggestion, by mIHC assay (shown above), we observed that *CD8*⁺ T cell infiltration was negatively correlated with *POSTN*⁺ fibroblasts while *SPP1*⁺ macrophages infiltration was positively correlated with *POSTN*⁺ fibroblasts ratio. These results indicated that *POSTN*⁺ fibroblasts interacted with *SPP1*⁺ macrophages and formed a desmoplastic TME, excluding *CD8*⁺ T cell from infiltration (**Fig. R6**).

7. In the discussion at page 21, authors state that the interaction between *POSTN*⁺ fibroblasts and *SPP1*⁺ macrophages contributes to ECM remodeling and coordinates to

form a desmoplastic microenvironment by enhancing tumor cell ECM-receptor interactions, cell-substrate junction organization and the PI3K–Akt signaling pathway. Authors should present data on differences in phosphorylation of AKT and downstream targets in cancer cells on tumor sections with varying amounts of POSTN+ fibroblasts and SPP1+ macrophages or even better their spatial distribution within larger tumor areas. Authors claim that their study highlights the potential value of identifying and establishing therapeutic strategies targeting POSTN+ fibroblasts, SPP1+ macrophages, or the molecules involved in their crosstalk to inhibit HNSCC progression. Therapeutic strategies targeting HNSCC progression are rather unlikely as most patients are diagnosed at advanced stages and fundamental goal of the therapy of primary tumors is a treatment with curative intend. Pharmacological prevention of progression might be relevant for premalignant lesions in the oral cavity, but this was not the focus of this study.

Response: For PI3K-Akt signaling pathway in tumor cells, we are sorry that we cannot obtained sufficient validation results. Therefore, we deleted related description about PI3K-Akt signaling pathway in the revised manuscript.

As for the discussion about therapeutic significance, we have revised inappropriate description in the revised manuscript.

8. Differences in the infiltrative frequency of CD4+ Tregs between the LN-out and LN-in groups are modest (Supplementary Fig. 5A-B) and should be confirmed by quantitative assessment (IF staining or FACS) in samples from a larger HNSCC cohort.

Response: According to reviewer's advice, we performed IHC on our lymph node validation cohort. We found that Foxp3⁺ cell ratio was significantly higher in ENE⁺ samples, which was consistent with *in silico* results (Fig. R7). We have added these results in the revised manuscript, and added Fig. R7 as Supplementary Fig. 5c.

Fig. R7. Difference of Treg infiltration between ENE⁻ and ENE⁺ lymph nodes. (a) Representative IHC staining of Foxp3 in lymph nodes. Scale bar, 200 µm and 50 µm as

indicated. (b) Proportion of Foxp3⁺ cell is compared between ENE⁻ and ENE⁺ samples (n=8 and 7 for each group). The two-sided Wilcoxon test was used to determine the statistical significance of the difference. **p*<0.05.

9. Are differences in expression of representative ligand-receptor pairs (Supplemental Fig. S5 F, violin plots) between CD8 Tex cells and tumor cells among LN-in versus LN-out samples statistically significant – please provide p values? Do LN-out samples show a higher relative frequency of cancer cells with phospho-ERK1/2 and phospho-STAT1/3 staining (by either IF or IHC staining) as compared to LN-in samples and is the staining intensity positively associated with the abundance of CXCL13⁺ Tex cells? In Fig. 5J, authors show the proportion of CXCL13⁺ Tex in LN-in and LN-out samples (n=10 for each group), but again information on the source of samples is missing.

Response: We have added the *p* value calculated by two-sided Wilcoxon test to assess statistical difference between LN-in and LN-out CD8 Tex cells in **Supplementary Fig. 5g**.

We turned to our validation cohort 3 to analyze pERK expression level between ENE⁻ and ENE⁺ samples (**Fig. R8**). We found that pERK⁺ epithelial ratio is higher in ENE⁺ samples, which indicated that tumor cells in ENE⁺ lymph nodes showed higher ERK signaling activation level. What's more, correlation analysis showed pERK⁺ epithelial ratio is positively correlated with CXCL13⁺ Tex ratio, which suggested that CXCL13⁺ Tex may induce ERK pathway activation during lymph node metastasis process. We have added these results in the revised manuscript, and added **Fig. R8** as **Supplementary Fig. 5j**. As for the clinical information of our validation cohort, we have updated it in **Supplementary Table 2-3**.

Fig. R8. Difference of pERK expression level between ENE⁻ and ENE⁺ lymph nodes. (a) Representative mIHC staining of lymph node samples. Dapi (blue), CK5 (green), and pERK (red), in individual and merged channels are shown. Scale bar= 50 μm. (b) Proportion of pERK⁺ Epi is compared between LN-in and LN-out samples (n=8 and 9). (c) Correlation of CXCL13⁺ Tex ratio with pERK⁺ epi ratio. ***p*<0.01.

10. The last section of the study addressed distinct phenotypes of malignant epithelial

cells favoring HNSCC recurrence. Authors probed the tumor niche atlas in recurrent stage and primary stage tumors from different patient, but no matched samples for primary and recurrent tumors were included limiting the enthusiasm on presented results. Fig. 6A shows differences in global CNV scores, but it is unclear whether hot spot regions of copy number gain or losses exist, which might explain some differences in gene expression (Fig. 6B). Are molecular differences shown in Fig. 6, such as cell cycle and immune checkpoint genes, oxidative phosphorylation or amplification levels of selected candidate genes also evident in bulk DNA- or RNA-seq data of published studies with primary and recurrent HNSCC from larger cohorts?

Response: Clinically, it is difficult to acquire paired primary and subsequent relapse fresh samples of HNSCC for scRNA analysis, because relapse may occur on several years after the treatment of the primary tumor or even never occur. We also realized this limitation and interpreted in the discussion section.

For the CNV issue, we draw the heatmap by R package infercnv correspond to **Fig. 6A** and found that compared to cells in A stage samples, cells in R stage samples exhibited more gain and loss events of copy number (**Fig. R9a**). In the meantime, the Venn plot showed these genes with copy number variation significantly intersected with differential expression genes from R stage versus A stage (**Fig. R9b**). There are 347 overlapping genes with copy number gain in up-regulated genes of R stage and 178 overlapping genes with copy number loss in down-regulated genes of R stage. These results indicated that copy number variation genes in R stage were associated with differential expression genes in R-stage based on the scRNA-seq results.

For external validation of our key conclusions, we turned to one scRNA-seq data (GSE234933, Science, 2023) [4] and one bulk RNA-seq data (GSE173855, Clinical Cancer Research, 2022) [5] for further analysis. GSE234933 includes primary and relapse tumor samples (unpaired) while no staging information was available for primary tumors, which means that primary samples could also include early-stage tumors and interfere analysis results. However, given this dataset is the only accessible scRNA-seq dataset with recurrent HNSCC samples, it was still chosen for our validation work. GSE173855 includes 34 patients with paired primary and recurrent samples, we finally obtained 18 paired samples with an advanced stage and an anatomical site of oral cavity.

We calculated proliferation scores for GSE234933 based on the proliferation signature genes as previous study [6] (marker genes listed in **Supplementary Table 4**) and found cell proliferation score was also higher in primary tumors as compared to relapse samples in GSE234933 (**Fig. R9c**). Oxidative phosphorylation pathway expression both up-regulated in relapse tumors in GSE234933 (**Fig. R9d**) and GSE173855 (**Fig. R9e**). These results are consistent with our in-house cohort, indicating a lower proliferation but higher metabolic phenotype of malignant cells in the relapse cancer. As for the cell-cell interaction between malignant cells and immune components in GSE234933, we observed a significantly stronger interaction of CD47-SIRPG pair between tumor cells-myeloid cells, tumor cells-CD8⁺ T cells, and tumor cells-CD4⁺ T

cells (**Fig. R9f**), highlighting that CD47 could be an immune checkpoint on relapse HNSCC tumor cells. Finally, for specific molecules on malignant cells, we found *CDKN2A*, *EGFR*, *VEGFA*, and *TGF- β 1* were also significantly higher expressed in malignant cells in R stage for GSE234933 (**Fig. R9g**), suggesting that they could be potential therapeutic targets for recurrent HNSCC. We have added these results in the revised manuscript, and added **Fig. R9a-b** as **Supplementary Fig. 6a-b**, and **Fig. R9c-g** as **Supplementary Fig. 6d-g**.

Fig. R9 InferCNV profiles and validation analysis in public datasets between cells in A and R stage samples. (a) Inferred large-scale CNVs between cells in A and R stages. Rows correspond to individual cells between stage R and A samples (colorbars on the left) and columns correspond to genes ordered by chromosomal location (black and grey bars at the top indicate different chromosomes). (b) Venn plots show the 347 overlapping genes between inferred copy number gain genes and up-regulated genes from scRNA DEGs (the left panel), the 178 overlapping genes between inferred copy number loss genes and down-regulated genes from scRNA DEGs (the right panel). The Fisher's exact test *p*-values of the overlaps are reported. (c) Violin plots show the cell proliferation of malignant cells in P and R stage. (d) Violin plots show the cell proliferation and oxidative phosphorylation scores of malignant cells in P and R stage.

(e) KEGG enrichment analysis in oxidative phosphorylation pathway of P and R samples. (f) Bubble plots show the interaction between malignant cells and myeloid cells, CD4⁺, and CD8⁺ T cells, based on selected ligand and receptor pairs. These scores are normalized expression level, and the sizes of the bubbles indicate the significance of the interactions, calculated by CellPhoneDB. (g) Violin plots show the expression of selected genes in malignant cells in P and R stage. Data from GSE234933 is analyzed for (c, d, f, and g) and data from GSE173855 is analyzed for (e).

11. Under the present guidelines of transparency authors should upload the scRNA-seq data at a publicly available database and provide all relevant data matrices for figures as source data.

Response: We have deposited raw sequencing reads of all single-cell experiments in the Genome Sequence Archive (GSA, <https://ngdc.cnbc.ac.cn/gsa-human/>) and with data accession no. HRA004648 under project PRJCA014927. We also provided all relevant data matrices for figures as Source Data, we have claimed this in Data Availability section in the revised manuscript.

Reviewer2

1. One of my major concerns relate to Figure 2 and Figure 6 and the analyses therein. In all previous scRNAseq and bulk RNAseq datasets, it is clear that tumors are distinct in individual tumours and hence the fact that they have been analysed together and put into the same continuum/trajectory is extremely worrying, unorthodox and scientifically incorrect. These are distinct genetic entities and the assumption that they can be analysed together is highly erroneous. It would have made more sense to focus on the few patients that had multiple samples form the same patient (eg NT-E or NT-E-A, A/E and LN+ etc). It makes no sense to run the cluster analyses the way it has been done and then declare that bunch of cancer related genes (mostly comparing normal with tumor) is prognostic!!! That to me is an unacceptable notion. Instead I would have preferred a much tighter longitudinal analyses, and then identify similarities between patient sets!

Response: Thanks for the reviewer's valuable comment. Indeed, the tumors are distinct in individual tumors. Thanks to single-cell technology, more and more studies have combined tumor cells from different patients to help researchers study the different states and potential trajectory characteristics of tumor cells [7-19].

As suggested by the reviewer, related to **Fig. 2**, we performed the trajectory analysis of epithelial cells obtained from same patients, including P2, P10, and P13 with NT, Pre and E samples (**Fig. R10**). Here, we found consistent development trajectories of these three patients with results of total malignant epithelial cells in **Fig. 2G**. These results support the reliability of our conclusion on cancer cell development during HNSCC progression. We have added these results in the revised manuscript, and added **Fig. R10a-c** as **Supplementary Fig. 2p-r**.

Fig. R10. Potential trajectory of epithelial cells of single patient. Potential trajectory of epithelial cells by single patients from P2 (a), P10 (b), and P13 (c) inferred by

Monocle2. The trajectory was divided into three states indicated as S1, S2, and S3. The cluster components of each state were shown in the bar plot.

2. Similarly, comparing 4 distinct recurrent patients (Figure 6) with a number of de novo cancer merely tells us the difference between these distinct individuals and says nothing about progression from early to late cancers, since the recurrences are NOT from the same patient (and I notice have an overrepresentation of female patients, which in itself could skew the data!). There are many statements made about the gene difference being attributed to invasion and metastasis, even though it is clear that these very patients were operated on BECAUSE they didn't have distant diseases, which totally blows the assumptions out of the water. At the very least, the authors should have attempted some bulk analysis on the original tumors that these recurrences originated from. These could have been mitigated if the authors had tried to run some orthogonal validation from existing single cell or even bulk datasets, but none of these were done.

Response: Thanks the reviewer's suggestion. We have validated the results of **Fig. 6** in two public datasets in scRNA-seq (GSE234933) and bulk RNA-seq (GSE173855) to obtain the consistent results (**Please refer to the response of the Reviewer #1's comment 10**).

As for the gender composition concern, we performed principal components analysis (PCA) on in-house data and validation data (GSE234933) through pseudo bulk methods (**Fig. R11**). We found A/P and R samples separated well, while male and female samples still distributed scattered, indicating that gender factor was not the predominant difference among these samples. Thanks for the reviewer's careful consideration and we will pay more attention to the gender issue during sample screening process in our future studies.

Fig. R11. Principal component analysis of samples from in-house data scRNA-seq (a) and public scRNA-seq GSE234933 (b).

3. The non tumor samples are a bit problematic in that there is no mention on how and where these were derived from and the fact that a significant proportion of these were aneuploid (or had CNVs). These need to be re-analysed: are the patient specific? do these influence the clusters which are malignant but have NT cells? perhaps it even gives you an opportunity to have 2 different NT- genetically normal epithelium and those with CNVs which are truly pre-cancerous? and these can be analysis separately in a spectrum.

Response: For the non-tumor samples (NT), we collected them from patients' oral epithelial tissues distant from tumor sites (paired with pre and E samples). We have added representative H&E image of samples in different stages in the revised manuscript (**Supplementary Fig. 1a**).

Then, we re-analyzed the epithelial cells in NT, Pre, and E samples from the representative patient P13 by Copykat algorithm and found through some epithelial cells in NT samples were annotated as aneuploid cells, their predicted copy numbers were actually smaller than those in pre and E samples (**Fig. R12a**). We further compared CNV scores of aneuploid cells from NT, Pre, and E samples, we found aneuploid cells from NT samples had the lowest CNV scores than cells from Pre and E samples (**Fig. R12b**). In the meantime, the density plot showed that there are significant differences in the distribution of absolute value of CNV of aneuploid epithelial cells at different stages. E and Pre groups have significantly higher density in high CNV compared to NT group (**Fig. R12c**).

Since our NT group is adjacent normal tissues, as suggested by the reviewer, we further calculated differentially expressed genes from aneuploid cells compared to diploid cells in NT samples from the representative patient P13 (**Fig. R12d-e**) and found that up-regulated genes related to oncogenesis process such as 'cell growth', 'epithelial cell proliferation', and 'Wnt signaling pathways' while down-regulated genes related to physiologic functions such as 'complement activation' and 'intestinal immune network for IgA production'. These results suggested that aneuploid epithelial cells in our NT samples may in the transitional status from *bona fide* normal cells to pre-cancerous cells. We have added these results in the revised manuscript, and added **Fig. R12b, d, and e** as **Supplementary Fig.2c, d, and e**, respectively.

Fig. R12. CNV of aneuploid epithelial cells in NT samples from a representative patient, P13. (a) Inferred large-scale CNVs of epithelial cells from NT, Pre, and E samples. Rows correspond to individual cells from NT, Pre, and E samples (colorbars on the left) and columns correspond to genes ordered by chromosomal location (black and grey bars at the top indicate different chromosomes). (b) Violin and box plots show CNV scores of aneuploid epithelial cells from NT samples calculated by R package copycat at different stages. (c) The density plot shows the absolute value of CNV distribution of aneuploid epithelial cells from NT samples. (d-e) Bubble plots of GO (d) and KEGG (e) pathways for differentially expressed genes in aneuploid epithelial cells compared to diploid epithelial cells from NT samples.

4.1 it makes more sense to analyse the data in this fashion for the fibroblasts and macrophages as in figures 3 and 4, and these represent some intriguing findings that certainly need further validation. Even in these, it doesn't make sense to add the LN samples representing progression across the malignancy scale for fibroblasts and macrophage subpopulations. Furthermore the latter needs more statistical tests to be believed and orthogonal validation. The authors should pull out published data from the original Puram or Kurten datasets they refer to, or can find more resources in the following publications: Quah et al (PMID: 36973261), Puram and colleagues (PMID: 37012457), and Zhang et al (PMID: 36928331), some or all of which could lend strong support to their conclusions, especially with regards to the cell chat and nechemet analyses of interactomes.

Response: Thank the reviewer's constructive comments. We have removed the samples

from lymph node in **Fig. 3C** and **G**. Other panels in **Fig. 3** and **4** didn't include LN samples.

As suggested by the reviewer, to validate the interaction of *POSTN*⁺ fibroblasts and *SPPI*⁺ macrophages during the progression, we downloaded the scRNA-seq datasets mentioned by the reviewer and other publicly available datasets (**Table for the reviewer 1 [Table R1]**). Dataset GSE103322 [20] based on SMART-seq2 only included 98 single cells of macrophages, dataset GSE139324 [11] based on 3' end of 10X genomic platform only include CD45⁺ cells, dataset GSE225331 [21] based on Fluidigm C1 obtain cells from patient derived cultures only included epithelial cells, dataset HRA003383 [22] is inaccessible. We included dataset GSE188737 [21], GSE182227 [23], and GSE234933 [4] based on 5' end of 10X genomic platform. These three datasets all included fibroblasts and macrophages. After downloading the expression matrix data and quality control, we performed graph-based clustering and annotated clusters with their respective markers. We compared cell subtypes in three public datasets with our cell types and found the most important macrophages (*SPPI*⁺ macrophages) and fibroblasts (*POSTN*⁺ fibroblasts) were also annotated in public datasets (**Fig. R13a-b**). We obtained 2801, 3423 and 1589 cells of *POSTN*⁺ fibroblasts and 274, 615 and 1987 cells of *SPPI*⁺ macrophages in GSE188737, GSE182227, and GSE234933 dataset, respectively.

Table. R1. Information of validation datasets

	PMID	Dataset	Platform	Note
1	29198524[20]	GSE103322	SMART-Seq2	Exclude
2	31924475[11]	GSE139324	10x 3'	Exclude
3	36973261[21]	GSE188737	10x 5'	
4	36973261[21]	GSE225331	Fluidigm C1	Exclude
5	37012457[23]	GSE182227	10x 5'	
6	36928331[22]	HRA003383	10x	Exclude
7	37535729[4]	GSE234933	10x 5'	

Then, we performed the interaction analysis of *POSTN*⁺ fibroblasts and *SPPI*⁺ macrophages by R package CellChat based on above three accessibly public datasets to validate our conclusion (**Fig. R13c-e**). We found that among all fibroblast subtypes, the interaction from *POSTN*⁺ fibroblasts to *SPPI*⁺ macrophages was strongest. Similarly, the interaction from *SPPI*⁺ macrophages to *POSTN*⁺ fibroblasts was superior to other macrophage subtypes. These results in validation datasets are consistent with those in our in-house cohort, which strongly supported our conclusion on *POSTN*⁺ fibroblasts-*SPPI*⁺ macrophages interaction in HNSCC tumor microenvironment.

Further, we also performed the interaction analysis of *POSTN*⁺ fibroblasts and *SPPI*⁺ macrophages by R package NicheNet based on three accessible public validation datasets. We found there were significant intersections between validation datasets and our in-house data for receptors, target genes, and target genes GO and KEGG enrichment pathways (**Fig. R14a-d**). For instance, the overlapping ligands, receptors and target genes of GSE182227 and our in-house data was taken as a representative result for NicheNet analysis, we found top regulatory ligands in *POSTN*⁺ fibroblasts included *HAS2*, *CCL2*, and *IL15*, which bounded to receptors on *SPPI*⁺ macrophages including *CD44*, *CCR1*, *CCR5*, and *IL2RG*, triggering downstream pathways such as NF- κ B signaling pathways and TNF signaling pathways (**Fig. R14e-h**). These regulatory patterns are indeed consistent with findings in our study. We have added these results in the revised manuscript, and added **Fig. R13c-e** as **Supplementary Fig. 4c**, and **Fig. R14** as **Supplementary Fig. 4h-k**.

Fig. R13. Validation of *POSTN*⁺ fibroblasts-*SPPI*⁺ macrophages interaction by R package CellChat based on public datasets. (a-b) Summary of cell types annotated in public validation datasets and in-house cohort. Macrophage subtypes are shown in (a) and fibroblast subtypes are shown in (b). (c-e) Lollipop plots show interaction strength by R package CellChat between different macrophage subtypes and fibroblast

subtypes. Results of GSE188737, GSE182227, and GSE234933 are shown in (c), (d), and (e), respectively. The upper panel represents interaction from different fibroblast subtypes to *SPPI*⁺ macrophages and the lower panel represents interaction from different macrophage subtypes to *POSTN*⁺ fibroblasts.

Fig. R14. Validation of *POSTN*⁺ fibroblasts-*SPPI*⁺ macrophages interaction by R package NicheNet based on public datasets. (a-d) Venn plots show overlapping target genes (a), receptors (b), target genes GO enrichment pathways (c), and target genes KEGG enrichment pathways (d) among GSE188737, GSE182227, GSE234933, and in-house data inferred by NicheNet algorithm. Overlapping ligands (e), target genes (e), receptors (f), target genes GO enrichment pathways (g), and target genes KEGG enrichment pathways (h) between GSE182227 and in-house data from R package NicheNet.

4.2 I would also suggest that they authors contextualise their fibroblast and macrophage subpopulations to currently known subtypes (eg iCAF, myCAF, ApCAF, M1 or M2 macrophages etc).

Response: As suggested by the reviewer, based on the marker genes of currently known subtypes as previous study [24] (marker genes listed in **Supplementary Table 4**), we calculated the signature score of iCAF, myCAF, and apCAF for each fibroblast subtypes. As shown in **Fig. R15a-c**, we found that *POSTN*⁺ fibroblasts had highest myCAF score ($p < 2.2e-16$), *CCL19*⁺ fibroblasts had highest iCAF score followed by *RSPO1*⁺ fibroblasts and *SFRP1*⁺ fibroblasts ($p < 2.2e-16$). However, *SEMA4A*⁺ fibroblasts and *DES*⁺ myofibroblasts are both low-scored in these three signatures, which may be attributed to data heterogeneity.

Then, we calculated M1 and M2 signature score of each macrophage subtype [25] (marker genes of M1 and M2 signature were listed in **Supplementary Table 4**). The results showed that *CXCL10*⁺ macrophages, *SPPI*⁺ macrophages, and *FOLR2*⁺ macrophages were all scored higher in M2 signature than M1 signature while *CIQC*⁺ macrophages got low score in both M1 and M2 signature (**Fig. R15d-e**). We have added these results in the revised manuscript, and added **Fig. R15b** as **Supplementary Fig. 3e**, and **Fig. R15e** as **Supplementary Fig. 3h**.

Fig. R15. Comparison of our fibroblast and macrophage subpopulations to currently known subtypes. Violin and box plots show the scores of iCAF signature (a), myCAF signature (b), apCAF signature (c), M1 signature (d), and M2 signature (e) of our fibroblast and macrophage subpopulations.

4.3 The dataset of 68 patients used to validate their findings has also not been well described- who are these patients, how were they treated? are they a consecutive or highly selected dataset?

Response: As for the clinical information of our validation cohort, it has been disclosed in **Supplementary Table 2**.

5. All the cellular networks data remain circumstantial without validations, and the authors are referred to the many excellent single cell papers described above that they can access and re-analyze to their specific condition and question. I also suggest that when analysing the interactome with tumor cells, these should be individual patient specific analyses rather than trying to combine malignant cells (see point 1). There was some attempt at looking at proximity between *POSTN* fibrioblasts and *SPP1*

macrophages, but I have no idea where this dataset comes from, what was done and how the analysis was conducted. Validation experiment such as these are the cornerstone to accept the circumstantial discovery data provided by expensing scRNAseq, and I do advise the authors to pay attention to these and expand on them, including re-analyses of existing bulk and scRNAseq data as mentioned before.

Response: Thanks for the reviewer's comments. Since P2, P10, and P13 provided their NT, pre, and E stage samples representing tumor progression process, we chose these three patients and analyzed cell-cell interaction based on individual patient respectively by R package CellChat (**Fig. R16a-d**). Apparently, interaction strengths of *POSTN*⁺ fibroblasts to *SPPI*⁺ macrophages, *SPPI*⁺ macrophages to *POSTN*⁺ fibroblasts, *POSTN*⁺ fibroblasts to tumor cells, and *SPPI*⁺ macrophages to tumor cells all increased from NT, to pre and E stages in three patients. Moreover, A-stage individuals were chosen for analyzing interaction between fibroblast subtypes and macrophage subtypes (**Fig. R16e-h**). Results showed that in A-stage individuals, interaction between *POSTN*⁺ fibroblasts and *SPPI*⁺ macrophages were strongest among all fibroblast subtypes and macrophage subtypes, which is consistent with results in **Fig. 4A**. We have added these results in the revised manuscript, and added **Fig. R16a-d** as **Supplementary Fig. 4a**, and **Fig. R16e-h** as **Supplementary Fig. 4b**.

As for our validation cohort, the 68 HNSCC samples are collected from our independent validation cohort, whose clinical information has been included in **Supplementary Table 2**. We have conducted validation experiments and updated these results in **Fig. 2**, **Supplementary Fig. 4**, and **Supplementary Fig. 5**. We hope the revised manuscript would meet the reviewer's standard.

Fig. R16. Validation of cell-cell interaction in individual patients. (a-d) Lollipop plots show interaction strength from $POSTN^+$ fibroblasts to $SPP1^+$ macrophages (a), $SPP1^+$ macrophages to $POSTN^+$ fibroblasts (b), $POSTN^+$ fibroblasts to tumor cells (c), and $SPP1^+$ macrophages to tumor cells (d) in NT, pre, and E stages from P2, P10, and P13. (e-h) Lollipop plots show interaction strength of fibroblast subtypes and macrophage subtypes in A-OSCC1 (e), A-OSCC3 (f), A-OSCC5 (g), and A-OSCC6 (h) samples.

6.1 The analysis of the ENE data has many of the same issues described before, but if these are matched to ENE negative nodes from the same patient, it supports the data even better and these need to be described...there is NO MENTION at all at how many or who the ENE+ and ENE- patients are, and again no attempt at orthogonal validation.

Response: Thanks for the reviewer's comments, we agree with that if ENE⁻ and ENE⁺ samples were matched, the analytic conclusion could have been more significant. However, most of our ENE⁻ and ENE⁺ lymph node samples were not matched except for LN1 and LN2 (from P6). Here, we analyzed LN1 and LN2 samples to validate our conclusions on CD8⁺ T cells. As shown in Figure 5, we observed higher CD8 Tex proportion in LN-out samples than LN-in samples (Fig. 5E) and higher *CXCL13*

expression level in CD8 Tex in LN-out samples (**Fig. 5I**). Here, in the paired LN-in and LN-out samples, we found consistently up-regulated CD8 Tex proportion as well as *CXCL13* expression level in the LN-out sample (**Fig. R17**).

The clinical information of our cohort has been supplemented in **Supplementary Table 1**, including ENE⁻ and ENE⁺ information.

Fig. R17 Comparison of CD8⁺ T cell between paired lymph node samples. (a) Bar chart shows the proportion differences of CD8⁺ T cell subclusters between LN-out and LN-in samples of P6. (b) Violin plot shows difference of *CXCL13* expression in CD8 Tex between LN-out and LN-in samples of P6.

6.2 The data concerning high *CXCL13* Tex is interesting, but I would also like to know if these are present in advanced or recurrent tumors as these can also represent exhaustion over time.

Response: First, we compared Tex proportion between A and R samples and found an increase in CD8 Tex proportion in R samples (**Fig. R18a**). Then, we found that in the A and R samples, *CXCL13* was specifically expressed by CD8 Tex subpopulation (**Fig. R18b**). In the meantime, *CXCL13* expression level was significantly higher in R stage sample (**Fig. R18c**). In summary, these results were consistent with **Figure 5** and showed CD8⁺ T cell exhaustion over time.

Fig. R18. Analysis of CD8⁺ T cell subtypes between A and R samples. (a) Bar chart shows the proportion differences of CD8⁺ T cell subtypes between A and R samples. (b) Feature plot shows *CXCL13* expression level in CD8⁺ T cells in A and R samples. CD8 Tex was circled by the dashed line. (c) Violin plot shows *CXCL13* in CD8 Tex in A and R samples. **** $p < 0.001$.

7. sweeping statements on therapeutics such as targeting interactions or CXCL13 are pointless, inaccurate and even their own data suggest is wrong- these will only target a specific subpopulation in your own dataset, so why target them?!

Response: Actually, *CXCL13*⁺ CD8⁺ T cells are a novel subpopulation of T cells which have been proved to play an important role during immune checkpoint inhibitor therapy and could be a biomarker for therapy response [26-28]. Here, for the first time, we identified a significant increase of *CXCL13*⁺ CD8⁺ T cells during extracapsular lymph node metastasis and their interaction with tumor cells. Based on current studies of *CXCL13*⁺ CD8⁺ T cells, we supposed that it might be important in HNSCC lymph node metastasis and could even be a therapy target. However, in the present, we have no further conclusive evidence to support our assumption. Therefore, according to the reviewer's advice, we have modified related inappropriate description in the revised manuscript.

All in all, this is a landscape paper that would be a valuable resource but has a number of erroneous assumptions and almost zero attempt at validation, which make these findings circumstantial at best.

Response: Thanks for the reviewer's patience and valuable comments. We have made point-to-point response to the reviewer's concerns above and we also supplemented experimental validations of key conclusions. We hope that our revised manuscript could meet the reviewer's standard.

Reviewer3

1. The authors should relate malignant and stromal cell subpopulations they uncover to previously described subpopulations in Puram et al, Cell, 2017 acknowledging of course that differences in scRNA-seq technique may create differences in subpopulations defined.

Response: Thanks for the reviewer's valuable suggestion. We have assessed the similarity of our malignant and stromal cell subpopulations and previously described subpopulations in Puram's work [20] (**Fig. R19**).

For malignant cell subpopulations (maker genes listed in **Supplementary Table 4**), C0 showed high score in epithelial differentiation1 signature while C1, C2, and C3 tend to follow epithelial differentiation pattern 2 ($p < 2.2e-16$). Interestingly, C1 showed high expression in cell cycle signature, indicating its more malignant phenotype than other clusters ($p < 2.2e-16$), which was coincidence with our findings. C4 is mainly composed of epithelial cells from NT samples, so it has no matched malignant cell subtypes in Puram's work.

As for the fibroblast subtypes (maker genes listed in **Supplementary Table 4**) (**Fig. R20**), *POSTN*⁺ fibroblasts and proliferating fibroblasts are similar to CAF1 ($p < 2.2e-16$), normal fibroblasts, *RSPO1*⁺ fibroblasts, *SFRP1*⁺ fibroblasts, and *CCL19*⁺ fibroblasts are tend to be CAF2 ($p < 2.2e-16$). However, *SEMA4A*⁺ fibroblasts and *DES*⁺ myofibroblasts in our data could not find particular counterparts in Puram's subtypes, which may due to the dataset heterogeneity. We have added these results in the revised manuscript, and added **Fig. R19a** as **Supplementary Fig. 2k**, **Fig. R20a** as **Supplementary Fig. 3f**.

Fig. R19. Comparison of our malignant cell subtypes to Puram malignant cell signatures. Violin and box plots show the scores of cell cycle signature (a), pEMT signature (b), epithelial cell differentiation1 signature (c), epithelial cell different2 signature (d), stress signature (e), and hypoxia signature (f) of our malignant cell

subpopulations. (g) Heatmap shows the relative malignant signature scores value of our malignant cell subpopulations.

Fig. R20. Comparison of our fibroblast subtypes to Puram fibroblast subpopulations. Violin and box plots showed the scores of CAF1 signature (a), CAF2 signature (b), and myofib signature (c) of our fibroblast subpopulations. (d) Heatmap shows the relative CAF signature scores value of our fibroblast subpopulations.

2. The authors should better relate malignant and stromal cell pseudotime subpopulation analysis to recently published similar data from Choi et al, Nature Communications, 2023. While the authors do reference Choi et al in the introduction, there needs to be a more specific discussion of why the present study aims to provide novel information and how the results relate to those previously published.

Response: In Choi and his colleagues' work [19], they pooled fibroblasts and epithelial cells together for trajectory analysis, studying the malignant cell-fibroblasts interaction and evolution during HNSCC progression and metastasis. However, in our study, we did not include fibroblast subtypes for pseudotime analysis, but we could still summarize developmental trends of malignant cells during lymph node metastasis (**Fig. R21**). When focusing on epithelial cells during lymph node metastasis (LN-in and LN-out samples), we observed that epithelial cells from intracapsular metastatic lymph nodes (LN-in) located more in S2 and S3 than S1, while extracapsular metastatic epithelial cells (LN-out) located mainly in S3, indicating an increasing malignant characteristic during extracapsular metastasis, which is actually reported for the first time.

Fig. R21. Pseudotime analysis of epithelial cells in NT, pre, E, A, LN-in, LN-out, LN-normal, and R samples. The merged result is showed in (a) and separate results are shown in (b).

3. The authors identify *POSTN*⁺ fibroblasts and *SPP1*⁺ macrophages as key stromal cell types involved in tumor progression, and a primary claim related to poor outcomes is from analysis of TCGA RNA-seq data. These findings should be contextualized with prior research. For example, Luo et al., Nature Communications, 2022 describe an interaction between CAFs and *SPP1*⁺ macrophages using scRNA-seq data. Wenhua et al., Cancer Medicine, 2023 report that overall *POSTN* expression in the TCGA cohort is not associated with survival. In addition, Bie & Zhang, J Immunol Res, 2021 find a similar association between *SPP1* and survival in TCGA.

Response: Thank the reviewer's valuable comments. We have contextualized our results with previous studies in discussion section in revised manuscript (highlighted). The Luo et al. [29] identified the interaction between CAFs and *SPP1*⁺ macrophages, however, they didn't distinguish which subpopulation of CAFs has the stronger interaction with *SPP1*⁺ macrophages, and they only described their results in tumor but did not distinguish tumor stages. The studies of Wenhua et al. [30] and Bie & Zhang [31] only reported the correlation of individual gene expression and survival. In this study, we revealed the dynamic alteration and interaction *POSTN*⁺ fibroblasts and *SPP1*⁺ macrophages during the HNSCC development and progression, which provided an important insight to investigate the roles of dynamic stromal microenvironment in HNSCC development.

Other comments:

1. The authors should provide histologic evidence to validate the identity of each sample analyzed – this is particularly important for pre-cancer samples to validate this designation histopathologically.

Response: Thanks for the reviewer's advice. We have added H&E images to **Supplementary Fig.1a**.

2. It is unclear how the authors designate early (E) vs. advanced (A) tumors. From Table S1, it appears that a T2N1 sample (P13) was listed as early, while a T3N1 sample (P8) was listed as advanced. Based on AJCC staging of oral cancers, both of these patients would be classified as overall stage III, making it unclear what differentiates the two. The analysis should be redone with a more consistent classification.

Response: Thanks for the reviewer's question. Actually we designated the clinical stage based on AJCC (ver.8) instruction. Briefly, clinical I-II stages were classified into early (E) stage and clinical III-IV stages were designated to advanced (A) stage. As for P13, we have made a mistake on his N stage assessment: his lymph node was swollen but showed pathologically negative in the end. We labeled the wrong information in the original table but we have revised clinical information in the latest manuscript. We are sorry for making you confused!

3. How was fibroblast and myeloid cell clustering conducted? Can the authors validate fibroblast and myeloid clusters in the context of published work? For example, Galbo et al., Clinical Cancer Research, 2021 identify 6 pan-cancer CAF subtypes – can the authors validate their own clustering in the context of these subtypes?

Response: According to the reviewer's suggestion, we have calculated pan-cancer CAF signature expression level on our fibroblast subtypes [32] (marker genes listed in **Supplementary Table 4**) (**Fig. R22**). We found that *RSPO1*⁺ fibroblasts, *SFRP1*⁺ fibroblasts, and *CCL19*⁺ fibroblasts showed iCAF phenotypes, *SEMA4A*⁺ fibroblasts showed highest iCAF2 signature score. Proliferating fibroblasts are similar to pCAFs. Notably, *POSTN*⁺ fibroblasts tend to be dCAF ($p < 2.2e-16$), related to desmoplastic component production, which is consistent with results in our own research. We have added these results in the revised manuscript, and added **Fig. R22b** as **Supplementary Fig. 3g**.

Fig. R22. Comparison of our fibroblast subtypes to pan-cancer CAF subtypes. Violin and box plots show the scores of pan_myCAF signature (a), pan_dCAF signature (b), pan_iCAF signature (c), pan_iCAF2 signature (d), and pan_pCAF signature (e) in our fibroblast subpopulations.

4. The immunofluorescence images provided all need to be brighter and clearer in order to enable independent interpretation of their utility in validating the findings of the sequencing analysis.

Response: Sorry for bringing the reviewer the bad experience in reviewing. We have adjusted the brightness and resolution of all immunofluorescence figures in the revised manuscript.

5. Please clarify what CD8-*Tex* cells are and how these are identified.

Response: The CD8_*Tex* means exhausted CD8⁺ T cells and their markers have been shown in the dot plot in **Supplementary Fig. 3j**.

6. The authors provide a variety of ligand-receptor pairs across CD8 *Tex* and tumor cells suggesting bidirectional signaling that is upregulated during extranodal extension. However, they do not provide any validation of these findings, making it unclear whether these are just an artifact of multiple hypothesis testing that results from analysis of large scale sequencing data or true biologically meaningful results. In the absence of additional validation, the authors should tone down their claims, particularly related to tumor cell reprogramming.

Response: Thank you for the reviewer's comments, as for the validation of results in lymph node metastasis section, actually we turned to our validation cohort 3 to analyze pERK expression level between ENE⁻ and ENE⁺ samples (**Fig. R8, please refer to the response of the Reviewer #1's comment 9**). We found that pERK⁺ epithelial ratio is higher in ENE⁺ samples, which indicated that tumor cells in ENE⁺ lymph nodes showed higher ERK signaling activation level. What's more, correlation analysis showed pERK⁺ epithelial ratio is positively correlated with *CXCL13*⁺ *Tex* ratio, which suggested that *CXCL13*⁺ *Tex* may induce ERK pathway activation during lymph node metastasis process.

7. The authors provide an analysis of differentially expressed genes in recurrent vs. advanced stage cancers and relate their analyses to potential therapeutic targets. However, given the small number of samples, these results should be validated using previously published data or functional experiments.

Response: We have validated the results of **Fig. 6** in two public datasets in scRNA-seq and bulk RNA-seq to obtain the consistent results (**Please refer to the response of the Reviewer #1's comment 10**).

Reference:

1. Sung H, Ferlay J, Siegel RL, Laversanne M, Soerjomataram I, Jemal A, et al. Global Cancer Statistics 2020: GLOBOCAN Estimates of Incidence and Mortality Worldwide for 36 Cancers in 185 Countries. *CA Cancer J Clin.* 2021;71(3):209-49.
2. Abba MC, Fabris VT, Hu Y, Kittrell FS, Cai W-W, Donehower LA, et al. Identification of novel amplification gene targets in mouse and human breast cancer at a syntenic cluster mapping to mouse ch8A1 and human ch13q34. *Cancer Res.* 2007;67(9):4104-12.
3. Suhail Y, Maziarz JD, Novin A, Dighe A, Afzal J, Wagner G, et al. Tracing the cis-regulatory changes underlying the endometrial control of placental invasion. *Proc Natl Acad Sci U S A.* 2022;119(6).
4. Bill R, Wirapati P, Messemaker M, Roh W, Zitti B, Duval F, et al. CXCL9:SPP1 macrophage polarity identifies a network of cellular programs that control human cancers. *Science.* 2023;381(6657):515-24.
5. Weber P, Künstner A, Hess J, Unger K, Marschner S, Idel C, et al. Therapy-Related Transcriptional Subtypes in Matched Primary and Recurrent Head and Neck Cancer. *Clin Cancer Res.* 2022;28(5):1038-52.
6. Sun Y, Wu L, Zhong Y, Zhou K, Hou Y, Wang Z, et al. Single-cell landscape of the ecosystem in early-relapse hepatocellular carcinoma. *Cell.* 2021;184(2).
7. Cao J, Spielmann M, Qiu X, Huang X, Ibrahim DM, Hill AJ, et al. The single-cell transcriptional landscape of mammalian organogenesis. *Nature.* 2019;566(7745):496-502.
8. Li J, Wang R, Zhou X, Wang W, Gao S, Mao Y, et al. Genomic and transcriptomic profiling of carcinogenesis in patients with familial adenomatous polyposis. *Gut.* 2020;69(7):1283-93.
9. Kim N, Kim HK, Lee K, Hong Y, Cho JH, Choi JW, et al. Single-cell RNA sequencing demonstrates the molecular and cellular reprogramming of metastatic lung adenocarcinoma. *Nat Commun.* 2020;11(1):2285.
10. Lee H-O, Hong Y, Etioglu HE, Cho YB, Pomella V, Van den Bosch B, et al. Lineage-dependent gene expression programs influence the immune landscape of colorectal cancer. *Nat Genet.* 2020;52(6):594-603.
11. Cillo AR, Kürten CHL, Tabib T, Qi Z, Onkar S, Wang T, et al. Immune Landscape of Viral- and Carcinogen-Driven Head and Neck Cancer. *Immunity.* 2020;52(1).
12. Pu W, Shi X, Yu P, Zhang M, Liu Z, Tan L, et al. Single-cell transcriptomic analysis of the tumor ecosystems underlying initiation and progression of papillary thyroid carcinoma. *Nat Commun.* 2021;12(1):6058.
13. Cui Zhou D, Jayasinghe RG, Chen S, Herndon JM, Iglesia MD, Navale P, et al. Spatially restricted drivers and transitional cell populations cooperate with the microenvironment in untreated and chemo-resistant pancreatic cancer. *Nat Genet.* 2022;54(9):1390-405.
14. Huang H, Wu L, Lu L, Zhang Z, Qiu B, Mo J, et al. Single-cell transcriptomics uncovers cellular architecture and developmental trajectories in hepatoblastoma. *Hepatology.* 2023;77(6):1911-28.
15. Wang Z, Wang Q, Chen C, Zhao X, Wang H, Xu L, et al. NNMT enriches for AQP5+ cancer stem cells to drive malignant progression in early gastric cardia adenocarcinoma. *Gut.* 2023.
16. Liu C, Zhang M, Yan X, Ni Y, Gong Y, Wang C, et al. Single-cell dissection of cellular and molecular features underlying human cervical squamous cell carcinoma initiation and progression. *Sci Adv.* 2023;9(4):eadd8977.
17. Jiang Y, Yang J, Liang R, Zan X, Fan R, Shan B, et al. Single-cell RNA sequencing highlights intratumor heterogeneity and intercellular network featured in adamantinomatous craniopharyngioma.

Sci Adv. 2023;9(15):eadc8933.

18. Zhang S, Fang W, Zhou S, Zhu D, Chen R, Gao X, et al. Single cell transcriptomic analyses implicate an immunosuppressive tumor microenvironment in pancreatic cancer liver metastasis. *Nat Commun.* 2023;14(1):5123.
19. Choi J-H, Lee B-S, Jang JY, Lee YS, Kim HJ, Roh J, et al. Single-cell transcriptome profiling of the stepwise progression of head and neck cancer. *Nat Commun.* 2023;14(1):1055.
20. Puram SV, Tirosh I, Parikh AS, Patel AP, Yizhak K, Gillespie S, et al. Single-Cell Transcriptomic Analysis of Primary and Metastatic Tumor Ecosystems in Head and Neck Cancer. *Cell.* 2017;171(7).
21. Quah HS, Cao EY, Suteja L, Li CH, Leong HS, Chong FT, et al. Single cell analysis in head and neck cancer reveals potential immune evasion mechanisms during early metastasis. *Nat Commun.* 2023;14(1):1680.
22. Zhang Y, Liu G, Tao M, Ning H, Guo W, Yin G, et al. Integrated transcriptome study of the tumor microenvironment for treatment response prediction in male predominant hypopharyngeal carcinoma. *Nat Commun.* 2023;14(1):1466.
23. Puram SV, Mints M, Pal A, Qi Z, Reeb A, Gelev K, et al. Cellular states are coupled to genomic and viral heterogeneity in HPV-related oropharyngeal carcinoma. *Nat Genet.* 2023;55(4):640-50.
24. Elyada E, Bolisetty M, Laise P, Flynn WF, Courtois ET, Burkhardt RA, et al. Cross-Species Single-Cell Analysis of Pancreatic Ductal Adenocarcinoma Reveals Antigen-Presenting Cancer-Associated Fibroblasts. *Cancer Discov.* 2019;9(8):1102-23.
25. Cheng S, Li Z, Gao R, Xing B, Gao Y, Yang Y, et al. A pan-cancer single-cell transcriptional atlas of tumor infiltrating myeloid cells. *Cell.* 2021;184(3).
26. Liu B, Zhang Y, Wang D, Hu X, Zhang Z. Single-cell meta-analyses reveal responses of tumor-reactive CXCL13+ T cells to immune-checkpoint blockade. *Nat Cancer.* 2022;3(9):1123-36.
27. Bassez A, Vos H, Van Dyck L, Floris G, Arijs I, Desmedt C, et al. A single-cell map of intratumoral changes during anti-PD1 treatment of patients with breast cancer. *Nat Med.* 2021;27(5):820-32.
28. Zhang Y, Chen H, Mo H, Hu X, Gao R, Zhao Y, et al. Single-cell analyses reveal key immune cell subsets associated with response to PD-L1 blockade in triple-negative breast cancer. *Cancer Cell.* 2021;39(12).
29. Luo H, Xia X, Huang L-B, An H, Cao M, Kim GD, et al. Pan-cancer single-cell analysis reveals the heterogeneity and plasticity of cancer-associated fibroblasts in the tumor microenvironment. *Nat Commun.* 2022;13(1):6619.
30. Wenhua S, Tsunematsu T, Umeda M, Tawara H, Fujiwara N, Mouri Y, et al. Cancer cell-derived novel periostin isoform promotes invasion in head and neck squamous cell carcinoma. *Cancer Med.* 2023;12(7):8510-25.
31. Bie T, Zhang X. Higher Expression of SPP1 Predicts Poorer Survival Outcomes in Head and Neck Cancer. *J Immunol Res.* 2021;2021:8569575.
32. Galbo PM, Zang X, Zheng D. Molecular Features of Cancer-associated Fibroblast Subtypes and their Implication on Cancer Pathogenesis, Prognosis, and Immunotherapy Resistance. *Clin Cancer Res.* 2021;27(9):2636-47.

Reviewers' Comments:

Reviewer #1:

Remarks to the Author:

The authors have addressed most of my questions and concerns to improve the quality and impact of this manuscript. However, the analysis of survival data for TCGA-HNSC as a validation cohort remains a weakness of the study. Although the authors have included experimental data from immunostaining of tissue samples and in vitro cell culture, the main conclusions are primarily based on in silico bioinformatics analysis, and the lack of compelling experimental evidence for causality and underlying mechanism of key findings limits the enthusiasm for this manuscript. Before publishing this study, the following issues must be addressed:

1. Clinical variables selected for multivariate Cox regression models, in particular age and sex, are not the most relevant prognostic risk factors for the TCGA-HNSC cohort. The authors need to include other variables with a significant impact on overall survival, such as HPV status or resection margin, for the analysis shown in Suppl. Fig. 2i and 2t, Suppl. Fig. 3c-d and Suppl. Fig. 4d.
2. Suppl. Fig. 2h indicates a median selection for cluster 4, but the number of cases is different (high n=77 and low n=417) – please correct.
3. Fig. 2L shows experimental data with TFDP1 overexpression in SCC9 cells and silencing in Cal27 cells. Please provide adequate controls for overexpression and gene silencing by Western blot analysis. Did TFDP1 overexpression or silencing have any effect on the survival or proliferation of the transfected cells compared to mock controls, which could confound the migration and invasion data presented?

Reviewer #2:

Remarks to the Author:

Thank you for the response and revisions. I am grateful to the authors attempt to orthogonally validate their findings in the numerous head and neck scRNAseq databases currently available. These would undoubtedly strengthen the findings of this manuscript. For the tumor samples, apart from showing the individual/patient level trajectory plots, I feel that the authors should admit to the limitations of the combined plots for tumor cells. Apart from these, I feel that the authors have satisfactorily responded to most of the reviewers queries. Can I also request that the authors not only deposit the scRNAseq data but also the metadata with the clinicopathologi/demographic details for each patient.

Reviewer #3:

Remarks to the Author:

While the authors address some of my concerns, there are others that are not adequately addressed. Please see below for my point-by-point response to the authors' responses.

Overall, the biggest concern remains the novelty of the present work. The authors state, in the introduction, that "Choi et al explored..." Given the strong similarities in methodology between Choi et al and the present work, the authors should clearly define within the introduction what they aim to explore and how they expect their methodology to produce different or novel results. While the authors do acknowledge ENE and recurrence as two areas of interest, these are not the primary focus of the paper.

In addition, there are grammatical errors throughout the manuscript that need to be corrected prior to consideration for publication.

Major Comments:

1. While the authors perform scoring for the malignant and fibroblast cell subpopulations in Puram et al, the interpretation provided is not adequate.

a. Regarding C1, a high score for cell cycle markers does not necessarily indicate a “more malignant” phenotype and should not be described as such. The authors’ finding that CibersortX deconvolution of TCGA RNA-seq data suggests an association between C1 and survival should be discussed further – is it simply that more proliferative tumors have a worse prognosis? Has this been shown previously? This should either be contextualized or removed.

b. C2 appears highest in p-EMT and may indicate this subset of cells, rather than an epithelial differentiation population. Puram et al showed an association between p-EMT and clinicopathologic outcomes in TCGA data – does the same association hold with subpopulation C2?

c. Fibroblast subtypes do not appear to cluster cleanly with CAF1, CAF2, and myofib, which is not surprising given differences in sequencing techniques used. However, the authors should still attempt to better categorize the CAF subpopulations beyond just descriptions based on single highly expressed genes.

2. While I appreciate the technical difference between including and excluding fibroblast subtypes from pseudotime analysis, this work still lacks a specific discussion of why it aims to utilize this subtle difference in methodology to investigate a novel question and why this difference in methodology is meaningful.

a. The results should also more closely be related to those of Choi et al given the similarities in methodology. Without this explicit context, it is unclear whether this is merely a confirmatory paper, or one with a novel approach/findings.

b. Figure R21 is not adequately labeled for the reviewer to draw independent conclusions. It appears to show that cells in the LN-in sample are primarily along S2 and S3, while cells in LN-out are primarily along S1 and S3 – how do the authors reconcile this with cells in the LN-normal sample also being along S1 and S3?

3. Thank you.

Other Comments:

1. Thank you.

2. Thank you.

3. In this case, what do the authors view as key differences between the multiple fibroblast clusters that correspond to iCAF? Are these all subtypes of inflammatory CAFs? Do they all express markers that suggest an interaction with the immune system? Should they better be represented as a single cluster? In particular, it is unclear to the reviewer from Figure 3A that RSPO1+ and SFRP1+ fibroblasts should be considered 2 separate clusters. In addition, the authors do not address my question about myeloid cell clustering – how was the number of clusters decided, and what is the significance of the different macrophage and dendritic cell clusters?

4. Despite some improvement, the images are still too dark and low power to independently interpret. For example, in Figure 3K, in the top right panel, there appear to be a lot more POSTN+ cells; however, it is unclear whether this is just a stromal rich area or whether there is actually epithelial tumor present within this section. An alternate, clearer representative section should be chosen, or corresponding H&E sections should be provided for confirmation that tumor is present.

5. Thank you.

6. Thank you.

7. Thank you.

Response Letter

Reviewer1

The authors have addressed most of my questions and concerns to improve the quality and impact of this manuscript. However, the analysis of survival data for TCGA-HNSC as a validation cohort remains a weakness of the study. Although the authors have included experimental data from immunostaining of tissue samples and in vitro cell culture, the main conclusions are primarily based on in silico bioinformatics analysis, and the lack of compelling experimental evidence for causality and underlying mechanism of key findings limits the enthusiasm for this manuscript. Before publishing this study, the following issues must be addressed.

Response: Thanks the reviewer for the careful evaluation and we have performed additional analysis to the reviewer's concern.

1. Clinical variables selected for multivariate Cox regression models, in particular age and sex, are not the most relevant prognostic risk factors for the TCGA-HNSC cohort. The authors need to include other variables with a significant impact on overall survival, such as HPV status or resection margin, for the analysis shown in Suppl. Fig. 2i and 2t, Suppl. Fig. 3c-d and Suppl. Fig. 4d.

Response: Thank the reviewer for the suggestion. We have replaced gender and age factors with surgical margin status and HPV status in the multivariate Cox regression model analysis and found that C1 of malignant epithelial cells, *POSTN*⁺ fibroblasts, *SPPI*⁺ macrophages, *TFDPI*, and the interaction signature score of *POSTN*⁺ fibroblasts and *SPPI*⁺ macrophages are independent predictors for HNSCC patients (**Figure for Reviewer [Fig. R] 1**). Due to the fact that only 77 individuals in the TCGA-HNSCC cohort have concurrent information on HPV, margin status, and survival information, we have included individuals without HPV information into the multivariate regression analysis, designating them as the 'Other' group. We added these results in the revised manuscript, and added **Fig. R1a** as **Supplementary Fig. 2i**, **Fig. R1b** as **Supplementary Fig. 2t**, **Fig. R1c-d** as **Supplementary Fig. 3c-d**, and **Fig. R1e** as **Supplementary Fig. 4d**.

Fig. R1. C1 of malignant epithelial cells, *POSTN*⁺ fibroblasts, *SPPI*⁺ macrophages, *TFDP1*, and the interaction signature score of *POSTN*⁺ fibroblasts and *SPPI*⁺ macrophages are independent predictors. Multivariate Cox regression model analysis, which included the factors of surgical margin status, TNM status, HPV status and the deconvoluted score of C1 of malignant epithelial cells (a), expression level of *TFDP1* (b), the top 50 signature score of *POSTN*⁺ fibroblasts (c), the top 50 signature score of *SPPI*⁺ macrophages (d), and the interaction signature score of *POSTN*⁺ fibroblasts and *SPPI*⁺ macrophages (e) in the TCGA-HNSCC data.

2. Suppl. Fig. 2h indicates a median selection for cluster 4, but the number of cases is different (high n=77 and low n=417) – please correct.

Response: Based on the CIBERSORTx algorithm, we calculated the proportion of each single cell type within the TCGA-HNSCC samples. Among the 494 TCGA-HNSCC samples used for survival analysis, there are 417 samples where the proportion of

cluster 4 is zero, and 77 samples where it is not zero. Therefore, there is an inconsistency in the number of samples for cluster 4 grouping.

3. Fig. 2L shows experimental data with TFDP1 overexpression in SCC9 cells and silencing in Cal27 cells. Please provide adequate controls for overexpression and gene silencing by Western blot analysis. Did TFDP1 overexpression or silencing have any effect on the survival or proliferation of the transfected cells compared to mock controls, which could confound the migration and invasion data presented?

Response: We have performed Western blot analysis to confirm the overexpression (OE) and gene knockdown (KD) efficiency of TFDP1 in SCC9 and Cal27 cell lines (Fig. R2). As for the proliferation capability, actually we have done CCK8 assays on cells with TFDP1 OE/KD and found no significant differences between OE/KD group and the control group (Fig. R3).

Fig. R2. Western blot analysis of TFDP1 of Cal27 and SCC9 cell lines with TFDP1 overexpression and knockdown.

Fig. R3. Relative viability evaluated by CCK8 assay of Cal27 (a) and SCC9 (b) cell lines with TFDP1 overexpression and knockdown.

Reviewer2

1. Thank you for the response and revisions. I am grateful to the authors attempt to orthogonally validate their findings in the numerous head and neck scRNA-seq databases currently available. These would undoubtedly strengthen the findings of this manuscript. For the tumor samples, apart from showing the individual/patient level trajectory plots, I feel that the authors should admit to the limitations of the combined plots for tumor cells. Apart from these, I feel that the authors have satisfactorily responded to most of the reviewers queries. Can I also request that the authors not only deposit the scRNAseq data but also the metadata with the clinicopathologi/demographic details for each patient.

Response: Thank the reviewer for his patience and valuable suggestions. We have added limitation description to the discussion section (highlighted). In the meantime, we have uploaded matrix data on Mendeley database (<https://data.mendeley.com/>) with the metadata sheet. As long as the article is published, we will release the data with the accession ID.

Reviewer3

While the authors address some of my concerns, there are others that are not adequately addressed. Please see below for my point-by-point response to the authors' responses.

Overall, the biggest concern remains the novelty of the present work. The authors state, in the introduction, that "Choi et al explored..." Given the strong similarities in methodology between Choi et al and the present work, the authors should clearly define within the introduction what they aim to explore and how they expect their methodology to produce different or novel results. While the authors do acknowledge ENE and recurrence as two areas of interest, these are not the primary focus of the paper.

In addition, there are grammatical errors throughout the manuscript that need to be corrected prior to consideration for publication.

Response: Thank the reviewer for his valuable comments. As for the novelty of the present work, we have added related discussion in the introduction and discussion section (highlighted). As for the grammatical errors, we have revised the manuscript thoroughly for additional English language editing.

Major Comments:

1. While the authors perform scoring for the malignant and fibroblast cell subpopulations in Puram et al, the interpretation provided is not adequate.

1.1 Regarding C1, a high score for cell cycle markers does not necessarily indicate a "more malignant" phenotype and should not be described as such. The authors' finding that CibersortX deconvolution of TCGA RNA-seq data suggests an association between C1 and survival should be discussed further – is it simply that more proliferative tumors have a worse prognosis? Has this been shown previously? This should either be contextualized or removed.

Response: We admit that it is inappropriate to claim "C1 showed high expression in cell cycle signature, indicating its more malignant phenotype than other clusters" in the last response letter since a higher proliferation pattern does not necessarily mean a more malignant phenotype and lead to poorer outcomes. Here, we constructed the CellCycle signature based on the gene module in Puram's work [1] and we found that patients with higher expression of CellCycle signature in the TCGA-HNSCC cohort showed poorer OS (**Fig. R4a**). This result indicated that although poor outcomes were not only caused by cell proliferation, high proliferation phenotype was indeed associated with poor outcomes. Moreover, we evaluated several malignant phenotype scores among the five clusters and observed that C1 owned highest scores, such as oxidative phosphorylation, E2F_targets, and MYC_targets (**Fig. R4b**), supporting our conclusion that C1 showed higher malignant phenotypes than other clusters.

Fig. R4. Analysis of biological characteristics of C1. (a) Kaplan-Meier curves show overall survival in CellCycle-high and –low in TCGA-HNSCC cohort. $P < 0.05$ in the two-sided log-rank test was considered statistically significant. (b) Violin and box plots show the score of different signatures of malignant cell subpopulations.

1.2 C2 appears highest in p-EMT and may indicate this subset of cells, rather than an epithelial differentiation population. Puram et al showed an association between p-EMT and clinicopathologic outcomes in TCGA data – does the same association hold with subpopulation C2?

Response: In order to analyze the association between clinical outcomes and specific clusters, we revisited Puram’s work [1] and found that they used in-house data rather than public TCGA-HNSCC cohort when comparing the clinical characteristics between p-EMT -high and –low groups. However, there are only 9 patients with related clinical information (e.g. stage) in our cohort (**Supplementary Table 1**), which is not sufficient for statistical analysis. Moreover, although p-EMT subcluster could be an interesting topic, it was not our focus in the current study.

1.3 Fibroblast subtypes do not appear to cluster cleanly with CAF1, CAF2, and myofib, which is not surprising given differences in sequencing techniques used. However, the authors should still attempt to better categorize the CAF subpopulations beyond just descriptions based on single highly expressed genes.

Response: Firstly, we mapped the top 50 gene signature of our fibroblast subpopulations on Puram’s fibroblast UMAP plots [1] (**Fig.R5a**). We found that *POSTN*⁺ fibroblasts signature was correspondent to CAF1 specifically and *SFRP1*⁺ fibroblasts were almost coincident with CAF2. Regarding *RSPO1*⁺ fibroblasts, only a subset of cells within CAF2 display the features of this subpopulation. However, other subclusters could not be matched, which may due to the differences in sequencing techniques or patients heterogeneity.

In order to better characterize CAF subpopulations in our data, we performed GO analysis on their marker genes and found these subpopulations exhibited different functions (**Fig.R5b**): fibroblasts were associated with cell matrix adhesion and fiber organization, which referred to physical functions of fibroblasts; *RSPO1*⁺ fibroblasts were associated with mesenchymal cell differentiation and bone mineralization, which

may promote mesenchymal differentiation and bone formation in the oral cavity; *POSTN*⁺ fibroblasts were associated with extracellular matrix organization and macrophage migration, which excluded T cell infiltration while recruited pro-tumor macrophages; *SFRP1*⁺ fibroblasts were associated with neutrophil activation and inflammatory response, showing an inflammatory-inducing function; *SEMA4A*⁺ fibroblasts were associated with hypoxia response, which could be a stressed cluster in the tumor microenvironment; *CCL19*⁺ fibroblasts were associated with leukocyte migration, which may enhance lymphocyte infiltration in the TME; DES_myofibroblasts were associated with muscle cell differentiation and muscle fiber development, just as physical function of myofibroblasts; and proliferating fibroblasts were a proliferating cluster, enriching in cell cycle-related pathways. Consequently, rather than generally categorizing subpopulations as CAF1, CAF2, and myofibroblast, the identification of clusters by marker genes enables a more precise and targeted study of these distinct groups.

Fig. R5. Analysis of fibroblast subpopulations. (a) Feature plots show expression of different fibroblast subpopulations Top 50 signatures of our data on fibroblasts in Puram’s work. (b) Bubble plot of GO analysis for differentially expressed genes in fibroblast subpopulations.

2. While I appreciate the technical difference between including and excluding fibroblast subtypes from pseudotime analysis, this work still lacks a specific discussion of why it aims to utilize this subtle difference in methodology to investigate a novel question and why this difference in methodology is meaningful.

Response: Choi’s research focused on the relationship between epithelial cells and fibroblasts in the tumor progression process, hence he combined fibroblasts and epithelial cells for trajectory analysis. However, our study focused on epithelial cells

and we primarily wanted to figure out the differentiation trajectory within tumor cells. Therefore, we did not include fibroblasts with epithelial cells in the pseudotime analysis. Our method for analyzing the impact of different cell types on each other in the entire tumor microenvironment to promote tumor progression is detailed cell-cell interaction analysis. Through this analysis, we have discovered the reprogramming role of cellular interaction between *SPPI*⁺ macrophages and *POSTN*⁺ fibroblasts and tumor cells, thus reshaping the desmoplastic tumor microenvironment.

2.1 The results should also more closely be related to those of Choi et al given the similarities in methodology. Without this explicit context, it is unclear whether this is merely a confirmatory paper, or one with a novel approach/findings.

Response: We have added comparison of Choi's work with our study and related discussion about the novelty of our work in the introduction and discussion section (highlighted).

2.2 Figure R21 is not adequately labeled for the reviewer to draw independent conclusions. It appears to show that cells in the LN-in sample are primarily along S2 and S3, while cells in LN-out are primarily along S1 and S3 – how do the authors reconcile this with cells in the LN-normal sample also being along S1 and S3?

Response: We generated density plots of pseudotime for cells from LN-out and LN-normal groups. As shown in the **Fig. R6**, although cells from LN-normal and LN-out all distributed in S3, there are significant differences in the distribution of pseudotime of cells from the two groups. To be more specific, at the end of S3, cells from the LN-out group showed significantly higher pseudotime density than cells from the LN-normal group, which indicated that epithelial cells in LN-out exhibited a more malignant pattern than those in LN-normal samples.

Fig. R6. The density plot of the pseudotime of cells from LN-out and LN-normal samples.

Other Comments:

3.1 In this case, what do the authors view as key differences between the multiple fibroblast clusters that correspond to iCAF?

Response: According to Galbo's study, the marker genes of iCAFs included *CFD*, *C3*, *CXCL14*, and *CXCL12*, who were enriched in inflammatory functions by GO analysis [2]. And in our study, we found that *RSPO1*⁺ fibroblasts, *SFRP1*⁺ fibroblasts, and *CCL19*⁺ fibroblasts showed high iCAF scores. However, these subpopulations are different from the following perspectives. First, as showed in **Fig. 3C** and **Supplementary Fig. 3a**, with the progression of HNSCC, the cell ratios of these clusters changed in different patterns. Specifically, the proportion of *RSPO1*⁺ fibroblasts gradually decreased during HNSCC progression, indicating that this subpopulation may occur more in the normal tissues. Second, we compared differentially expressed genes (DEGs) among these three subpopulations and found that the top 2 marker gene of *SFRP1*⁺ fibroblasts, *PLA2G2A* and *CFD* ranked top in the marker genes of iCAFs. In contrast, marker genes of *RSPO1*⁺ fibroblasts and *CCL19*⁺ fibroblasts did not showed in the top genes of iCAFs. Additionally, the functional annotation showed that *RSPO1*⁺ fibroblasts were associated with physical development (*e.g.* Regulation of lymphocyte proliferation, mesenchymal cell differentiation, and Bone mineralization), *SFRP1*⁺ fibroblasts were associated with inflammatory response, and *CCL19*⁺ fibroblasts were associated with leukocyte migration (**Fig. R7**).

In summary, we claimed that although *RSPO1*⁺ fibroblasts, *SFRP1*⁺ fibroblasts, and *CCL19*⁺ fibroblasts all showed relatively high iCAFs scores, the biological functions of these clusters were not all related to the inflammatory response, which in turn supported the advantage of our method that to define subpopulations with marker genes.

Fig. R7. Bubble plot of GO analysis for differentially expressed genes in *RSPO1*⁺ fibroblasts, *SFRP1*⁺ fibroblasts, and *CCL19*⁺ fibroblasts.

3.2 Are these all subtypes of inflammatory CAFs?

Response: They are not all subtypes of iCAFs.

3.3 Do they all express markers that suggest an interaction with the immune system? Should they better be represented as a single cluster?

Response: As shown in **Fig. R7**, *RSPO1*⁺ fibroblasts were related to lymphocyte proliferation, *SFRP1*⁺ fibroblasts were related to neutrophil activation, and *CCL19*⁺ fibroblasts were related to leukocyte chemotaxis. Therefore, we hypothesize that they may have interactions with other immune cells. Then, we calculated interaction weight between *RSPO1*⁺/*SFRP1*⁺/*CCL19*⁺ fibroblasts and other immune cells and found that although they all interact with immune cells, the interaction strengths are diversified (**Fig. R8**). For example, the interaction strengths of CD8⁺ T cell to them rank as: *CCL19*⁺ fibroblasts > *RSPO1*⁺ fibroblasts > *SFRP1*⁺ fibroblasts, while the interaction strengths of B cells to them rank as: *SFRP1*⁺ fibroblasts > *CCL19*⁺ fibroblasts > *RSPO1*⁺ fibroblasts, showing differences between these subpopulations.

Fig. R8. Lollipop plot shows the interaction strengths by R package CellChat between *RSPO1*⁺/*SFRP1*⁺/*CCL19*⁺ fibroblasts and other immune cells.

3.4 In particular, it is unclear to the reviewer from Figure 3A that *RSPO1*⁺ and *SFRP1*⁺ fibroblasts should be considered 2 separate clusters.

Response: At the resolution of 0.5 used for our cluster identification, *RSPO1*⁺ fibroblasts and *SFRP1*⁺ fibroblasts emerged as two distinct groups: their interaction strengths with various immune cells differ significantly, as depicted in **Fig. R8**. Moreover, GO analysis of their marker genes revealed distinct functional profiles for each, further emphasizing their unique roles, as show in **Fig. R9**. To be more specific, *RSPO1*⁺ fibroblasts were associated with regulation of lymphocyte proliferation, mesenchymal cell differentiation, and bone mineralization while *SFRP1*⁺ fibroblasts were associated with inflammatory response. Therefore, these two subpopulations should not be identified as one single cluster.

Fig. R9. Bubble plot of GO analysis for differentially expressed genes in *RSP01*⁺ fibroblasts and *SFRP1*⁺ fibroblasts.

3.5 In addition, the authors do not address my question about myeloid cell clustering – how was the number of clusters decided, and what is the significance of the different macrophage and dendritic cell clusters?

Response: For fibroblasts, we initially identified the clusters as belonging to the general categories of fibroblasts using marker genes such as *COL1A1*, *ACTA2*, and *PDGFRB*. Then, by setting the resolution to 0.5, we further classified the subgroups based on each cluster's markers, such as *POSTN* in *POSTN*⁺ fibroblasts.

As for myeloid cells, we first determined that a cluster belonged to the myeloid cell category using marker genes like *CD68*. After setting the resolution to 0.5, we further classified the subgroups based on each cluster's markers. For distinguishing macrophages and dendritic cells (DCs), we mainly used characteristic markers of DCs such as *CDC1*, *CLEC9A*, and *LILRA4*; while for macrophages, markers like *SPP1*, *FOLR2* were used to define the specific subgroups.

4. Despite some improvement, the images are still too dark and low power to independently interpret. For example, in Figure 3K, in the top right panel, there appear to be a lot more *POSTN*⁺ cells; however, it is unclear whether this is just a stromal rich area or whether there is actually epithelial tumor present within this section. An alternate, clearer representative section should be chosen, or corresponding H&E sections should be provided for confirmation that tumor is present.

Response: We are sorry for the dark image in **Fig. 3** and we have thoroughly upregulated the brightness and replaced the representative images of **Fig. 3K** (shown

below as Fig. R10).

Fig. R10. Representative images of multiplex immunohistochemistry (mIHC) staining of POSTN⁺ fibroblasts (POSTN⁺ α-SMA⁺ double positive) and SPP1⁺ macrophages (SPP1⁺ CD68⁺ double positive) in HNSCC tumor and nonmalignant samples. Scale bar = 50 µm. The quantitative results are shown on the right. Data represent mean ± SD. *p* values were calculated by Kruskal-Wallis test.

Reference:

1. Puram SV, Tirosh I, Parikh AS, Patel AP, Yizhak K, Gillespie S, et al. Single-Cell Transcriptomic Analysis of Primary and Metastatic Tumor Ecosystems in Head and Neck Cancer. *Cell*. 2017;171(7).
2. Galbo PM, Zang X, Zheng D. Molecular Features of Cancer-associated Fibroblast Subtypes and their Implication on Cancer Pathogenesis, Prognosis, and Immunotherapy Resistance. *Clin Cancer Res*. 2021;27(9):2636-47.

Reviewers' Comments:

Reviewer #1:

Remarks to the Author:

The authors responded satisfactorily to points 2 and 3 of my concerns. However, regarding point 1, the number of HPV-positive tumors (n=17) for TCGA-HNSC is underestimated in the multivariate Cox regression model analysis. In the past, several studies have used viral transcript reads from RNA-seq data to infer a more appropriate number of HPV-positive tumors for this cohort (e.g., PMID: 27339696, PMID: 36774364).

Authors must also check the reference list and proper citation in the main text. For example, on page 3 lines 91-94 and pages 12-13 lines 502-507, Choi et al. does not match the reference [8].

Reviewer #3:

Remarks to the Author:

I thank the authors for their revisions and clarifications. The authors have sufficiently addressed my concerns regarding the comparison between Choi et al and the present work through the added text in the introduction and discussion. This text significantly strengthens the work by placing it in context of published literature. I appreciate the additional clarifications regarding methodology and interpretation of results.

Minor comments:

1) There is a grammatical error in line 513: "...were stratifies..."

2) In line 522, the authors state, "...uncovered the underlying mechanism of malignant cell-mediated tumor relapse, and contributed to precise therapeutics for patients with primary and recurrent tumors according to the distinct treatment target selections. This conclusion is overstated and should be toned down.

Response Letter

Re: NCOMMS-23-19404C

Reviewer1

1. The authors responded satisfactorily to points 2 and 3 of my concerns. However, regarding point 1, the number of HPV-positive tumors (n=17) for TCGA-HNSC is underestimated in the multivariate Cox regression model analysis. In the past, several studies have used viral transcript reads from RNA-seq data to infer a more appropriate number of HPV-positive tumors for this cohort (e.g., PMID: 27339696, PMID: 36774364).

Response: Thank the reviewer for the comments and we have performed additional analysis to the reviewer's concern. We used HPV positive samples in TCGA-HNSC cohort identified by Song Cao *et al.* [1] and by Abdurrahman Elbasir *et al.* [2] for multivariate Cox regression analysis (**Figure for Reviewer [Fig. R] 1**). We have replaced previous figures with new ones and used **Fig. R1a** as **Supplementary Fig. 2i**, **Fig. R1b** as **Supplementary Fig. 2t**, **Fig. R1c-d** as **Supplementary Fig. 3c-d**, and **Fig. R1e** as **Supplementary Fig. 4d**.

Fig. R1. C1 of malignant epithelial cells, *TFDPI*, *POSTN*⁺ fibroblasts, *SPPI*⁺ macrophages, and the interaction signature score of *POSTN*⁺ fibroblasts and *SPPI*⁺ macrophages are independent predictors. Multivariate Cox regression model analysis, which included the factors of surgical margin status, TNM status, HPV status and the deconvoluted score of C1 of malignant epithelial cells (a), expression level of *TFDPI* (b), the top 50 signature score of *POSTN*⁺ fibroblasts (c), the top 50 signature score of *SPPI*⁺ macrophages (d), and the interaction signature score of *POSTN*⁺ fibroblasts and *SPPI*⁺ macrophages (e) in the TCGA-HNSCC data.

2. Authors must also check the reference list and proper citation in the main text. For example, on page 3 lines 91-94 and pages 12-13 lines 502-507, Choi et al. does not match the reference [8].

Response: Thank the reviewer for the careful evaluation and we have modified citation in the text.

Reviewer3

I thank the authors for their revisions and clarifications. The authors have sufficiently addressed my concerns regarding the comparison between Choi et al and the present work through the added text in the introduction and discussion. This text significantly strengthens the work by placing it in context of published literature. I appreciate the additional clarifications regarding methodology and interpretation of results.

Minor comments:

1) There is a grammatical error in line 513: "...were stratifies..."

Response: We have corrected this error in the text.

2) In line 522, the authors state, "...uncovered the underlying mechanism of malignant cell-mediated tumor relapse, and contributed to precise therapeutics for patients with primary and recurrent tumors according to the distinct treatment target selections. This conclusion is overstated and should be toned down.

Response: We have modified inappropriate descriptions in the discussion section (highlighted).

Reference:

1. Cao S, Wendl MC, Wyczalkowski MA, Wylie K, Ye K, Jayasinghe R, et al. Divergent viral presentation among human tumors and adjacent normal tissues. *Sci Rep.* 2016;6:28294.
2. Elbasir A, Ye Y, Schäffer DE, Hao X, Wickramasinghe J, Tsingas K, et al. A deep learning approach reveals unexplored landscape of viral expression in cancer. *Nat Commun.* 2023;14(1):785.